# NELF prevents transcriptional readthrough into DNA replication zones in cancer cells

Chihiro Nakayama [1,2,7], Qi Fang [1,7], Yasukazu Daigaku [3✉], Yuki Aoi [4], Shoko Ito [1], Mami Takahashi[3], Reo Shimatani[1,2], Tamiko Minamisawa[3], Yagiz Ozturk [5], Hiroshi Kimura [6], Ali Shilatifard [4], Michael Tellier [5✉] & Takayuki Nojima [1✉]

## Abstract

**Regulation of RNA polymerase II (Pol II) transcription is closely associated with cell proliferation. However, it remains unclear how the Pol II transcription program is rewired in cancer to promote uncontrolled growth. Here, we find that expression of NELFCD, a known negative transcription elongation factor, is upregulated in colorectal tumors. Auxin-dependent protein degradation of NELF-C in combination with nascent transcript sequencing demonstrates a direct role of NELF-C on Pol II transcription in this cancer. Strikingly, we demonstrate that the acute loss of NELF-C protein globally redistributes termination factors and perturbs Pol II transcription termination. These changes drive pervasive Pol II transcription into DNA replication zones, leading to transcription-replication conflict that may block the cell cycle in G1 or early S phase. Our findings reveal a previously unrecognized role of NELF in transcription termination and highlight NELF as a potential therapeutic target in colorectal cancer.**

**Keywords** DNA Replication; NELF; RNA Polymerase II; Transcription-replication Conflict; Transcription Termination
**Subject Categories** Cancer; Cell Cycle; Chromatin, Transcription & Genomics

## Introduction

Large-scale studies of cancer genome sequencing have revealed important somatic mutations that drive cancer evolution. Perturbation of RNA polymerase II (Pol II) transcription by genetic alterations results in changes to the gene expression program, favoring tumor proliferation. Notably, dysregulation of Pol II transcription in cancer is associated with transcriptional "addiction", a process whereby tumors select specific transcription factors to promote cancer cell growth (Bradner et al, 2017). This

dependence results in a higher sensitivity of a cancer cell to the perturbation of specific transcription factors that could be exploited therapeutically. However, it remains largely unclear how Pol II transcriptional activity differs in tumors as compared to normal tissues.

Multiple steps such as initiation, elongation, and termination contribute to the eukaryotic Pol II transcription cycle (Vervoort et al, 2022). After transcription initiation, Pol II is paused at a promoter-proximal region located at 20–40 nucleotides (nt) downstream of the transcription start site (TSS)(Core et al, 2008; Kwak et al, 2013; Nojima et al, 2015). This step is mainly regulated by two protein complexes; negative elongation factor (NELF) consisting of subunits NELF-A, -B, -C/D, and E, and 5,6-dichloro-1-b-D-ribofuranosylbenzimidazole (DRB)-sensitivity inducing factor (DSIF) comprising SPT4 and SPT5 (Aoi and Shilatifard, 2023; Wu et al, 2003). In vitro transcription assay systems have demonstrated that Pol II transcription elongation is suppressed by adding solely NELF or a combination of NELF and DSIF proteins (Yamaguchi et al, 1999). Pol II is then released from such pausing by the cyclin-dependent kinase 9 (CDK9)-mediated phosphorylation of NELF and SPT5 (Egloff, 2021), which was structurally resolved by recent cryo-electron microscopy (cryo-EM) analysis (Vos et al, 2018a; Vos et al, 2018b). An acute depletion of NELF-C protein results in a shift downstream of Pol II pausing to the +1 nucleosome dyad-associated downstream region of the promoter-proximal region (Aoi et al, 2020). Also, rapid depletion of SPT5 protein causes Pol II degradation, demonstrating that SPT5 contributes to Pol II stabilization (Aoi et al, 2025; Aoi et al, 2021). These findings indicate that NELF and DSIF control Pol II transcription at both pausing and elongation steps (Aoi and Shilatifard, 2023). Furthermore, a recent study proposed that PNUTS-PP1-mediated dephosphorylation of SPT5 contributes to Pol II transcription termination (Cortazar et al, 2019). However, it remains unexplored whether NELF also plays a role in transcription termination.

At the transcript end site (TES), the endonuclease CPSF73, as a part of the cleavage and polyadenylation (CPA) complex, cleaves the nascent RNA at 20-30nt downstream of the polyadenylation signal (PAS, AAUAAA) (Eaton and West, 2020; Proudfoot, 2016).

[1]Medical Institute of Bioregulation, Kyushu University, Fukuoka, Japan. [2]Graduate School of Medical Sciences, Kyushu University, Fukuoka, Japan. [3]Cancer Institute, Japanese Foundation for Cancer Research, Tokyo, Japan. [4]Feinberg School of Medicine, Northwestern University, Chicago, IL, USA. [5]Division of Molecular and Cell Biology, University of Leicester, Leicester, UK. [6]Cell Biology Center, Institute of Integrated Research, Institute of Science Tokyo, Meguro, Tokyo, Japan. [7]These authors contributed equally: Chihiro Nakayama, Qi Fang. ✉E-mail: yasukazu.daigaku@jfcr.or.jp; mt477@leicester.ac.uk; nojima.takayuki.058@m.kyushu-u.ac.jp

This CPA RNA cleavage is essential to recruit the nuclear 5'-3'exonuclease, Xrn2 to the 5'end of the downstream cleaved RNA. Xrn2 degrades nascent RNA produced by the elongating Pol II complex to trigger Pol II removal from chromatin. In addition, CPA RNA cleavage is coupled to mRNA polyadenylation that adds a poly(A) tail to the cleaved upstream RNA. This poly(A) tailed RNA is then released from chromatin into the nucleoplasm. The coupling between RNA cleavage and degradation is critical to transcription termination of protein-coding (pc) genes (Eaton and West, 2018; Sousa-Luís et al, 2021). On the other hand, Pol II transcription of replication-dependent histone (RDH) genes is terminated independently of PAS and Xrn2. Although CPA factors (such as CPSF73, CPSF100, and CstF64) together with stem-loop binding factor and U7 small nuclear RNA (snRNA) cleave downstream of the RDH transcript, no poly(A) tail is added to its 3'end (Eaton and West, 2020; Sun et al, 2020). Another critical RNA cleavage machinery, referred to as the Integrator complex contains INTS11, a homolog of CPSF73. INTS11 cleaves Pol II nascent transcripts of both small and long noncoding RNAs (lncRNAs) as well as 5' end regions of pre-mRNAs (Baillat et al, 2005; Dasilva et al, 2021; Lai et al, 2015; Tatomer et al, 2019; Wagner et al, 2023b).

To dissect mechanisms of Pol II transcription and its coupled RNA processing, several technologies have been developed to detect newly transcribed RNAs (nascent RNAs) (Nojima and Proudfoot, 2022). For example, high-throughput sequencing methods for newly synthesized RNAs labeled with modified nucleotides in isolated nuclei, such as precise run-on sequencing (PRO-seq) (Kwak et al, 2013), showed transcription activity and Pol II pausing. Metabolic RNA labeling with 4-Thiouridine in living cells is also employed to monitor RNA synthesis, called transient transcriptome-sequencing (TT-seq) (Schwalb et al, 2016). Furthermore, we have previously developed the polymerase-intact nascent transcript-sequencing (POINT-seq) (Sousa-Luís et al, 2021) method that profiles TSSs, co-transcriptional RNA cleavage and degradation, co-transcriptional RNA splicing, and read-through transcripts.

Perturbation of transcription termination leads to genomic stresses such as transcription-replication (T-R) conflict, principally caused by collision between transcription and replication machineries (such as PCNA) or between co-transcriptional RNA-DNA hybrid and incoming replication forks (Gómez-González and Aguilera, 2019). An siRNA screen of T-R conflict-suppressing factors in human cells identified CPA factors such as WDR33 that directly bind to the PAS (Teloni et al, 2019). Notably, our previous study showed that siRNA knockdown of the histone chaperone protein SPT6 causes T-R conflict via a genome-wide activation of lncRNA transcription and an impairment of their transcription termination due to a failure of the Integrator complex recruitment to elongating Pol II (Nojima et al, 2018).

In addition to the effect on the replication fork via T-R conflict, a relationship between replication initiation and transcription has been highlighted by multiple studies (Chen et al, 2019; Koyanagi et al, 2022; Petryk et al, 2016). Particularly, DNA replication initiation preferentially occurs at genomics regions that neighbor the TSS and TES of actively transcribing genes. The regulation of replication initiation is divided into multiple steps (Costa and Diffley, 2022). First, the origin recognition complex (ORC) binds DNA at the replication initiation site and subsequently the pre-

loaded replicative helicase MCM2-7 which assembles with CDC6 and CDT1 are recruited to the ORC-bound DNA site. After a double hexamer of MCM2-7 complexes are topologically loaded onto DNA, the activity of cyclin-dependent kinases (CDKs) and DBF4-dependent kinases (DDKs) promote formation of the preinitiation complex which includes some components of the replisome GINS, CDC45, and DNA polymerase as well as TOPBP1 (Dpb11in yeast). Finally, DNA unwinding is initiated, facilitating the formation of the replisome.

In this study, we demonstrate that expression of NELF-C is significantly upregulated in colorectal tumors. To investigate the direct role of NELF-C protein in this colorectal cancer, we employed an auxin-degron system to acutely degrade NELF-C protein in the human colorectal cancer cell line DLD-1. We observe that loss of NELF-C results in a reversible G1 cell cycle arrest. Strikingly, POINT-seq and TT-seq analysis reveal that acute depletion of NELF-C protein induces a transcription termination defect in pc genes. Notably, loss of NELF-C protein generated extended readthrough transcripts, resulting in Pol II invasion into DNA replication initiation zones. Furthermore, NELF-C loss increases the proximity between elongating Pol II and MCM2 or PCNA while also reducing the protein level of the DNA replication licensing factor CDT1 on chromatin, resulting in an aberrant cell cycle G1-S transition and less synthesis of nascent DNAs. Overall, our findings provide insight into NELF function at the gene 3' end. We propose that the NELF complex restricts deleterious transcription readthrough to avoid interference of dysregulated Pol II with DNA replication initiation and/or elongation.

## Results

### Expression of NELFCD transcripts is highly upregulated in colorectal cancer cells

Colorectal tumors are one of the most frequent cancer types worldwide and are associated with patient-specific transcriptional heterogeneity (Dalerba et al, 2011; Lee et al, 2020). In order to understand transcriptional addiction in colorectal cancer, we re-analyzed RNA-seq data from normal (N) and tumor (T) tissue from the large intestine in Genotype-Tissue Expression (GTEx) and The Cancer Genome Atlas (TCGA) databases (Fig. 1A; Table EV1). For this comparative analysis, we focus on a gene category "Pol II transcription cycle" (104 genes) which consists of initiation (I), elongation (E), and termination (T) (Fig. 1B). We found that expression of the NELFCD gene transcript, as well as CPA genes such as CSTF1, CSTF2, and CPSF3, are significantly upregulated in colorectal carcinoma (COAD) and rectum adenocarcinoma (READ), while SUPT4H1 gene (coding SPT4 protein) is unchanged (Figs. 1B and EV1A). Inversely, known inhibitor genes of the cell cycle, such as CDKN1A (coding P21 protein) and CDKN1C (coding P57 protein), are down-regulated in COAD and other cancer types (Fig. EV1B,C), consistent with an increased cellular proliferation. While the most significant upregulation of NELFCD occurs in COAD and READ, we also observed a slight increase in NELFCD expression in other cancer types (Fig. EV1C,D). Similarly, we re-analyzed comparative proteomic data of paired human colon cancer and adjacent normal tissues (Vasaikar et al, 2019), detecting upregulated protein levels of all NELF subunits, but not the SPT4

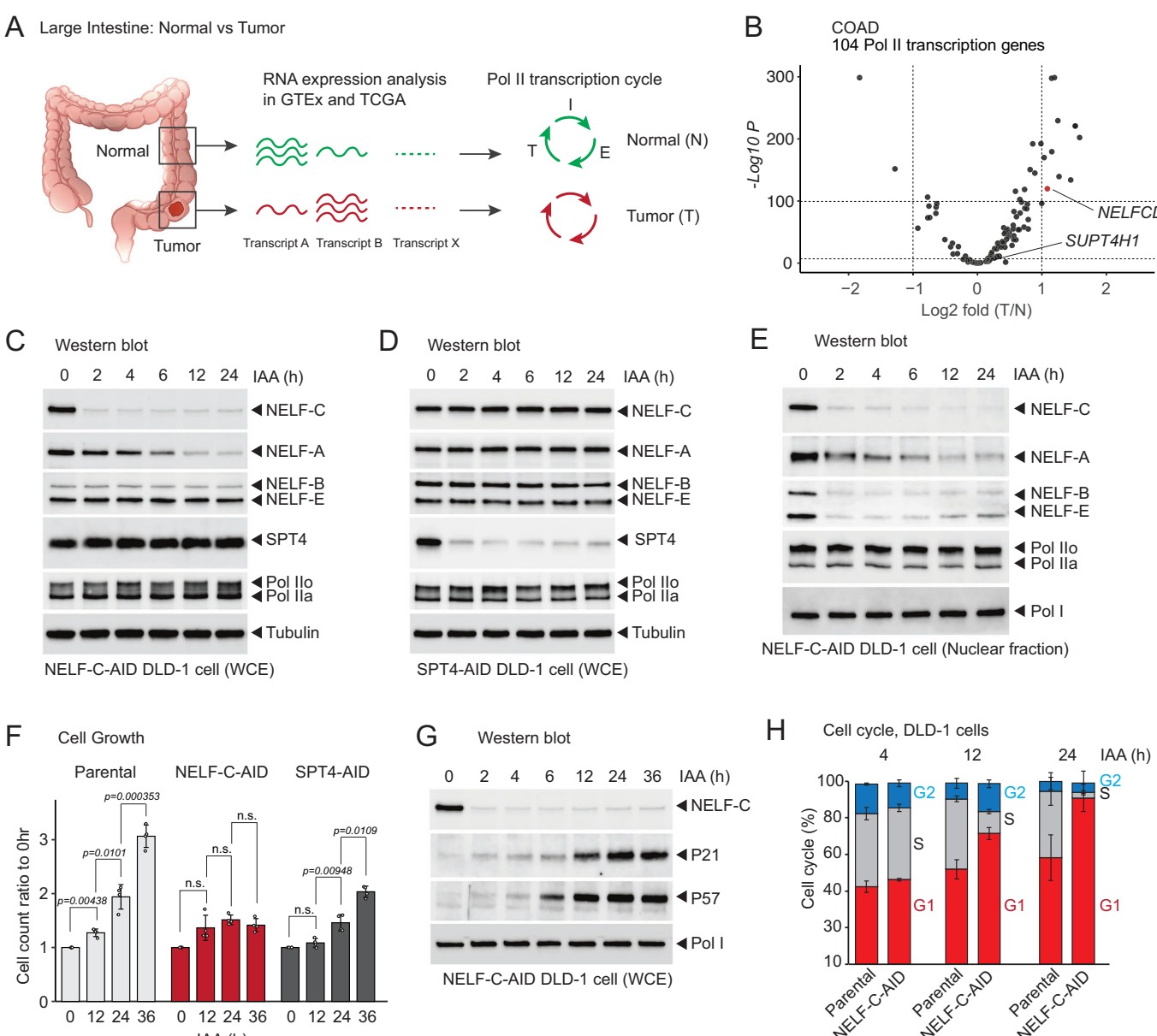

**Figure 1. Expression and cellular function of NELFCD in colorectal cancer cells.**

(A) Schematic diagram of Pol II transcription cycle in normal (N) tissue and tumor (T) of the large intestine. RNA expression level of Pol II transcription cycle-associated genes in GTEx and TCGA databases were analyzed. I initiation, E elongation, T termination. (B) Volcano plot for fold change of T vs N of 104 Pol II transcription-associated genes in COAD. *NELFCD* and *SUPT4H1* genes are indicated with an arrow. *NELFCD* gene is highlighted in red. Statistical test: differential expression analysis was performed using DESeq2 (negative binomial generalized linear model with Wald test); *P* values were adjusted for multiple testing using the Benjamini–Hochberg method (adj.P.Val). The numbers of tumor (T) and normal (N) samples for COAD are provided in Table EV1. (C) Western blot of NELF-C-AID DLD-1 whole-cell extract (WCE) using the indicated antibodies. The treatment time (h) of IAA is also indicated. Western blot image of NELF-C is reused in (G). (D) Western blot of SPT4-AID DLD-1 WCE using the indicated antibodies. The treatment time (h) of IAA is also indicated. (E) Western blot of NELF-C-AID DLD-1 cell nuclear fraction using the indicated antibodies. The treatment time (h) of IAA is indicated. Western blot images of NELF-C and Pol I are reused in Fig. EV2A. (F) Cell count ratios of 12, 24, and 36 h IAA to 0 h IAA are shown. Parental, NELF-C-AID, and SPT4-AID DLD-1 cells are displayed in light gray, red, and dark gray, respectively. Error bars represented the mean ± SEM (biological replicates, *n* = 4). Statistical test: paired two-sample *t* test (Student's *t* test), two-sided, performed within each Group comparing the same Dish across adjacent timepoints (0 h vs 12 h, 12 h vs 24 h, 24 h vs 36 h) on the 0h-normalized ratios. *P* values are shown. n.s. (not significant, *P* > 0.05). (G) Western blot of NELF-C-AID DLD-1 whole-cell extract (WCE) using the indicated antibodies. The treatment time (h) of IAA is indicated. Western blot image of NELF-C is reused in (C). (H) Cell cycle (%) of parental and NELF-C-AID DLD-1 cells following the indicated treatment time (h) of IAA. Error bars represented the mean ± SEM (biological replicates, *n* = 3). Source data are available online for this figure.

protein in primary colon cancer (Fig. EV1E). This result suggests that the Pol II transcription program in READ and COAD is highly dependent on NELF-CD protein (Fig. EV1F). Consequently, we decided to investigate NELF-CD function on the Pol II transcription cycle in colorectal cancer.

## Nuclear NELF complex controls the cell cycle

NELF-C and NELF-D (the nine amino acids shorter version of NELF-C) are transcriptional isoforms with a different N-terminus domain generated from the *NELFCD* gene. To dissect the function of NELF-C/D, we employed the previously established DLD-1 colorectal cancer cells expressing NELF-C or SPT4 proteins that have been tagged with an auxin-inducible degron (AID) on their C-terminus (Aoi et al, 2020; Aoi et al, 2021). Notably, SPT4 is used as a negative control since it is not upregulated in colorectal tumors (Figs. 1B and EV1C). Following addition of auxin (IAA), efficient depletion of NELF-C/D-AID protein (named NELF-C-AID in this study) was detected by western blotting of whole-cell extract (WCE) (Fig. 1C). Similarly, SPT4-AID protein was specifically degraded upon IAA treatment (Fig. 1D). While the degradation of NELF-C protein occurs in less than two hours (2 h), other components of the NELF complex are not affected to the exception of NELF-A (Fig. 1C), which is known to be closely associated with NELF-C (Vos et al, 2018b). However, western blot performed on the nuclear fraction reveals that the depletion of NELF-C results in a rapid nuclear loss of NELF-B and NELF-E and at a slower rate of NELF-A (Fig. 1E), suggesting that NELF-C stabilizes the nuclear NELF complex. Importantly, the loss of the whole NELF complex did not affect the protein and phosphorylation levels of Pol II largest subunit in the nuclear fraction (Fig. 1E).

Long-term degradation of NELF-C protein has been previously found to result in a severe cell growth defect (Aoi et al, 2020). However, it remains unclear whether this is due to cell death or cell cycle arrest. To investigate this further, we performed cell growth assays for 36 h in the absence or presence of IAA in the NELF-C-AID and SPT4-AID cell lines (Fig. 1F). Following IAA treatment, NELF-C-AID cells stopped growing with no detection of dead cells after 12 h (Fig. 1F, red bars) while SPT4-AID cells continued to proliferate, but slower than parental cells (Fig. 1F, gray bars).

To investigate the mechanism behind the growth arrest following NELF-C degradation, we performed western blot of WCE against the known cell cycle inhibitors P21 and P57 (Fig. 1G) and observed a clear increase in the expression of those two proteins after 6–12 h of IAA treatment. NELF depletion generally decreases gene expression with the exception of fly cells (Gilchrist et al, 2008; Williams et al, 2015; Wu et al, 2003). Concomitantly with expression of cell cycle inhibitors and growth arrest after 6–12 h of IAA treatment, we observed an increase in nuclear γH2AX, a DNA damage marker protein by western blot (Fig. EV2A) and immunofluorescence analyses (Fig. EV2B,C).

These observations led us to investigate the effect of NELF-C protein degradation on the cell cycle. Strikingly, our cell sorting analysis displayed a large increase in G1 phase coupled with a concomitant decrease in S-phase cell populations following 12 h IAA treatment (Fig. 1H). To investigate whether the effect of NELF-C loss on cell growth and cell cycle is reversible, we performed washout experiments where following 24 h of pre-treatment with IAA to degrade NELF-C, IAA was replaced by

DMSO (washout) and the cells were monitored for 48 h post-treatment (Fig. 2A). By western blot, we observed a partial and then total recovery of NELF-C and NELF-A protein levels at 24 and 48 h, respectively (Fig. 2B). Simultaneously to the recovery of NELF protein levels, we also observed a complete reduction to background levels of P21 and P57 proteins from 24 h post-treatment (Fig. 2B). This loss of the cell cycle inhibitors in the washout condition is also associated with a partial recovery in the growth rate after 48 h (Fig. 2C) and a complete recovery of the proportion of cells in G1 and S phases at 24 h post-treatment (Fig. 2D). Together, our data indicate that the nuclear NELF complex is required for establishment of an efficient cell cycle progression.

## Acute NELF-C depletion induces a Pol II transcription termination failure

In order to dissect the function of NELF and DSIF complexes on elongating Pol II in colorectal cancer cells, we employed POINT-seq technology in NELF-C-AID and SPT4-AID DLD-1 cells. POINT analysis involves the isolation of nascent Pol II transcripts by immunoprecipitation with Pol II CTD antibody from stringently purified chromatin fraction using 1 M Urea and 3% Empigen detergent (Fig. 3A). Our POINT-seq method detected a substantial transcription readthrough of pc gene, *RPS23* in control DMSO-treated DLD-1 cells (Fig. 3B). Importantly, 4 h IAA treatment in NELF-C-AID cells significantly extended such transcription readthrough (Fig. 3B). We also performed a time-course experiments (0, 10 m, 20 m, 30 m, 40 m, and 4 h) of IAA treatment in NELF-C-AID cells (Fig. EV3A). Notably, ~40% (Pol I normalized, on average, *n* = 3) of NELF-C protein was detected in 40 m IAA, while none was detected in 4 h IAA in the nuclear fraction of NELF-C-AID cells. To investigate termination defect at indicated time points in NELF-C loss, we performed RT-qPCR with primers for introns and downstream of *RPS23*, *HELLS*, and *CLIC4* genes (Fig. EV3B). Our RT-qPCR analysis showed a large increase in the ratio (DoG/intron) at 4 h IAA in NELF-C-AID cells, indicating that this analysis recapitulated the termination defect as POINT-seq. However, we detected slight increases in the ratio (DoG/intron) at both 20 m and 40 m IAA in NELF-C-AID cells. These results suggest that NELF-C may be present in excess in the cancer cells, as the transcription termination machinery remains functional even with a partial reduction in NELF-C levels. On the other hand, 3 h SPT4 depletion decreases POINT-seq signals throughout the transcription unit, indicating that SPT4 enhances transcription elongation. These transcriptional effects are detected genome-wide by metagene analysis of expressed pc genes (Fig. 3C, *n* = 17,963). To determine the extent of the transcription termination defect, we quantified a ratio of POINT-seq signals in the 2.5 kb downstream region of PAS over the gene body (TSS to TES) to determine a termination index (TI). In two biological replicates, our POINT-seq analysis detected higher TI at 4 h IAA than at 4 h DMSO treatments in NELF-C-AID DLD-1 cells, but not in SPT4-AID cells (Fig. 3D).

We also performed the TI analysis in human colorectal cancer cell line, HCT116 which expresses AID-tagged CPSF73 and XRN2 proteins, that are major transcription termination factors of pc genes and found a similar trend of termination defect across all expressed genes. Precisely, the TI level at 4 h NELF-C loss is similar, albeit slightly less increased, to CPSF73 loss, but lower than Xrn2 loss (Fig. EV3C). Thus, we conclude that NELF-C is required

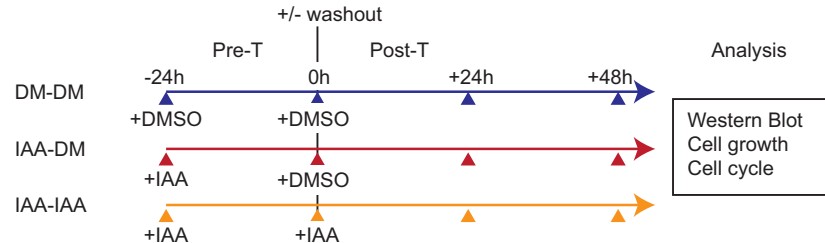

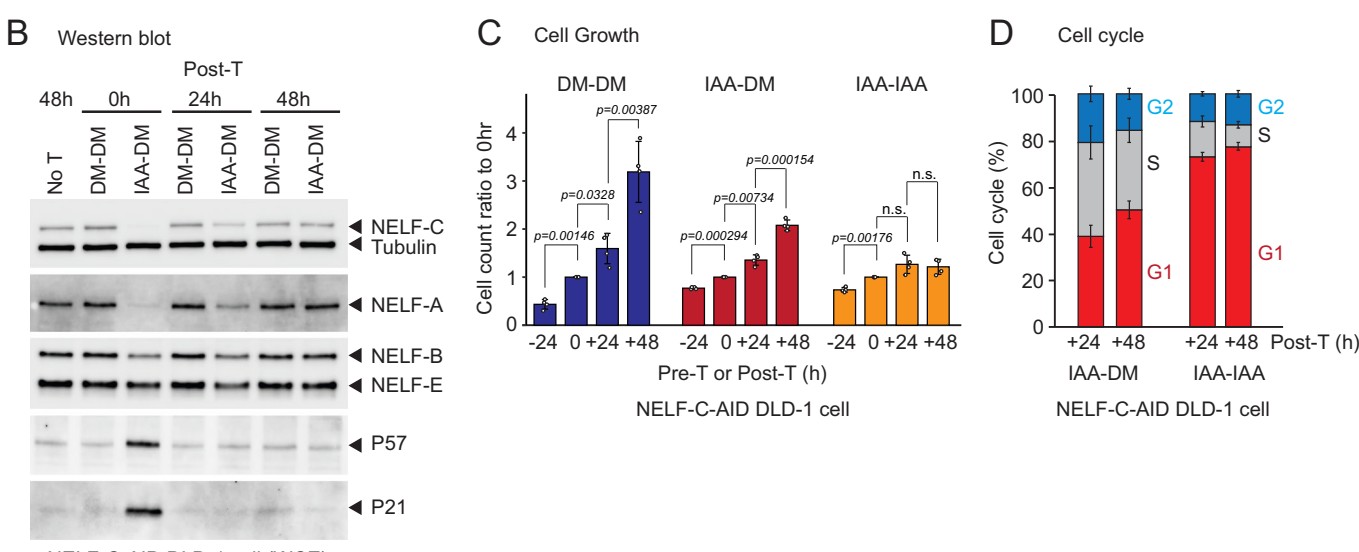

**Figure 2. NELF-C reversibly controls the cell cycle.**

(**A**) Schematic timeline of pre-treatment (Pre-T, 24 h) and post-treatment (Post-T, up to 48 h) followed by western blot, cell growth, and cell cycle analyses. For washout, DMSO (DM) or IAA were used. (**B**) Western blot of NELF-C-AID DLD-1 WCE using the indicated antibodies. No treatment (No-T) and post-treatment time (h) of IAA is indicated. (**C**) Cell count ratios of -24 (Pre-T), +24 (Post-T), and +48 h (Post-T) DMSO or IAA to 0 h (washout) are shown. DM-DM, IAA-DM, and IAA-IAA are displayed in blue, red, and orange, respectively. Error bars represented the mean ± SEM (biological replicates, $n = 4$). Statistical test: paired two-sample $t$ test (Student's $t$ test), two-sided, performed within each Group comparing the same Dish across timepoints (−24 h vs 0 h, 0 h vs 24 h, 24 h vs 48 h). $P$ values are shown. n.s. (not significant, $P > 0.05$). (**D**) Cell cycle (%) of NELF-C-AID DLD-1 cells (Pre-T, 24 h) in the indicated Post-T time (h) of DM or IAA. Error bars represented the mean ± SEM (biological replicates, $n = 3$). Source data are available online for this figure.

for Pol II termination of pc genes, independently of the DSIF protein SPT4.

As a defect in pre-mRNA splicing is known to also promote a transcription termination defect (Dye and Proudfoot, 1999; Reimer et al, 2021), we investigated the effect of NELF-C protein on co-transcriptional splicing. Pladienolide B (PlaB), which inhibits early stages of splicing reaction through the binding to the U2 snRNP components SF3B1 protein, was used to evaluate co-transcriptional splicing. A 3 h treatment of PlaB did not affect protein levels of NELF-C and Pol II in WCE of NELF-C-AID DLD-1 cells (Fig. EV3D). As previously reported, PlaB induced premature transcription termination (PTT) in pc genes, accompanying an inhibition of co-transcriptional splicing (Caizzi et al, 2021; Sousa-Luís et al, 2021). In addition, we found in these experiments that NELF-C degradation by 3 h IAA treatment in addition to PlaB result in an extension of PTT transcripts in *FAM120B* gene (Fig. EV3E). Notably, co-transcriptional constitutive splicing was unaffected by NELF-C depletion, although PlaB dramatically

inhibits this splicing fraction (Fig. EV3F). These results indicate that the NELF complex contributes to Pol II transcription termination independently of co-transcriptional splicing.

NELF depletion caused a cell cycle arrest that started from 12 h IAA treatment of NELF-C-AID DLD-1 cells (Fig. 1H). This led us to analyze nascent RNAs at longer time points. While our POINT-seq data detected a Pol II termination defect of *RPS23* and *HELLS* genes at 4 h IAA, POINT-seq signals at 12 h and 24 h IAA were significantly reduced throughout their gene bodies (Fig. 3E). Metagene analysis of POINT-seq confirmed a global termination defect after 4 h IAA treatment and the global decrease in transcription after 12 and 24 h IAA (Fig. 3F). In addition, compared to 0 h IAA, higher TIs of POINT-seq were detected at 4 h IAA, but not at 12 h and 24 h IAA (Fig. 3G). Similarly, re-analysis of a spike-in PRO-seq of NELF-B-dTAG in HAP1 cells (Santana et al, 2024) shows a widespread termination defect on non-overlapping and expressed pc genes ($n = 3459$) after 2 h of a degradation inducer VHL treatment (Fig. EV3G). Our results

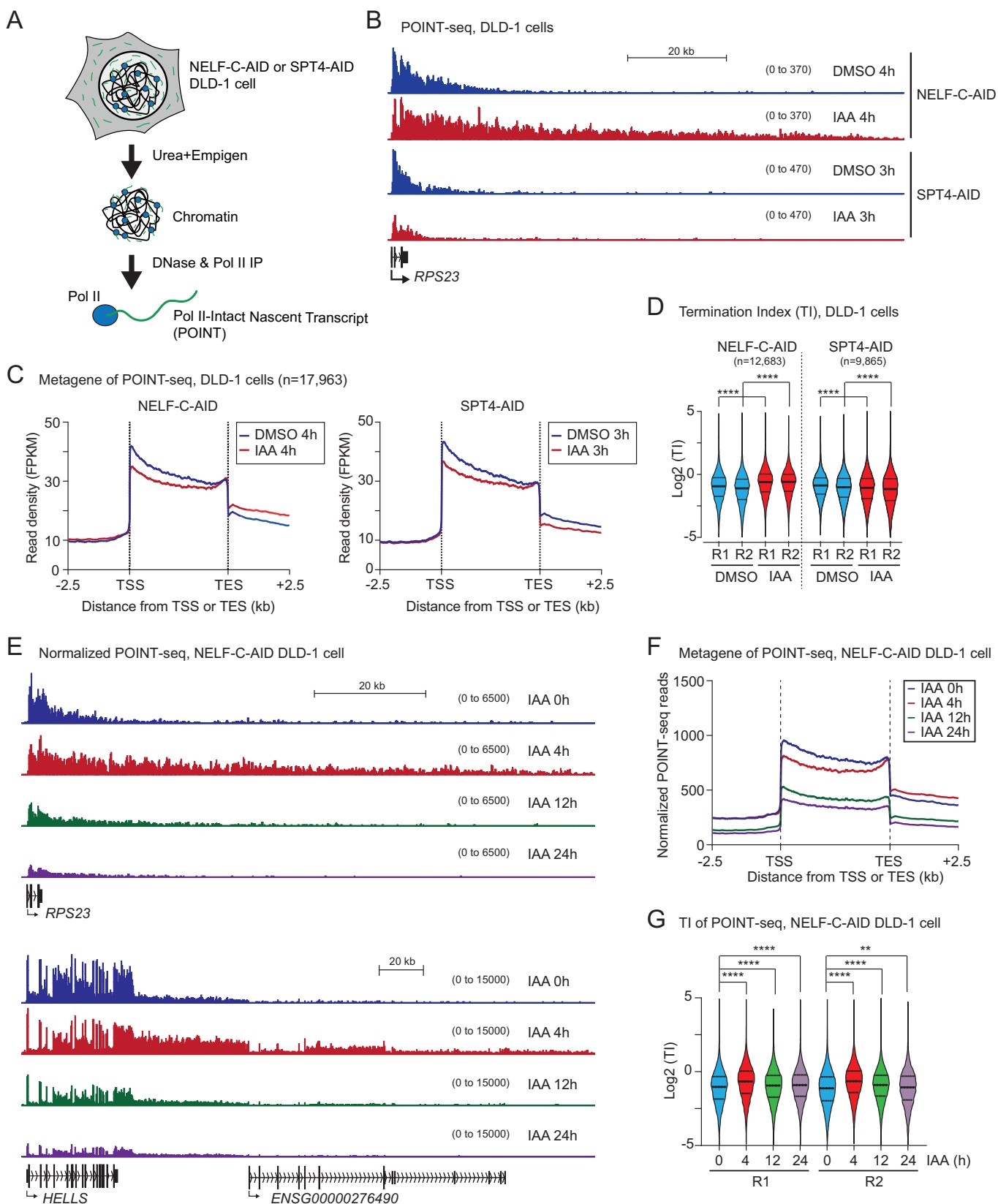

**A** NELF-C-AID or SPT4-AID DLD-1 cell

Urea+Empigen

Chromatin

DNase & Pol II IP

Pol II

Pol II-Intact Nascent Transcript (POINT)

**B** POINT-seq, DLD-1 cells

20 kb

(0 to 370) DMSO 4h
(0 to 370) IAA 4h
NELF-C-AID

(0 to 470) DMSO 3h
(0 to 470) IAA 3h
SPT4-AID

*RPS23*

**C** Metagene of POINT-seq, DLD-1 cells (n=17,963)

NELF-C-AID

SPT4-AID

**D** Termination Index (TI), DLD-1 cells

NELF-C-AID (n=12,683)   SPT4-AID (n=9,865)

**E** Normalized POINT-seq, NELF-C-AID DLD-1 cell

20 kb

(0 to 6500) IAA 0h
(0 to 6500) IAA 4h
(0 to 6500) IAA 12h
(0 to 6500) IAA 24h

*RPS23*

20 kb

(0 to 15000) IAA 0h
(0 to 15000) IAA 4h
(0 to 15000) IAA 12h
(0 to 15000) IAA 24h

*HELLS*   *ENSG00000276490*

**F** Metagene of POINT-seq, NELF-C-AID DLD-1 cell

**G** TI of POINT-seq, NELF-C-AID DLD-1 cell

**Figure 3. Acute NELF depletion induces a Pol II transcription termination failure.**

(A) Schematic diagram of POINT-seq strategy. Chromatin of NELF-C-AID or SPT4-AID DLD-1 cells was stringently isolated using Urea and Empigen detergent. Pol II intact nascent transcript (POINT) was precipitated with Pol II CTD antibody from DNA-digested chromatin fraction. (B) Example view of POINT-seq on *RPS23* gene in the indicated cell lines treated for 4 or 3 h with DMSO or IAA. (C) Metagene analysis of POINT-seq on scaled transcription unit −/+2.5 kb in the indicated cell lines treated for 4 or 3 h with DMSO or IAA. (D) Violin plots of Termination Index (TI) in the indicated cell lines treated for 4 or 3 h with DMSO or IAA. Two biological replicates are shown. Statistical test: Wilcoxon signed-rank test. ****$P < 0.0001$. Violin plot: minimal-to-maximal value, box center line: median. (E) Example view of SIRV-normalized POINT-seq on *RPS23* and *HELLS* genes in NELF-C-AID DLD-1 cells (0, 4, 12, and 24 h IAA). (F) Metagene analysis of POINT-seq on scaled transcription unit −/+2.5 kb in NELF-C-AID DLD-1 cells (0, 4, 12, and 24 h IAA). (G) Violin plots of TI in NELF-C-AID DLD-1 cells (0, 4, 12, and 24 h IAA). Two biological replicates are shown. Statistical test: Wilcoxon signed-rank test. **$P = 0.0021$, ****$P < 0.0001$. Violin plot: minimal-to-maximal value, box center line: median.

suggest that loss of NELF complex directly impacts Pol II transcription termination and promotes nascent readthrough RNA synthesis downstream of the PAS.

## Pol II transcription termination may be connected to promoter proximal Pol II pausing

To determine whether there is a link between promoter proximal Pol II pausing and transcription termination defect, we re-analyzed the previously published PRO-seq data. Indeed, we detected a shift of Pol II pausing downstream in non-overlapping and highly expressed pc genes (Fig. EV4A), as previously reported (Aoi et al, 2020). We also measured Pol II pausing index, which is calculated based on pausing level ratioed to the amount of elongating Pol II across the gene body. NELF-C depletion reduces paused Pol II at promoter proximal region but reduces Pol II even more across the gene body, resulting in a higher Pol II pausing index in 4 h IAA condition (Fig. EV4B). Next, to investigate promoter activity, we employed the ATAC-seq method at 4 h and 24 h IAA treatments. ATAC-seq profiles (Fig. EV4C) upon NELF-C depletion demonstrated that ATAC-seq signals are increased at 4 h IAA on promoter regions, suggesting that rapid NELF-C loss enhances transcription initiation. Additionally, consistent with decreased POINT-seq after 24 h IAA, this indicates that stable loss of NELF-C suppresses transcription initiation. These observations demonstrate that NELF-C regulation of Pol II pausing at promoter-proximal sites may be connected to its effect on transcription termination.

Based on TI, we classified genes into two categories: NELF-Dependent termination (NDT, $n = 2484$) and NELF-Independent termination (NIT, $n = 975$) genes (Fig. EV5A) to further investigate what factors could contribute to transcription termination defect after NELF-C depletion. We analyzed gene length (Fig. EV5B), pA+RNA expression level (Fig. EV5C), and pausing index (Fig. EV5D,E) in both NDT and NIT gene categories. Although NIT genes show a slightly longer gene length than NDT genes (Fig. EV5B), RNA expression level and pausing index are similar for both categories. However, gene classification with TI may not be an appropriate approach to dissect NELF-controlled transcription termination as we have to keep only genes that are expressed at a high level and located more than 2.5 kb away from an upstream and/or downstream gene as these parameters result in false-positive and false-negative results in the TI calculation.

## NELF may control Pol II elongation to recruit transcription termination factors

As transcription elongation activity (EA) is known to affect transcription termination (Muniz et al, 2021), we next investigated

whether Pol II EA correlates with TI change. In nascent RNA analysis, a CDK9 inhibitor treatment showed transcription inhibition waves, indicating that Pol II pause-release step is affected by CDK9 inhibition, but not the elongation step (Jonkers et al, 2014). In this study, we measured the EA by performing TT-seq with a specific CDK9 inhibitor NVP-2 for 0.5 h and 1 h (Fig. EV6A,B). An example of the transcription inhibition waves, and the stronger effect on the inhibition waves at 4 h NELF-C loss, are visible on *PTPN13* gene (Fig. EV6A). The global EA analysis demonstrated that NELF-C loss significantly increases Pol II transcription EA on genes longer than 142 kb (+17% in median), whereas we did not observe this increase on shorter genes (60-141 kb) due to less signals detected in the first 100 kb following 1 h CDK9 inhibition (Fig. EV6C). Our comparative analysis showed a positive, but limited ($R = 0.19$), correlation between increased Pol II EA and termination defect only in very long genes (Fig. EV6D). It should be noted that our Pol II EA result should be supported by additional methods such as DRB block/release approach (Fuchs et al, 2014; Gregersen et al, 2020; Saponaro et al, 2014), since the difference in Pol II EA between NELF-depleted and control cells is small, compared to the ones observed in CDK9 inhibition (Booth et al, 2018).

To investigate the EA-TI relationship in most pc genes, we measured Pol II elongation index (EI). We estimated Pol II EI for short-long genes by a previously established method with modification (Caizzi et al, 2021; Gressel et al, 2017; Zumer et al, 2021). Pol II EI was determined by the ratio of newly synthesized RNAs which are labeled with 4sU (TT-seq signals) over intact nascent RNAs which are derived from elongating Pol II (POINT-seq signals), across gene body (Fig. 4A). We observed significantly less POINT-seq signals in non-overlapping pc genes in NELF-C-depleted cells (IAA 4 h), compared to DMSO control (Fig. EV6E, left). On the other hand, there are no or little differences in TT-seq signals between DMSO and IAA 4 h (Fig. EV6E, middle). We then calculated Pol II EI (TT-seq/POINT-seq and TT-seq/PRO-seq) for non-overlapping pc genes (Fig. EV6E, right, F). Importantly, both measurements of Pol II EI (TT-seq/POINT-seq and TT-seq/PRO-seq) show a clear and significant increase in Pol II EI in the gene body. Notably, Pol II EI hugely drops after 3′ end of pc genes (TES or PAS), supporting the previous observation of Pol II slowing down after the PAS (Cortazar et al, 2019). Also, we found that NELF-C loss increases Pol II EI across genes. This is consistent with the higher elongation rate observed in the "TT-seq + NVP-2" experiment that has a limited number of genes. These results also suggest that the higher Pol II speed after NELF-C loss is more general and affects most of the expressed pc genes (not only long genes >142 kb). However, unexpectedly, Pol II EI is decreased before TES. This trend is the same for all expressed pc genes

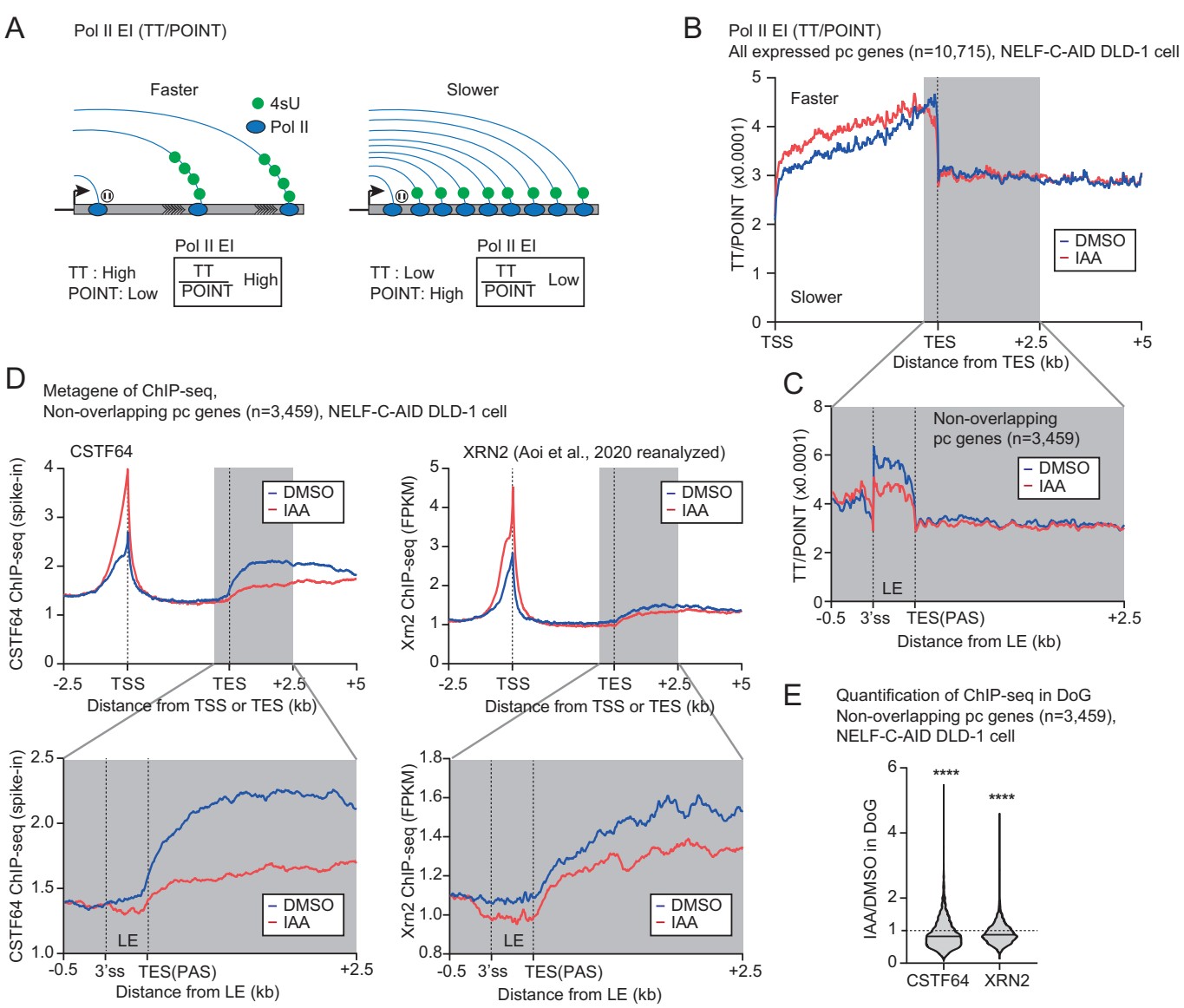

**Figure 4. NELF may control transcription termination via elongation activity and recruitment of termination factors at 3′ ends of genes.**

(A) Schematic diagram of Pol II elongation index (EI) analysis. Pol II EI is determined as TT/POINT. 4sU (Green circle), Pol II (blue oval). (B) Pol II EI for all expressed pc genes across normalized transcription units in NELF-C-AID DLD-1 cells treated for 4 h with DMSO or IAA. Last exon (LE)-0.5 kb ~ +2.5 kb window is highlighted in gray. (C) Pol II EI for non-overlapping pc genes across LE. (D) Metagene of CSTF64 and XRN2 ChIP-seq for non-overlapping pc genes across normalized transcription units (top) and LE (bottom) in NELF-C-AID DLD-1 cells treated for 4 h with DMSO or IAA. Last exon (LE)-0.5 kb ~ +2.5 kb window is highlighted in gray. (E) Quantification of CSTF64 and XRN2 ChIP-seq signals (IAA/DMSO) in 2.5 kb DoG regions of non-overlapping pc genes. Statistical test: Kruskal–Wallis test. ****P < 0.0001. Violin plot: minimal-to-maximal value, box center line: median.

(Fig. 4B) and for three gene classes, which are determined by gene size (<21, 21–87, and >87 kb) (Fig. EV6G). Since Pol II EI drops just before PAS upon NELF-C depletion, we next checked the profile over last exons (LEs) of non-overlapping pc genes (Fig. 4C). The analysis shows a reduction in Pol II EI across LE after NELF-C loss, suggesting that elongating Pol II prematurely slows down near PAS (or LE).

To determine whether the change in Pol II EA at the end of pc genes after NELF-C depletion affects the recruitment of termination factors, we performed and re-analyzed ChIP-seq of CPA (i.e., CSTF64), XRN2, and Integrator (i.e., INTS3) proteins. ChIP-seq

analyses show that both the CSTF64 and the XRN2 signals are significantly increased at TSS after NELF-C depletion. Furthermore, the CSTF64 signal is greatly decreased, and the XRN2 signal is slightly decreased downstream of non-overlapping pc genes by 4 h NELF-C loss (Fig. 4D, top). Importantly, the decreases are starting across LE (Fig. 4D, bottom). Together with quantification of ChIP-seq signals in the DoG region (Fig. 4E), these results suggest that NELF-C depletion fails to recruit CSTF64, and possibly XRN2, at the LE/PAS, leading to Pol II transcription termination defect. On the other hand, INTS3 ChIP-seq analysis shows that its signal is slightly increased across PAS of non-overlapping pc genes

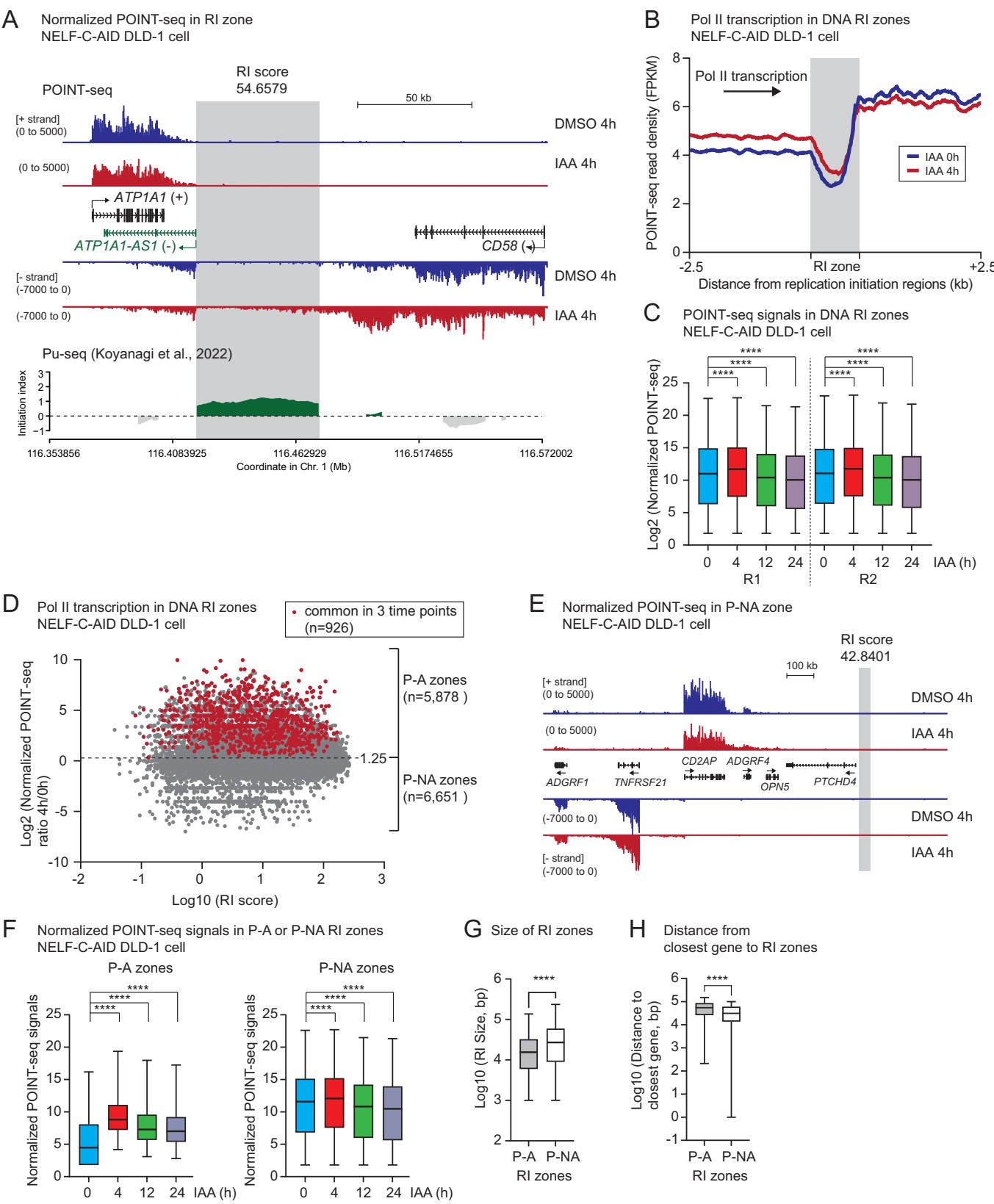

**A** Normalized POINT-seq in RI zone
NELF-C-AID DLD-1 cell

**B** Pol II transcription in DNA RI zones
NELF-C-AID DLD-1 cell

**C** POINT-seq signals in DNA RI zones
NELF-C-AID DLD-1 cell

**D** Pol II transcription in DNA RI zones
NELF-C-AID DLD-1 cell

**E** Normalized POINT-seq in P-NA zone
NELF-C-AID DLD-1 cell

**F** Normalized POINT-seq signals in P-A or P-NA RI zones
NELF-C-AID DLD-1 cell

**G** Size of RI zones

**H** Distance from
closest gene to RI zones

**Figure 5. NELF loss causes Pol II transcription invasion into the DNA replication initiation zone.**

(A) Example view of SIRV-normalized POINT-seq on *CD58* gene adjacent to a RI zone in NELF-C-AID DLD-1 cells (4 h DMSO and IAA). DNA replication initiation (RI) zone is highlighted in gray. RI score: 54.6579. Re-analyzed Pu-seq data shows RI zones in green. (B) Metagene analysis of POINT-seq on scaled RI zones $-/+2.5$ kb in NELF-C-AID DLD-1 cells (4 h DMSO and IAA). (C) Box plots of SIRV-normalized POINT-seq signals in RI zones in NELF-C-AID DLD-1 cells (0, 4, 12, and 24 h IAA). Two biological replicates are shown. Statistical test: Wilcoxon signed-rank test. ****$P < 0.0001$. Box plot: minimal-to-maximal value, box center line: median, bounds of box: interquartile (25 and 75%). (D) Scatter plots of SIRV-normalized POINT-seq signals and RI score in NELF-C-AID DLD-1 cells (4 h DMSO and IAA). Cut-off value is $Log_2(1.25)$ in SIRV-normalized POINT-seq. Pol II-affected (P-A) and not affected (P-NA) zones are classified as higher and lower than the cut-off value, respectively. RI zones that are above the cut-off in the three timepoints (4, 12, 24 h IAA) are indicated in red. (E) Example view of SIRV-normalized POINT-seq on *CD2AP* gene adjacent to a P-NA zone in NELF-C-AID DLD-1 cells (4 h DMSO and IAA). DNA RI zone is highlighted in gray. RI score: 42.8401. (F) Box plots of SIRV-normalized POINT-seq signals in P-A ($n = 5878$) and P-NA ($n = 6651$) RI zones in NELF-C-AID DLD-1 cells (0, 4, 12, and 24 h IAA). Statistical test: Wilcoxon rank-sum test. ****$P < 0.0001$. Box plot: minimal-to-maximal value, box center line: median, bounds of box: interquartile (25 and 75%). (G) Size in bp of P-A ($n = 5878$) or P-NA ($n = 6651$) RI zones. Statistical test: Wilcoxon rank-sum test. ****$P < 0.0001$. Box plot: minimal-to-maximal value, box center line: median, bounds of box: interquartile (25 and 75%). (H) Distance in bp between the closest gene and the P-A ($n = 5878$) and P-NA ($n = 6651$) RI zones. Statistical test: Wilcoxon rank-sum test. ****$P < 0.0001$. Box plot: minimal-to-maximal value, box center line: median, bounds of box: interquartile (25 and 75%).

(Fig. EV6H), suggesting the Integrator complex could be more recruited to promote Pol II transcription termination. However, re-analysis of PRO-seq data with shINTS11 depletion in HeLa cells fails to detect an increase of TI in the top 25% genes affected by NELF-C degradation (Fig. EV6I). This result indicates that Integrator is not involved in NELF-mediated Pol II transcription termination.

## NELF loss causes Pol II transcription invasion into DNA replication initiation zone

DNA replication origin mapping using 5-ethynyl-2'-deoxyuridine (EdU)-labeled DNAs has demonstrated that nascent DNA synthesis is preferentially initiated in the intergenic regions of human U2OS and HeLa cell lines (Macheret and Halazonetis, 2018). Consistent with this study, another technology to map replicative DNA polymerases, called Pu-seq (polymerase usage-sequencing) maps DNA replication initiation and termination zones over genomic regions with reciprocal demarcations in usage of leading and lagging strand polymerase in HCT116 human colorectal cancer cell (Koyanagi et al, 2022). Notably, Pu-seq technology also predicted that sites of replication initiation are generally located in intergenic regions around highly transcribed pc genes. This suggests that Pol II transcription and DNA replication are mutually exclusive in human cancer cells. In fact, perturbation of Pol II transcription termination causes T-R conflict, a major source of genomic stress in human cells (Nojima et al, 2018; Teloni et al, 2019).

Based on the DNA replication initiation (RI) score (from 0.0363 to 265.5028) determined by quantification of the Pu-seq signals, 12,529 DNA RI zones were identified. To investigate Pol II transcription activity in the RI zone, we compared the POINT-seq signals at 4 h IAA to that of 4 h DMSO treatments in NELF-C-AID cells. As a representative example, POINT-seq signals in the *CD58* gene are displayed (Fig. 5A). The mapped RI zone (score 54.6549) is located downstream of the *CD58* gene and upstream of *ATP1A1-AS1* lncRNA gene. Importantly, both genes are transcriptionally active. The POINT-seq technology detected extended readthrough transcripts into the RI zone only in the absence of NELF-C protein (4 h IAA) (Fig. 5A). Similarly, readthrough transcripts were detected in the RI zone (score 22.7711) downstream of another example gene *FOCAD*, but again only in the absence of NELF-C protein (Fig. EV7A). *IFN* genes in the cluster in or near this RI region were encoded on the opposite strand of the *FOCAD* gene. However, *IFN* genes are not expressed. The generality of Pol II

invasion in the RI regions after 4 h IAA is demonstrated in metagene analysis (Fig. 5B) and quantification of POINT-seq (Fig. 5C). A stable loss of NELF-C displayed decreased POINT-seq signals in the RI regions (12 h and 24 h IAA), in agreement with the global decrease in transcription (Fig. EV7B). As expected from the lack of a transcription termination defect, acute loss of SPT4 protein showed no significant increase of POINT-seq signals in the RI zones (Fig. EV7C). To investigate whether Pol II termination defect generally increases transcription levels in the RI zones, we re-analyzed published TT-seq data in DLD-1 cells, which show termination defect after 1 h PNUTS depletion (Wang et al, 2024). Like loss of NELF-C, loss of PNUTS increases TT-seq signal in the RI zones (Fig. EV7D). In addition, PlaB treatment dramatically reduced the POINT-seq signals in the RI zone (Fig. EV7E), since PlaB causes PTT in over 20% of expressed pc genes of HCT116 cells (Sousa-Luís et al, 2021).

To extract the RI zones that are potentially perturbed by dysregulated Pol II transcription readthrough in the absence of NELF-C protein, we classified them into two categories using the ratio of 4 h to 0 h IAA POINT-seq signals (cut-off: 1.25): the Pol II-affected (P-A, $n = 5878$) and not affected (P-NA, $n = 6651$) zones (Fig. 5D). Common P-A zones between 4, 12, and 24 h IAA are shown in red ($n = 926$). This result suggests that not all the RI zones are occupied by elongating Pol II upon loss of NELF complex. For example, Pol II transcription termination was impaired in the representative pc gene *CD2AP*. However, the readthrough transcription did not reach to the neighboring RI zone (score 42.8401) (Fig. 5E). As an example of a P-NA zone, Pol II termination was not greatly perturbed for the *SPOPL* gene after acute loss of NELF-C protein, resulting in no POINT signal accumulation in the downstream RI zone (score 41.0896) (Fig. EV7F). Quantification of TI (4 h vs 0 h IAA) in NELF-C-AID DLD-1 cells demonstrated that pc genes adjacent to the P-A zones has a higher TI compared to genes close to P-NA zones (Fig. EV7G). As expected, POINT-seq signals were increased in P-A zones following NELF-C depletion, although those were unchanged in P-NA zones (Fig. 5F). Notably, POINT-seq signals in P-NA zones is higher than that in P-A zones. This may be due to the larger size of RI zones in P-NA category (Fig. 5G). In addition, we measured the distance between the closest gene and each RI zone, showing that P-A zones are more separated from the transcription unit than P-NA zones (Fig. 5H). This suggests that P-A zones need to be protected from Pol II transcription invasion. Indeed, 1039 RI zones were identified in

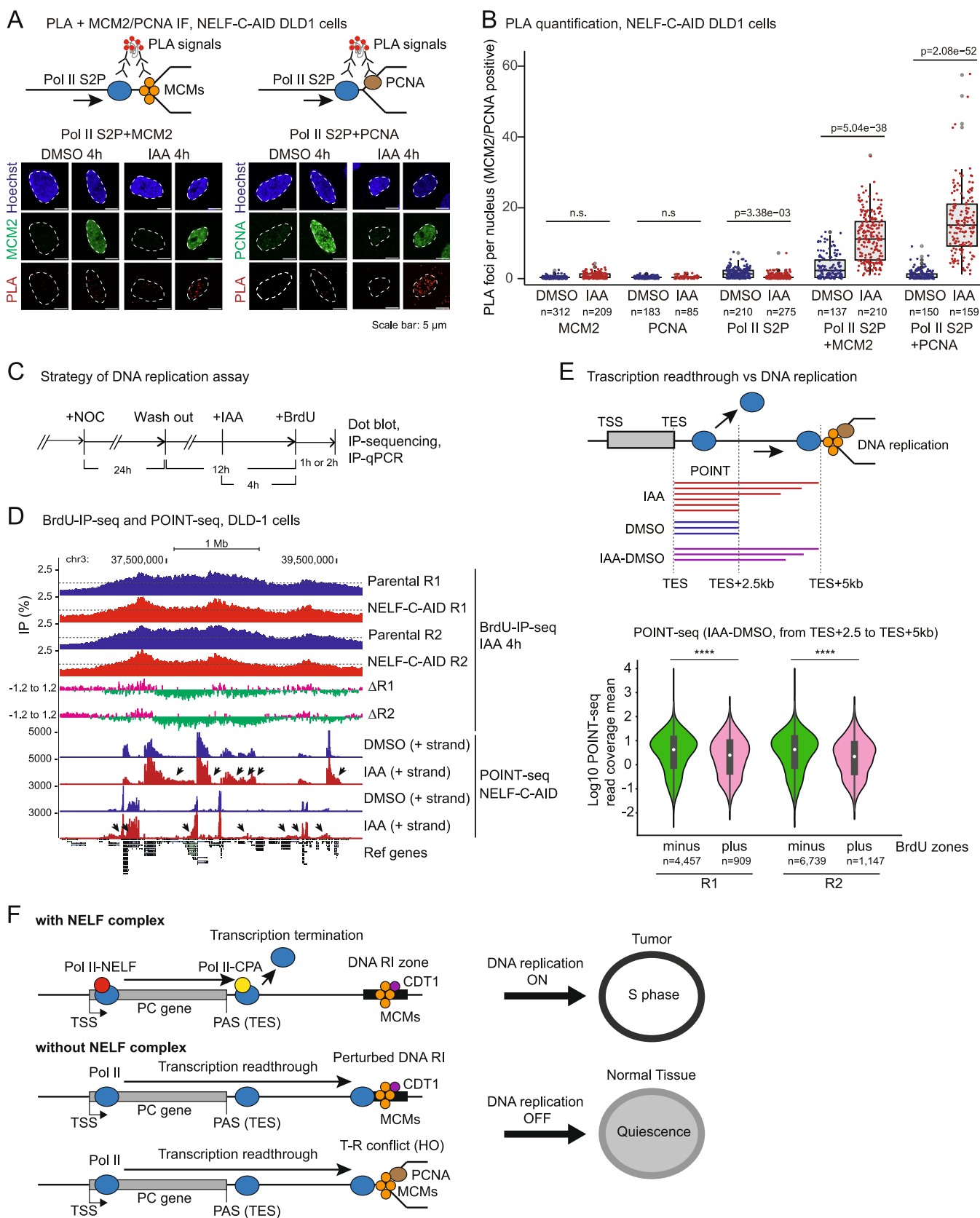

A  PLA + MCM2/PCNA IF, NELF-C-AID DLD1 cells

B  PLA quantification, NELF-C-AID DLD1 cells

C  Strategy of DNA replication assay

D  BrdU-IP-seq and POINT-seq, DLD-1 cells

E  Trascription readthrough vs DNA replication

POINT-seq (IAA-DMSO, from TES+2.5 to TES+5kb)

F  with NELF complex

without NELF complex

**Figure 6. Acute depletion of NELF-C protein causes a conflict between transcription and replication and suppresses DNA replication.**

(A) Representative images of PLA with the indicated antibodies after 4 h DMSO and IAA in MCM2 (right) or PCNA (left) expressing NELF-C-AID DLD-1 cells. Blue: Hoechst, Green: MCM2 or PCNA, Red: PLA. Scale bar size is 5 µm. (B) Box plots of the PLA foci per nucleus after 4 h DMSO and IAA in MCM2 or PCNA expressing NELF-C-AID DLD-1 cells. The PLA was performed with the indicated antibodies. Box plots show the median (center line) and interquartile range (box, 25th–75th percentiles); whiskers indicate 1.5×IQR. Dots represent individual nuclei. Statistical test: Brunner–Munzel test. P values are shown. n.s. (not significant). (C) Schematic representation of the BrdU-IP assay performed following release from M phase. (D) BrdU-IP-seq of two biological replicates (R1 and R2) for the indicated chromosome 3 window in parental and NELF-C-AID DLD-1 cells treated with IAA for 4 h. IP efficiencies (IP/input) are shown. BrdU plus (magenta) and minus (green) zones in Parental minus NELF-C-AID (Δ). Dashed lines indicate 1.25 in BrdU-IP (%). Normalized POINT-seq profiles for (+) and (−) strands in NELF-C-AID DLD-1 cells treated with IAA for 0 h (DMSO) or 4 h (IAA) are shown. Pol II transcription termination defect are indicated by arrows. (E) Quantification of the POINT-seq (IAA 4 h – DMSO) signal over minus and plus BrdU zones located downstream of expressed pc genes (from TES + 2.5 to TES + 5 kb). Violin plots show the distribution (kernel density) of log10 POINT-seq read coverage mean values. Embedded box plots indicate the median (center; white dot/line) and the interquartile range (box; 25th–75th percentiles). Whiskers extend to 1.5×IQR; whisker endpoints represent the minimum and maximum values within this range. Statistical test: Brunner–Munzel test. ****$P < 0.0001$. (F) Model of NELF-mediated transcription termination and DNA replication initiation or elongation. With NELF, Pol II transcription is terminated at proximal PAS before Pol II reaches DNA RI zone. Without NELF, Pol II transcription is extended to distal PAS and impair replication in the RI zone. This perturbs DNA RI and/or causes T-R conflict (HO, Head-ON), resulting in cell quiescence.

the category which has a RI score >10 and a ratio of POINT-seq (4 h/0 h) >twofold. Perturbation of these zones are potentially associated with cell cycle arrest.

## Acute depletion of NELF-C protein causes a conflict between transcription and replication

Our results showing the deleterious effects of Pol II readthrough into P-A zones following NELF-C depletion led us to investigate if there is a conflict between dysregulated transcription termination and DNA replication. First, we employed an in situ proximity ligation assay (PLA) which is widely used to measure the T-R conflict (Brown et al, 2022; Hampp et al, 2016; Krishnan et al, 2023; Matos et al, 2020; Petropoulos et al, 2024). The PLA detects fluorescent foci in the cells when proteins of interest are closely located (within 40 nm). Our PLA detect faint signals of MCM2 (required for DNA replication initiation and elongation), PCNA (required for DNA replication elongation), and Pol II S2P (transcription elongating form) antibodies alone in either DMSO or IAA. Notably, the number and intensity of PLA foci are increased by 4 h IAA in NELF-C-AID DLD-1 cells when combinations of Pol II S2P and MCM2 antibodies or Pol II S2P and PCNA antibodies are used in randomly chosen cells (Fig. EV8A for cell images and EV8B for quantifications). We also performed PLA in MCM2 or PCNA-positive cells to reduce background and found more significant increases in PLA foci number and intensity in Pol II S2P + MCM2 and Pol II S2P + PCNA (Fig. 6A for cell images and 6B for quantifications). These results clearly indicate that T-R conflict was induced by 4 h depletion of NELF-C protein.

CDT1 protein is required to fire the DNA replication origins in G1 phase but is degraded in S phase to prevent aberrant DNA replication initiation (Ratnayeke et al, 2023). As NELF-C depletion promotes cell cycle arrest, we investigated the protein level of DNA replication initiation factor, CDT1 on chromatin in the absence of NELF-C protein. The chromatin fractions of NELF-C-AID DLD-1 cells were isolated in a mild condition (Narain et al, 2021) not using either Urea or Empigen detergent. Western blot analysis shows CDT1 protein level is reduced on chromatin by 4 h and 24 h NELF-C loss (Fig. EV8C). To increase the CTD1 protein levels, we used a small compound MLN-4924 that stabilizes CDT1 protein by inhibiting NEDD-8 activating enzyme (Lin et al, 2010) for 24 h after a 24 h IAA pre-treatment in NELF-C-AID DLD-1 cells. Although MLN-4924 did not affect cell cycle in DMSO condition, the reduced cell population in S phase

following NELF-C loss was largely recovered by MLN-4924 treatment (Fig. EV8D), indicating that cell cycle arrest is promoted, at least partially, by a reduction of chromatin-bound CDT1.

Lastly, we investigate whether NELF-C loss affects the rate of DNA synthesis. To accomplish this, we measured the incorporation of the thymine analog, 5'-Bromo-2'-Deoxyuridine (BrdU) at the onset of S phase after release from cell cycle synchronization from M phase (Fig. 6C). BrdU dot blot assay revealed that 4 h depletion of NELF-C does not result in a significant change in the overall incorporation of BrdU, which represents global DNA synthesis (Fig. EV8E). To analyze BrdU genome-wide incorporation, we sequenced IP products of BrdU-labeled DNA fragments on an Illumina platform. Our sequencing results of BrdU-IP are reproducible and predominantly detected genomic regions replicating in the same part of cell phase which is mainly early S phase (Fig. EV8F), since cells were synchronized in M phase by nocodazole and then released (Fig. 6C). The signal of BrdU incorporation at late-replicating region is very flat and appears identical between NELF-C-AID and parental DLD-1 cells. Also, high BrdU signals were accumulated in transcriptionally active gene-rich regions. This observation is supported by Pu-seq data, which indicate that DNA replication initiation zones are located at adjacent intergenic regions of highly transcribed pc genes (Koyanagi et al, 2022). Next, to investigate the effect of NELF-C loss on DNA replication, we subtracted BrdU-IP signals of NELF-C-AID over Parental DLD-1 cells at 4 h IAA (two biological replicates, referred to as ΔR1 and ΔR2), and found that NELF-C loss reduced BrdU signals in the regions where BrdU was highly incorporated in DLD-1 cells (referred to as BrdU minus zones, shown in green) (Fig. EV8F). We also found that NELF-C loss increased BrdU signals mainly at flat signal sites which is late-replicating region (referred to as BrdU plus zones, shown in magenta). Notably, NELF-C loss reduced BrdU-IP signals in where, at least, transcription termination defect was observed (Figs. 6D and EV8F). We validated DNA synthesis, using the BrdU-IP-qPCR assay, at regions such as DoG of FOCAD gene, where transcription termination defect is induced by NELF-C loss (Fig. EV8G). The qPCR signals were normalized with ELAVL2 gene which is in late-replicating zones. Our BrdU-IP-qPCR assay confirmed that 4 h depletion of NELF-C leads to a reduced rate of DNA synthesis in regions with termination defect (Fig. EV8G). This is consistent with BrdU-IP sequencing result in DoG region of FOCAD gene (Fig. EV8F). Finally, to investigate whether transcription termination defect is

genome-widely associated with suppression of DNA replication in NELF-C loss, we subtracted POINT-seq signals of DMSO (IAA 0 h)-treated NELF-C-AID DLD-1 cells from that in IAA 4 h at 2.5 kb DoG regions (from TES + 2.5 kb to TES + 5 kb) and quantified the subtracted POINT signals in both BrdU minus and plus zones (Fig. 6E, top). Our statistical analysis shows that extended or increased transcription readthrough is significantly associated with BrdU minus zones, rather than BrdU plus zones, in NELF-C-depleted cells (Fig. 6E, bottom), suggesting that transcription termination defect contributes to reduced DNA synthesis in gene-rich regions and broadly impacts the replication landscape.

In addition, we investigated whether NELF depletion impairs expression of replication-dependent histone (RDH) genes, which are critical for a productive DNA synthesis in S phase. Nascent RNAs of RDH genes are co-transcriptionally cleaved by CPSF73 protein together with stem-loop binding protein (SLBP), FLASH, and U7 snRNP that binds to the histone downstream elements (Marzluff and Koreski, 2017; Sabath et al, 2013), resulting in production of non-polyadenylated histone mRNAs. A previous study implicated that shRNA knockdown of NELF perturbs 3'end RNA processing of RDH gene and increases pA+ RDH RNAs (Narita et al, 2007).

Consistently, our heatmap and quantification analysis of our nuclear pA+RNA-seq data (two biological replicates) demonstrated that the expression and the polyadenylation of RDH genes are dramatically upregulated by NELF-C depletion (Fig. EV9A,B). For example, pA+ RNA levels of H2BC18 and H2BC21 genes are increased up to ~25-fold at 24 h IAA (Fig. EV9C). In addition, our POINT-seq analysis detected Pol II transcription termination defects on H2BC21 (Fig. EV9D), as well as the other RDH genes in the cluster (Fig. EV9E), following NELF-C depletion. Metagene of POINT-seq displayed a clear termination defect of RDH genes at 4 h IAA treatment (Fig. EV9F). In contrast, stable loss of NELF-C reduces over time the transcription levels of RDH genes (Fig. EV9G) similar to canonical protein-coding genes. Next, we check the expression level of RDH H3.1/H3.2 protein after NELF-C depletion (Fig. EV9H). We detected less H3.1/H3.2 signal in 24 h IAA. This reduction may be caused by the reduction of the S phase in NELF-depleted cells. In addition, this result suggests that pA+RDH RNAs are not efficiently translated. Importantly, H3.1/H3.2 protein level is unaffected in 4 h IAA, suggesting that RDH expression does not control the cell cycle after acute loss of NELF-C. However, we do not exclude the possibility that the reduction of RDH protein level may affect cell cycle progression at a later stage.

Overall, our study proposes the model that perturbation of Pol II transcription termination caused by NELF depletion impairs DNA replication licensing and/or DNA replication elongation (i.e., Head-On collision), leading to cell cycle arrest in G1 or early S phase (Fig. 6F).

## Discussion

Gene mutations, deletions, and amplifications disrupt transcriptional programs and transform normal cells into cancer cells (Bradner et al, 2017). Transcriptional dysregulation is established by altered expression levels of particular transcription and RNA processing factors. Therefore, cancer cells can be highly dependent on such transcription-associated factors. We found that transcripts of CPA factors are highly expressed in COAD (Fig. EV1A). This observation suggests that CPA-dependent Pol II transcription termination is more efficient in tumors than in normal tissue. Indeed, JTE-607, an inhibitor of CPSF73 leads several cancer cells to apoptosis (Cui et al, 2023; Kakegawa et al, 2019; Tajima et al, 2010). We additionally found that RNA expression of NELFCD is upregulated especially in colorectal tumors (Figs. 1B and EV1A). NELF-C expression is required for cell proliferation (Fig. 1F), in agreement with a previous study showing that siRNA knockdown of NELF-CD protein suppresses tumor growth in mice (Song et al, 2018). Therefore, NELF proteins could potentially be potent targets of anticancer drugs. Indeed, knockout of NELF-B gene impairs cell proliferation and differentiation in mouse embryonic stem cells (Williams et al, 2015). Transposon insertion into the NELF-A gene in fly embryos leads to a developmental failure (Wang et al, 2010). Taken together, these observations support the view that the NELF complex has an evolutionary conserved role in cell proliferation.

NELFs were originally identified as negative transcription factors together with DSIF in vitro (Yamaguchi et al, 1999). Our present study demonstrates that acute loss of NELF-C protein perturbed Pol II transcription termination in colorectal cancer cells, while depletion of the DSIF component SPT4 had no effect. In this study, although we could not fully examine the DSIF complex's role in transcription termination since loss of the other DSIF component SPT5 destabilizes the Pol II complex (Aoi et al, 2025; Aoi et al, 2021), we found that NELF-C promotes Pol II transcription termination independently of SPT4.

Many previous studies employed nascent RNA analysis combined with CDK9 inhibitor (DRB block/release or Flavopiridol (FP) block) and revealed a consistent rate of elongation of ~2 kb/min, regardless of the methods. For instance, DRB block/release such as DRB/TTchem-seq (Gregersen et al, 2020), DRB/GRO-seq (Saponaro et al, 2014), and 4sUDRB-seq (Fuchs et al, 2014) showed 2.07 (median), 3.1 (median), 3.5 (average) kb/min, whereas FP block such as GRO-seq with FP (Jonkers et al, 2014) showed 2.4 (average) kb/min. It appears that the rates slightly vary depending on the experimental systems and computational approaches, rather than the use of DRB block/release or FP block. Notably, the NVP-2 (instead of FP) block method which was employed in this study has been used to study the rate of elongation (Aoi et al, 2021). The DRB block/release method is suitable for studying the role of elongation factors during early elongation, such as the role of SPT5 phosphorylation by CDK9, but does not directly address any mechanisms during late elongation. Previous studies support the idea that establishment of elongation-potent Pol II by CDK9 takes place during the early checkpoint, but not during late elongation (Booth et al, 2018; Jonkers et al, 2014). In contrast, the NVP-2 (or FP) block method is a direct approach to address the mechanisms underlying late elongation, given that it detects a defect in the elongation of Pol II that has already passed the early checkpoint. However, SPT5 is also required for the rate of elongation beyond the early checkpoint, implying that CDK9-independent function of SPT5 may contribute to stimulate the rate of elongation throughout gene bodies (e.g., via a DNA-RNA clamp) (Bernecky et al, 2017). Also, our NVP-2 block analysis detects a limited effect of increased Pol II elongation activity after NELF-C depletion. Therefore, DRB block/release analysis will contribute to understanding the more detailed function of Pol II elongation rate.

POINT-seq signals in the gene body were reduced by NELF-C depletion, which is consistent with PRO-seq data (Aoi et al, 2020). A decrease in both PRO-seq and POINT-seq signals across gene body does not clarify if there is a reduced number of transcribing polymerases and/or a faster Pol II, as both phenotypes will reduce Pol II signal. Notably, our result of Pol II EI experiments (TT-seq/ POINT-seq and TT-seq/PRO-seq) demonstrated that NELF loss accelerates Pol II ER over most pc genes (Figs. 4B and EV4I), supported by TT-seq+NVP-2 block experiment across a limited number of genes (Fig. EV4D,E). Consistent with previous finding of heat shock gene transcription in the absence of NELF (Aoi et al, 2020), our results suggest that Pol II EI may be connected to Pol II pausing at promoter proximal site. However, it remains unclear how NELF depletion links to an increase of Pol II EI. Loss of NELF proteins shifts Pol II pausing from a promoter-proximal site to its downstream site called 2nd pause (Aoi et al, 2020). As NELF-C ChIP signal is predominantly detected at TSS, this suggests that NELF-C could be required to ensure the formation of a properly regulated Pol II elongation complex across a gene. The Jensen group recently proposed that Pol II pausing at the promoter pausing zone (up to 3 kb downstream of TSS) may contribute to RNA quality control (Garland and Jensen, 2024). A possibility is therefore that the loss of NELF-C promotes transcription of transcripts that should not have passed quality control, resulting in poor processing and transcriptional readthrough, potentially due to a higher Pol II ER. Faster Pol II elongation may be established at the promoter pausing zone in the absence of the NELF complex, and so may interfere with the recognition of the PAS or may fail to be caught by a torpedo activity of Xrn2 on a subset of genes. As capping and Integrator recruitment are not affected by NELF-C depletion, the quality control pathways regulated by ARS2 could be affected by NELF-C loss (Aoi et al, 2020).

It remains unclear how exactly NELF-C loss influences Pol II transcription termination of pc genes. Previous biochemical approaches have detected an interaction between the Integrator complex and the NELF complex in HeLa S3 cells (Stadelmayer et al, 2014), suggesting that this interaction has a role in transcription termination since loss of the Integrator complex attenuates pc gene transcription (Dasilva et al, 2021; Wagner et al, 2023a). However, it is unlikely that the Integrator complex is involved in NELF-mediated transcription termination, since our ChIP-seq experiment shows an increased enrichment of INTS3 protein at 3' end of pc genes (Fig. EV4K) and PRO-seq data (Dasilva et al, 2021) following a knockdown of INTS11 does not recapitulate NELF-C transcriptional readthrough (Fig. EV4L).

From our results, we know that NELF-C depletion increases POINT-seq and TT-seq signals downstream of the PAS of most genes. In support of a role of NELF complex in transcription termination, re-analysis of HAP1 NELF-B-dTAG spike-in PRO-seq from Price's group (Santana et al, 2024) also shows a widespread transcriptional readthrough downstream of pc genes after 2 h of VHL treatment (Fig. EV3G). These results support our model that loss of NELF complex, at least via degradation of NELF-B and NELF-C, promotes a nascent transcriptional termination defect and is not due to RNA remaining associated with chromatin. As NELF-C degradation promotes a redistribution of CTSF64 and XRN2 towards the promoter region to the detriment of the PAS (Fig. 4D), we propose that the loss of CPA factors at the 3'end of pc genes, in combination with the higher transcriptional elongation rate,

promotes the transcription termination defect. A second interesting observation is the slowdown of Pol II and lack of CSTF64 recruitment across LE of pc genes in the absence of NELF-C (Fig. 4C,D). In line with this, Murphy and Fisher groups reported that CDK9 inhibition disrupts the elongation complex on LE (Laitem et al, 2015; Tellier et al, 2022) and dephosphorylates XRN2 (Sanso et al, 2016), which impairs transcription termination, respectively. Taken together, these findings led us to a hypothesis that NELF-C may be closely associated with CDK9 activity to control Pol II EA, CSTF64, and XRN2 recruitment, and also XRN2 phosphorylation near PAS. However, there are several questions to be explored. How Pol II slows down before PAS in the absence of NELF-C protein? How the slower Pol II elongation before PAS is linked to a failure of CPA factor recruitment resulting in termination defect? Thus, further investigation is needed for better understanding of the mechanism.

T-R conflict is a major source of DNA damage. Previous studies have demonstrated that collisions can occur between the running DNA replisome and Pol II (Head-On) or Pol II-derived RNA:DNA hybrid (Co-directional) (Garcia-Muse and Aguilera, 2016; Goehring et al, 2023). Indeed, BRCA2/RNaseH2 and BRAC1/SETX suppress T-R conflict by resolving R-loop structure at TES (Goehring et al, 2024; Hatchi et al, 2015). Additionally, expression of oncogenic proteins HRAS$^{V12}$ induces global upregulation of transcription and R-loop formation, leading to replication stress (Kotsantis et al, 2016). Overexpression of other oncogenic proteins such as CCNE1 and MYC activate replication origin within highly transcribed pc genes, resulting in T-R conflict (Macheret and Halazonetis, 2018). An siRNA screen demonstrated that loss of WDR33, a CPA factor, causes global perturbation of Pol II transcription termination and a higher cell sensitivity to DNA replication stress (Teloni et al, 2019). This finding suggests that Pol II transcription activity needs to be restricted to certain regions by transcription termination, although the possibility that the replication stress was indirectly generated by long period (72 h) siRNA depletion of WDR33 cannot be ruled out. Furthermore, loss of CPSF73 and Xrn2 proteins severely disrupts transcription termination of pc genes (Davidson et al, 2019; Eaton et al, 2018; Sousa-Luís et al, 2021). Therefore, inhibition or depletion of these proteins induces catastrophic DNA damage and cell death (Cui et al, 2023; Morales et al, 2016). In addition, our previous study demonstrated that siRNA-depletion of SPT6 protein globally upregulates lncRNA transcription and perturbs its termination (Nojima et al, 2018). This leads to T-R conflict followed by cell cycle arrest in G0 phase and then cellular senescence. Our re-analysis of published TT-seq data in PNUTS-dTAG DLD-1 cells (Wang et al, 2024) shows an increase of TT-seq signal in replication initiation zones after PNUTS depletion (Fig. EV7D), suggesting that rapid loss of PNUTS could also affect cell cycle and DNA replication via transcriptional readthrough, although it is unclear how much they are affected. In this study, our PLA strongly suggests that acute NELF-C loss (4 h IAA) induces T-R conflict. We do not exclude the possibility that the formation of Pol II S2P and MCM2 or PCNA complexes are increased by NELF-C loss, since it was reported that Pol II interacts with MCMs (Yankulov et al, 1999) and PCNA (Fenstermaker et al, 2023) in vitro. Even so, we propose a novel source of T-R conflict (Head-On) that occurs between Pol II and DNA replication origin firing in G1 or early S phase. Concomitant to increased T-R conflicts following NELF-C loss (4 h IAA), cell

cycle inhibitor proteins such as P21 and P57 are significantly upregulated at later time point (6 h IAA), and cell cycle arrest is detected at 12 h IAA (Fig. 2C,D). This suggests that DNA replication in early S phase is preferentially affected by transcriptional dysregulation following NELF loss, although we cannot rule out that mRNA stability and export for cell cycle inhibitor genes are increased after NELF-C loss.

Pol II transcription termination was globally impaired by rapid depletion of the NELF complex, but was limited over time to a subset of genes (~3300 pc genes affected after 24 h IAA). This limited defect is still sufficient in sustaining cell cycle arrest without inducing cell death, indicating that cells can still enter quiescence following NELF loss. In fact, normal tissues that contain non-replicative somatic cells express NELF-C protein at a lower level (Song et al, 2018). Our study now provides insight into T-R conflict in replicative cells that contributes to establish cell quiescence. Therefore, we conclude that inhibition or degradation of NELF-C protein can be an attractive approach to arrest, at least, colorectal cancers.

# Methods

### Reagents and tools table

| Reagent/resource | Reference or source | Identifier or catalog number |
|---|---|---|
| **Experimental models** | | |
| Parental DLD-1 OsTIR1 cell (human) | Previous study https://doi.org/10.1016/j.molcel.2020.02.014. | Available from Shilatifard Lab by request. |
| NELF-C-AID DLD-1 OsTIR1 cell #7-10B (human) | Previous study https://doi.org/10.1016/j.molcel.2020.02.014. | Available from Shilatifard Lab by request. |
| SPT4-AID DLD-1 OsTIR1 cell #5-1D (human) | Previous study https://doi.org/10.1016/j.molcel.2021.08.006. | Available from Shilatifard Lab by request. |
| **Recombinant DNA** | | |
| N/A | | |
| **Antibodies** | | |
| Mouse monoclonal anti-Pol II CTD, Total | Previous study https://doi.org/10.1016/j.cell.2015.03.027. | MABI601, Available from Kimura Lab by request. |
| Rabbit polyclonal anti CSTF64 | Bethyl | Cat# A301-092A |
| Rabbit polyclonal anti-INTS3 | Proteintech | Cat# 16620-1-AP |
| Spike-in Antibody | Active Motif | Cat# 61686 |
| Mouse monoclonal anti-MCM2 | Cell Signaling | Cat# 12265 |
| Mouse monoclonal anti-PCNA | Santa Cruz | Cat# sc-56 |
| Rabbit polyclonal anti RNA Polymerase II/POLR2A [p Ser2] | Novus | NB100-1805 |
| Mouse monoclonal anti-POLR2A (N-terminal) | Santa Cruz | Cat# sc-55492 |
| Mouse monoclonal anti-RPA194 (Pol I) | Santa Cruz | Cat# sc-48385 |

| Reagent/resource | Reference or source | Identifier or catalog number |
|---|---|---|
| Rabbit monoclonal anti-NELFB (COBRA1) | Abcam | Cat# ab167401 |
| Mouse monoclonal anti-NELFA | Santa Cruz | Cat# sc-365004 |
| Rabbit monoclonal anti-NELFE | Abcam | Cat# ab170104 |
| Rabbit monoclonal anti NELFC/D (TH1L) | Cell Signaling | Cat# 12265 |
| Mouse monoclonal anti CDT1 | Santa Cruz | Cat# sc-365305 |
| Rabbit monoclonal anti-SPT4 | Cell Signaling | Cat# 64828 |
| Mouse monoclonal anti-tubulin (clone DM1A) | Sigma | Cat# T6199 |
| Rabbit polyclonal anti P57 (Kip2) | Cell Signaling | Cat# 2557 |
| Rabbit monoclonal anti p21 (Cip1) | Cell Signaling | Cat# 2947 |
| Mouse monoclonal anti-H3 | Active Motif | Cat# 39763 |
| Mouse monoclonal anti-histone H2A.X | Santa Cruz | Cat# sc-517336 |
| Rabbit monoclonal anti-histone H2A.X Ser139p | Cell Signaling | Cat# 2577S |
| Goat anti-Mouse IgG H&L (HRP) | Abcam | Cat# ab205719 |
| Goat anti-Rabbit IgG H&L (HRP) | Abcam | Cat# ab205718 |
| Goat anti-Rabbit IgG (H + L) Highly Cross-Adsorbed Secondary Antibody, Alexa Fluor 488 | Invitrogen | Cat# A-11034 |
| ProLong Glass Antifade Mountant | Invitrogen | Cat# P36982 |
| CoverGrip Coverslip Sealant | Biotium | Cat# 23005 |
| Mouse anti-BrdU antibody Clone B44 | BD Bioscience | Cat# 347580 |
| **Oligonucleotides and other sequence-based reagents** | | |
| PCR primers | This study | Table EV2 |
| **Chemicals, enzymes, and other reagents** | | |
| Dulbecco's Modified Eagle Medium (DMEM -High glucose) | Nacalai Tesque | Cat# 29113-53 |
| Fetal Bovine Serum (FBS) | Gibco | Cat# 10270-106 |
| Penicillin–Streptomycin Mixed Solution | Nacalai Tesque | Cat# 26253-84 |
| Auxin (IAA) | Sigma | Cat# I2886 |
| Dimethyl sulfoxide (DMSO) | Sigma | Cat# D8418 |
| Pladienolide B (PlaB) | Santa Cruz | Cat# SC-391691 |
| MLN-4924 | Cell Signaling | Cat# 85923 |
| Benzonase | Millipore | Cat# E1014 |

| Reagent/resource | Reference or source | Identifier or catalog number |
|---|---|---|
| NuPAGE LDS Sample Buffer (4x) | Thermo Fisher Scientific | Cat# NP0007 |
| Chemi-Lumi One Super | Nacalai Tesque | Cat# 02230-30 |
| 4-15% Mini-PROTEAN Precast Protein Gels, 12 well | BioRad | Cat# 4561085 |
| Empigen ~30% | Sigma | Cat# 30326 |
| Turbo DNase | Thermo Fisher Scientific | Cat# AM2239 |
| RiboLock RNase inhibitor | Thermo Fisher Scientific | Cat# EO0381 |
| Tri Reagent | Cosmo Bio | Cat# TR118 |
| Dynabeads M280 Sheep Anti-mouse IgG | Thermo Fisher Scientific | Cat# 11202D |
| SPRISelect reagent | Beckman Coulter | Cat# B23317 |
| Spike-in SIRV-Set2 | Lexogen | Cat# 050.0 |
| Spike-in Chromatin | Active Motif | Cat# 53083 (Note: this is for ChIP-seq) |
| 4-Thiouridine | Sigma | Cat# T4509 |
| NVP-2 | Med Chem Express | Cat# HY-12214A |
| MTSEA-biotin-XX | Biotium | Cat# 90066 |
| Dynabeads MyOne Streptavidin C1 | Thermo Fisher Scientific | Cat# 65001 |
| Novex 6% TBE gel, 12 well | Invitrogen | Cat# EC62652BOX |
| SuperScript™ IV First-Strand Synthesis System | Invitrogen | Cat# 18091200 |
| Formaldehyde | Sigma | Cat# F8775 |
| Glycine | Nacalai Tesque | Cat# 17109-35 |
| 4%-Paraformaldehyde Phosphate Buffer Solution | Nacalai Tesque | Cat# 09154-85 |
| Duolink® In Situ Detection Reagents Red | Sigma | Cat# DUO92008 |
| Goat anti-Mouse IgG (H + L) Cross-Adsorbed Secondary Antibody, Alexa Fluor™ 488 | Thermo | Cat# A-11001 |
| Duolink® In Situ PLA® Probe Anti-Rabbit PLUS | Sigma | Cat# DUO92002 |
| Duolink® In Situ PLA® Probe Anti-Mouse MINUS | Sigma | Cat# DUO92004 |
| Duolink® In Situ washing buffer, fluorescence | Sigma | Cat# DUO82049-4 |
| FlourSave Reagent | Merck Millipore | Cat# 345789-20 ML |
| Hoechst 33342 | Nacalai Tesque | Cat# 19172-51 |
| Nocodazole | FUJIFILM | Cat# 140-08531 |
| 5-Bromo-2'-deoxyuridine | Sigma-Aldrich | Cat# B5002 |
| Dynabeads Protein G | Invitrogen | Cat# 10003D |
| Agencourt AMPure XP Kit | BECKMAN COULTER | Cat# A63880 |
| Dynabeads Protein A | Thermo Fisher Scientific | Cat# 10002D |
| RNaseA | Thermo Fisher Scientific | Cat# EN0531 |

| Reagent/resource | Reference or source | Identifier or catalog number |
|---|---|---|
| KAPA SYBR Fast qPCR kit (RocheLightCyclerqPCR) | Roche | Cat# KK4611 |
| NEBNext Ultra II Directional RNA library prep kit for Illumina | NEB | Cat# E7760 |
| Direct-zol RNA microprep | Zymo Research | Cat# R2061 |
| NEBNext PolyA mRNA Magnet isolation module | NEB | Cat# E7490 |
| ATAC-Seq Kit | Active Motif | Cat# 53150 |
| Click-iT Plus EdU Alexa Fluor488 Flow Cytometry Assay Kit | Thermo Fisher Scientific | Cat# C10633 |
| NEBNext Magnesium RNA Fragmentation Module | NEB | Cat# E6150 |
| NEBNext rRNA Depletion Kit v2 | NEB | Cat# E7400 |
| NEBNext Ultra II DNA Library Prep Kit for Illumina | NEB | Cat# E7645 |
| Blood & Cell Culture DNA Midi Kit with 100/G Genomic-tips | QIAGEN | Cat# 13343 |
| Monarch® Genomic DNA Purification Kit | NEB | Cat# T3010S |
| DNA Clean & Concentrator Kit | Zymo Research | Cat# D4013 |
| Monarch® Genomic DNA Purification Kit | NEB | Cat# T3010S |
| DNA Clean & Concentrator Kit | Zymo Research | Cat# D4013 |
| **Software** | | |
| FastQC (v0.11.5) | https://www.bioinformatics.babraham.ac.uk/projects/fastqc/ | N/A |
| Cutadapt (v1.18) | https://cutadapt.readthedocs.io/en/stable/ | Martin, 2011 |
| TrimGalore (v0.4.4) | https://www.bioinformatics.babraham.ac.uk/projects/trim_galore/ | N/A |
| STAR (v2.7.0) | https://github.com/alexdobin/STAR | Dobin et al, 2013 |
| SAMtools (v1.9) | http://www.htslib.org/ | Li et al, 2009 |
| BEDtools (v2.29.2) | https://bedtools.readthedocs.io/en/latest/ | Quinlan and Hall, 2010 |
| deepTools2 (v3.4.2) | https://deeptools.readthedocs.io/en/develop/ | Ramirez et al, 2016 |
| bedGraphToBigWig (v2.10) | https://hgdownload.soe.ucsc.edu/admin/exe/linux.x86_64/ | N/A |

| Reagent/resource | Reference or source | Identifier or catalog number |
|---|---|---|
| Salmon (v1.2.1) | https://salmon.readthedocs.io/en/latest/index.html | Patro et al, 2017 |
| DESeq2 (v1.30.1) | https://bioconductor.org/packages/release/bioc/html/DESeq2.html | Love et al, 2014 |
| Apgelm (v1.18.0) | https://bioconductor.org/packages/release/bioc/html/apeglm.html | Zhu et al, 2019 |
| dplyr (v1.1.2) | https://cran.r-project.org/web/packages/dplyr/index.html | N/A |
| ggplot2 (v3.4.3) | https://cran.r-project.org/web/packages/ggplot2/index.html | N/A |
| tidyr (v1.3.0) | https://cran.r-project.org/web/packages/tidyr/index.html | N/A |
| stringr (v1.5.0) | https://cran.r-project.org/web/packages/stringr/index.html | N/A |
| Enhanced Volcano (v1.14.0) | https://github.com/kevinblighe/EnhancedVolcano. | N/A |
| Harmonizome3.0 | https://maayanlab.cloud/Harmonizome/gene_set/RNA+Polymerase+II+Transcription/Reactome+Pathways | Rouillard et al, 2016 |
| proRate | https://github.com/yuabrahamliu/proRate | N/A |
| Cellpose (v2.0) | https://www.cellpose.org/ | Stringer et al, 2021 |
| **Other** | | |
| Countess 3 FL Automated Cell Counter | Thermo Fisher Scientific | Cat# AMQAF2000 |
| Qubit 4 Fluorometer | Thermo Fisher Scientific | Cat# Q33238 |
| Mini-PROTEAN, Tetra Cell, Electrophoresis | BioRad | Cat# 1658001JA |
| Mini Trans-Blot Cell | BioRad | Cat# 1703930JA |
| XCell SureLock Mini-Cell | Thermo Fisher Scientific | Cat# EI0001 |
| ChemiDoc Touch Imaging System | BioRad | Cat# 17001401JA |
| Thermo Mixer C | Eppendorf | Cat# 5382000023 |
| NovaSeq6000 | Illumina | Novogene |
| **Deposited Data** | | |
| Raw sequencing data (POINT-seq, ATAC-seq, TT-seq, ChIP-seq and nuclear pA+RNA-seq) | This study | GEO: GSE253121 (https://www.ncbi.nlm.nih.gov/geo/query/acc.cgi?acc=GSE253121) |
| Re-analyzed ChIP-seq data | Previous study (https://doi.org/10.1016/j.molcel.2020.02.014) | GEO: GSE144786 |
| Re-analyzed PRO-seq data | Previous study (https://doi.org/10.1016/j.molcel.2020.02.014) | GEO: GSE144786 |
| Re-analyzed POINT-seq data (CPSF73, Xrn2) | Previous study (https://doi.org/10.1016/j.molcel.2021.02.034) | GEO: GSE159326 |
| Processed TCGA and GTEx count data | Previous study (https://doi.org/10.1038/sdata.2018.61) | Figshare https://doi.org/10.6084/m9.figshare.5330539 |
| Raw sequencing data (BrdU-IP seq) | This study | GEO: GSE294597 (https://www.ncbi.nlm.nih.gov/geo/query/acc.cgi?acc=GSE294597) |

## Cell culture

Details of DLD-1 cell lines (See Reagents and tools table) were previously published (Aoi et al, 2020; Aoi et al, 2021). Cells were cultured in high-glucose Dulbecco's Modified Eagle's Medium (DMEM) with 10% fetal bovine serum (FBS) and penicillin/streptomycin. All the cells were maintained in a humidified incubator at 5% $CO_2$, 37 °C.

## IAA treatment and cell count

AID-tagged protein degradation was induced by adding 500 μM auxin (IAA) to the culture media. In the recovery experiment, cells were washed twice with pre-warmed PBS before adding fresh media to ensure the complete removal of IAA. Cell counting was completed using the Countess 3 FL system (Thermo Fisher Scientific).

## Western blot

The primary and secondary antibodies used in this study are listed in the Reagents and Tools Table. Chemi-Lumi One Super was used for the chemiluminescence. Images were captured by the Chemidoc Touch V3 system (BioRad). The blot signals were quantified by Image Lab software (BioRad).

## Polymerase intact nascent transcript sequencing (POINT-seq)

POINT-seq was performed as previously described (Sousa-Luís et al, 2021) with the following modifications. Typically, 10 million cells were lysed with 4 mL ice-cold hypotonic buffer (10 mM Tris-Cl pH 7.5, 10 mM NaCl, 2.5 mM $MgCl_2$, 0.5% NP-40) on ice for 5 min. The cell lysate was carefully underlaid with 1 mL sucrose cushion buffer (hypotonic buffer+10% (w/v) sucrose). The nuclear fraction was pelleted by centrifugation (300× g, 5 min at 4 °C), then washed with ice-cold hypotonic buffer (without NP-40) once. The purified nuclear fraction was first resuspended in 300 μL (200–500 μL, depending on the pellet size) ice-chilled NUN1 buffer

(20 mM Tris-HCl pH 7.9, 75 mM NaCl, 0.5 mM EDTA pH 8.0, and 50% glycerol) and then lysed in 3 mL (at least tenfold volume of NUN1) ice-cold NUN2 buffer (20 mM HEPES-KOH pH 7.6, 300 mM NaCl, 0.2 mM EDTA pH 8.0, 7.5 mM MgCl$_2$, 1% NP-40, 1 M Urea, 3% Empigen BB, 1×Protease inhibitor Complete EDTA-free, and 1×PhosSTOP). Note: The cotton candy-like chromatin should appear within a few min. The chromatin was washed with ice-cold PBS once. Chromatin-bound Pol II was digested by DNase mixture (10 mM Tris-HCl pH 7.5, 400 mM NaCl, 10 mM MnCl$_2$, 2 Units/µL RiboLock, and 0.2 Units/µL Turbo DNase) at 37 °C for 15 min at 1200 rpm in the ThemoMixer C (Eppendorf). EDTA pH 8.0 (final 1 mM) was used to stop the DNase reaction, and the supernatant was collected by high-speed centrifugation (14,000× g, 10 min at 4 °C). A five-fold volume of ice-cold NET-2E buffer (50 mM Tris-HCl pH 7.4, 150 mM NaCl, 0.05% NP-40, and 3% Empigen BB) was used for the dilution of the supernatant. The 10 µg of anti-Pol II antibody MABI601 were conjugated with 200 µL anti-mouse IgG Dynabeads solution in 1 mL NET-2 buffer and incubated at 4 °C for 1 h. The conjugated beads were washed with NET-2 buffer once and added to the diluted supernatant and then incubated at 4 °C for 1 h (Note: Avoid overnight). After the incubation, beads were washed with 1 mL of ice-cold NET-2E buffer at least six times. Pol II-bound intact nascent transcripts (POINTs) were treated with Turbo DNase once and then purified using Direct-zol kit following the manufacturer's protocol. An equal amount of SIRV spike-in oligos were added to each POINTs before the library preps. The POINTs were fragmented to 150-300 nucleotides for sequencing according to the protocol of NEBNext Ultra II Directional RNA library prep kit. The POINT-seq libraries were applied to the Illumina PE150 NovaSeq6000 service (Novogene).

### Transient transcriptome sequencing (TT-seq)

TT-seq samples were prepared as previously described (Schwalb et al, 2016) with the following modifications. NELF-C-AID DLD-1 cells were treated with DMSO or 500 µM IAA for 4 h. Cells were then treated with CDK9 inhibitor NVP-2 at 250 nM for 0, 0.5 or 1 h, followed by in vivo RNA labeling with 500 µM 4-Thiouridine for 10 min. Total RNA was extracted using TRIzol. In total, ~200 µg total RNA was fragmented at 94 °C for 2 min using NEBNext Magnesium RNA Fragmentation Module. Biotinylation of the fragmented 4sU-labeled RNA was performed using 33 µg/ml MTSEA-biotin-XX in the presence of 20% DMF for 1.5 h at room temperature. The biotinylated RNA was purified using Dynabeads MyOne Streptavidin C1 and eluted in 100 mM DTT. Library preparation with ~100 ng of the enriched RNA was performed using NEBNext rRNA Depletion Kit v2 and NEBNext Ultra II Directional RNA Library Prep Kit. DNA libraries were sequenced on NovaSeq 6000 (Illumina).

### Nuclear-pA + RNA-seq

The nuclear fraction of the DLD-1 cell was prepared using POINT protocol. The total RNAs were extracted using Tri Reagent from the nuclear fraction and subsequently purified by following the manufacturer's protocol of Direct-zol kit. NEBNext PolyA mRNA Magnet isolation module was employed to enrich the nuclear pA+ RNAs from 5 µg of the nuclear RNAs. The 100 ng of nuclear pA+ RNAs were fragmented to 150-300 nucleotides for sequencing

according to the protocol of NEBNext Ultra II Directional RNA library prep kit. The nuclear-pA+ RNA-seq libraries were applied to the Illumina PE150 NovaSeq6000 service (Novogene).

### ATAC-seq

ATAC-seq samples were prepared from 1 million NELF-C-AID DLD-1 cells by following the manufacturer's protocol of ATAC-seq kit (Active Motif). IAA (500 µM) was incubated for 0 h, 4 h, and 24 h to degrade NELF-C protein. The libraries were applied to the Illumina PE150 NovaSeq6000 service (Novogene).

### Immunofluorescence

Parental and NELF-C-AID DLD-1 cells were seeded on 13-mm sterile round coverslips overnight at 37 °C to achieve 70–80% confluence. After DMSO or IAA treatments, cells on coverslips were washed twice with PBS and fixed using 4% formaldehyde/PBS for 10 min at room temperature. The coverslips were washed with PBS-T (PBS and 0.1% Tween) and permeabilized with 0.2% TritonX-100 in PBS for 10 min at room temperature. Following permeabilization, cells were blocked with 2% BSA/PBS-T for 1.5 h at room temperature. The coverslips were incubated with primary antibody (Rabbit anti-P-Histone H2A.X (S139) (2577S, Cell Signalling Technology) 1:500) in 2% BSA/PBS-T for 2 h at room temperature. The coverslips were then washed three times with PBS-T for 5 min each on a shaker. The coverslips were incubated with secondary antibody (Goat anti-Rabbit IgG (H + L) Highly Cross-Adsorbed Secondary Antibody, Alexa Fluor Plus 488 (A-11034, Invitrogen), 1:500) diluted in 2% BSA/PBS-T for 1 h at room temperature. The coverslips were washed three times with PBS-T for 5 min each on a shaker and counterstained with DAPI (2 µg/ml in PBS-T) for 10 min at room temperature. The coverslips were washed three times with PBS, mounted with a drop of media (ProLong Glass Antifade Mountant (P36982, Invitrogen)), and then sealed to glass slides (CoverGrip Coverslip Sealant (23005, Biotium)). Cells were imaged using a Visitech Infinity3 HAWK confocal laser microscope.

### Proximity ligation assay (PLA)

The PLA experiments were performed as previously published (Matos et al, 2020). In brief, NELF-C-AID DLD-1 cells were cultured on the glass coverslips until 50 ~ 60% confluency and washed once with PBS. The cells were pre-treated with CSK buffer (0.2% TritonX-100, 20 mM HEPES-KOH pH 7.9, 100 mM NaCl, 3 mM MgCl$_2$, 300 mM sucrose, 1 mM EGTA) for 3 min on ice to remove cytoplasm. Following the pre-treatment, the cells were fixed with paraformaldehyde (Nacalai Tesque) for 5 min at room temperature, followed by ice-cold methanol treatment at −20 °C for 20 min. Then, the cells were permeabilized with 1× PBS containing Triton-X100 for 4 min at room temperature. After permeabilization, the cells were blocked with 3% BSA for 1 h at room temperature. Next, primary antibodies were diluted as follows: PCNA (SC-56, Santa Cruz) and MCM2 (12265, Cell Signalling) at 1:500, and Pol II Ser2P (NB100-1805, Novus) at 1:1000. For secondary antibodies, anti-mouse minus and anti-rabbit plus PLA probes (PLA kit, Sigma) were used. Cells were imaged using a Zeiss LSM980 (with Airyscan 2) microscope.

## RT-qPCR

Total RNAs were purified from NELF-C-AID DLD-1 whole cells with Direct-zol kit (ZYMO). According to the manufacturer's protocol, 500 ng of total RNAs were used with the Superscript III kit (Invitrogen) and N6 random primers (Invitrogen) for cDNA synthesis. The cDNAs were amplified with the indicated primers (PCR primer sequences are available in Table EV2.) and KAPA SYBR Fast qPCR kit (Roche) for quantitative real-time PCR (LightCycler 96, Roche). The experiments were repeated three times for three biological replicates. For absolute quantification of qPCR signals, genomic DNAs of NELF-CD-AID DLD-1 cells were serially diluted.

## ChIP-seq

Cells were cultivated on 150 mm dishes until 50% confluency, fixed by the addition of 1% formaldehyde (Sigma) for 15 min at 37 °C, and quenched by 125 mM glycine for 10 min at 37 °C. The cells were collected by scraping, washed 3 times with cold PBS, resuspended in 1.5 mL L1 buffer (50 mM Tris pH 8.0; 2 mM EDTA pH 8.0; 0.1% NP40; 10% glycerol; protease inhibitors (Nacalai Tesque)) per 10 million cells, and lysed on ice for 5 min. The nuclei were collected by centrifugation at $800 \times g$ for 5 min at 4 °C and lysed in 1.5 mL of L2 buffer (0.2% SDS; 10 mM EDTA; 50 mM Tris pH 8.0; protease inhibitors (Nacalai Tesque)). The suspension was sonicated in 15 mL conical tubes in a cooled Bioruptor 2 (Diagenode) for 15 min at low settings. Sonicated chromatin was centrifuged for 10 min at 14,000 rpm and collected the supernatant. The shared chromatin was diluted 10× in the dilution buffer. For calibration, 4 ng of spike-in chromatin (Active Motif) was added to each IP sample. In total, 60 μg of Protein A Dynabeads (Thermo Fisher Scientific) for each sample were pre-washed twice with dilution buffer (0.5% NP40; 200 mM NaCl; 50 mM Tris pH 8.0; protease inhibitors). The chromatin was incubated with pre-washed beads for 1 h at 4 °C to clear unspecific binding proteins to the beads. 3 μg of the pre-cleared chromatin was incubated with 6 μg of α-CSTF64 (A301-092A, Bethyl) or α-INTS3 (16620-1-AP, Proteintech) and 4.4 μg of spike-in specific antibody (Active Motif), O/N at 4 °C on rotator. Next, 60 μL pre-washed beads were added to the chromatin+antibody sample and incubated for 1–2 h at 4 °C on a rotator. After the incubation, the beads were washed two times with buffer A (0.05% SDS; 0.5% TritonX; 2 mM EDTA; 150 mM NaCl; 20 mM Tris pH 8.0), three times with buffer B (0.02% SDS; 0.5% NP40; 2 mM EDTA; 250 mM NaCl; 20 mM Tris pH 8.0), two times with buffer C (0.05% DOC; 0.5% NP40; 2 mM EDTA; 500 mM NaCl; 20 mM Tris pH 8.0) and finally once again with buffer D (1 mM EDTA; 10 mM Tris pH 8.0). The immunoprecipitated DNA fragments were eluted from the beads with 100 μL elution buffer (1% SDS, 100 mM NaHCO3) for 15 min at 30 °C, and repeated this step twice. After elution, for de-crosslinking, the DNA fragments were incubated for 4 h at 65 °C in the presence of 10 μg RNase A (EN0531, Thermo Fisher Scientific) on ThemoMixer C (Eppendorf). The immunoprecipitated DNA was then purified with the DNA Clean & Concentrator Kit (ZYMO) according to the manufacturer's protocol and used for library preparation. NEBNext Ultra II DNA Library Prep Kit for Illumina (NEB) was used to prepare libraries for sequencing. The ChIP-seq libraries were sequenced in NovaSeq6000 PE150 platform (Illumina). Two biological replicates of CstF64 and IntS3 ChIP experiments were performed, each for DMSO 4 h and IAA4h conditions.

## Fluorescence-activated cell sorter (FACS) analysis

To examine the effect of recovery from the degradation of NELF-C protein on cell cycle, at 24 h time point after 24 h IAA (500 μM) treatment in NELF-C-AID DLD-1 cells, IAA was washed off by rinsing cells on culture dishes with media twice. These cells were further incubated with media without IAA for 24 h or 48 h. Alternatively, to examine the effect of MLN-4924, at the same time point, 300 nM of MLN-4924 or DMSO were added to cell culture media and further incubated for 24 h or 48 h. Next, cells were incubated with media containing 10 μM of 5-ethynil-2'-deoxyuridine (EdU) for 1.5 h. After adding 0.1% sodium azide to media, cells were trypsinized and harvested. Using Click-iT Plus EdU Alexa Fluor488 Flow Cytometry Assay Kit, $3 \times 10^6$ cells were subjected to fixation, permeabilization, and the detection of EdU. After these treatments, cells were then incubated with 40 μg/mL of propidium iodide and 0.1 mg/mL RNase A for 30 m at room temperature and analyzed using BD Accuri C6 Plus (BD bioscience). Data were processed using FlowJo Software (v10.10, BD Biosciences).

## BrdU-IP quantitative PCR (qPCR) assay

Cell cultures ($0.75 \times 10^5$ cells/ml) were seeded in 25 ml of medium in 15 cm dishes. After 48 h of incubation, 50 ng/ml nocodazole (Noc) was added to the culture to synchronize the cells in the M phase. Following 24 h of Noc treatment, cells were released from the M phase by washing off the Noc through two media exchanges. After a 12 h release, BrdU was added to the cell cultures and incubated for the indicated period. Cells were then harvested, and genomic DNA was extracted using the Blood & Cell Culture DNA Midi Kit with 100/G Genomic-tips (Qiagen). After measuring DNA concentration, 10 μg of DNA was dissolved in 200 μL of TE buffer and subjected to sonication using ultrasound (30 s on, 30 s off, for 60 cycles) with the Bioruptor II (BMbio). A 15 μL aliquot of the sonicated DNA solution was set aside as the input sample, and the remaining 150 μL was heated at 96 °C for 10 min and snap-cooled in ice-cold water to denature the DNA before immunoprecipitation with an anti-BrdU antibody (Clone B44, BD Biosciences).

To the denatured DNA solution, an equal volume of 2× BrdU-IP buffer (200 mM sodium phosphate, pH 7.3, 280 mM NaCl, 0.1% TritonX-100) was added. Next, 20 μL of BrdU antibody (3.75 μg) pre-bound to Protein G Dynabeads (Life Technologies) was added to the solution and incubated at 37 °C for 1 h. The tube containing the solution was placed on a magnetic stand, and the beads were washed twice with 1×BrdU-IP buffer and once with TE buffer (10 mM Tris-HCl, pH 8.0, 1 mM EDTA). Bead-bound DNA was eluted in 50 μL of elution buffer (50 mM Tris-HCl, pH 8.0, 1% SDS). Both the input DNA and BrdU-IP DNA were purified using AMPure beads (Beckman Coulter) following the manufacturer's instructions and dissolved in 100 μL of 1 mM Tris-HCl (pH 8.0). These DNA solutions were subjected to qPCR analysis using the QuantStudio 1 Real-Time PCR System (Applied Biosystems). To assess the efficiency of local BrdU incorporation, BrdU-IP values were divided by input DNA values to calculate the IP/input ratio

for each primer pair locus. The IP/input values for each locus were normalized to that of the ELAVL2 locus, a late-replication region not expected to replicate at the onset of the S phase (ELAVL2 locus = 1). PCR primer sequences are available in Table EV2.

## BrdU-IP sequencing

The same DNA samples precipitated with anti-BrdU antibody and used for the qPCR assay were also subjected to library preparation for Illumina sequencing, as previously described (Daigaku et al, 2017). A total of 4 ng of immunoprecipitated single-stranded DNA (ssDNA) fragments in 30 µl of 1 mM Tris-HCl (pH 8.0) was mixed with 5 µl of 3 mg/ml custom-synthesized 8 N random primer, heated at 95 °C, and immediately snap-cooled on ice. The samples were then incubated with 2.5 µl (12.5 units) of Klenow fragment (NEB) and 5 µl of 2 mM dNTPs in a total volume of 50 µl to convert single-stranded DNA (ssDNA) into double-stranded DNA (dsDNA). The resulting dsDNA was purified using AMPure XP beads (Beckman Coulter), and its concentration was determined by fluorometric quantification (PicoGreen; Life Technologies). Fragment size distribution was analyzed with an Agilent TapeStation. Subsequently, all purified dsDNA was used for Illumina library preparation with the NEBNext Ultra II DNA Library Prep Kit (NEB). The libraries were sequenced as 150-bp paired-end reads on an Illumina NovaSeq X Plus platform (Macrogen, Tokyo, Japan).

## Chromatin fraction for western blot

The protocol was derived from a previous study (Narain et al, 2021) with minor modifications. In brief, nuclear extraction was obtained by pipetting (20 times) instead of homogenizing, and the subsequent lysis step was extended to 30 min. The chromatin pellet was digested using Benzonase-containing buffer (150 mM HEPES, pH 7.9, 1.5 mM MgCl$_2$, 150 mM potassium acetate, and 100 Units/mL Benzonase) and then denatured in 1× LDS buffer for 10 min at 70 °C.

## Quantification and statistical analysis

### TCGA and GTEx analyses

The processed TCGA and GTEx data were downloaded from a previous study (Wang et al, 2018). In this study, only paired-end samples in normal tissue (N) and tumor (T) types which have more than 95 samples were employed. Final sample number after screening is indicated in Table EV1. For comparison of normalized expression across N and T types, RSEM FPKM data were used in gglot2. For differential expression analysis in DESeq2, RSEM count data were used for only the genes with a fold change (T/N) < −1.5 or >1.5 and an adjusted P value below 0.05. Data were visualized by EnhancedVolcano. RSEM counts of genes associated with RNA polymerase II transcription were analyzed in Harmonizome 3.0.

### Gene annotation

The list of pc genes was extracted from the Gencode V41 annotation, based on the hg38 version of the human genome. To obtain the annotation of the highest transcript isoform expressed for each pc gene, transcript expression was quantified with Salmon on the biological replicates of nuclear pA+ RNA-seq treated with DMSO, resulting in a list of 10,715 expressed pc genes. The list of

3459 non-overlapping and highly expressed pc genes was obtained by keeping only genes that were: separated by at least 2.5 kb upstream and downstream of their TSS and poly(A) site, respectively, longer than 2 kb, and have an average normalized read counts across the gene body and across the termination region in POINT-seq DMSO and IAA 4 h > 50. The list of exons used for the splicing analysis was obtained by extracting the location of exons in Gencode V41 annotation from the most expressed transcript isoform of each of the 10,715 expressed pc genes. The list of last exons was then obtained by keeping the final exon of the most expressed transcript isoform from expressed pc genes that contain at least two exons and are longer than 2 kb.

### POINT-seq analysis

Adapters were trimmed with Cutadapt in paired-end mode with the following options: --minimum-length 10 -q 15,10 -j 16 -A GATCGTCGGACTGTAGAACTCTGAAC -a AGATCGGAA-GAGCACACGTCTGAACTCCAGTCAC. Trimmed reads were mapped to the human GRCh38.p13 reference sequence or the SIRV-Set 2: SIRV isoform Mix E0 sequences (See Lexogen website) with STAR and the parameters: --runThreadN 16 -- readFilesCommand gunzip -c -k --limitBAMsortRAM 20000000000 --outFilterMultimapNmax 1 --outFilterScoreMin 10 --outSAMtype BAM SortedByCoordinate. SAMtools was used to retain the properly paired and mapped reads (-f 3). For non-spiked POINT-seq, FPKM-normalized bigwig files were created with deepTools2 bamCoverage tool with the parameters -bs 10 -p max –filterRNAstrand forward (or reverse) --normalizeUsing RPKM.

For SIRV spiked POINT-seq, SAMtools view with the –s option was used to subsample all the bam files to the bam file containing the lowest number of reads and then to create strand-specific BAM files. The normalization factor was calculated as: (number of SIRV reads)/(number of human reads + number of SIRV reads). Spiked normalized bedGraph files were created with BEDtools genomecov and the options: -bg -split –scale (1/normalization factor). Bigwig files were generated from the BedGraph files with the bedGraphToBigWig tool.

### RNA-seq analysis

Adapters were trimmed with Cutadapt in paired-end mode with the following options: --minimum-length 10 -q 15,10 -j 16 -A GATCGTCGGACTGTAGAACTCTGAAC -a AGATCGGAA-GAGCACACGTCTGAACTCCAGTCAC. The remaining rRNA reads were removed by mapping the reads to the rRNA genes defined in the human ribosomal DNA complete repeating unit (GenBank: U13369.1) with STAR and the parameters --run-ThreadN 16 --readFilesCommand gunzip -c -k --outReadsUn-mapped Fastx --limitBAMsortRAM 20000000000--outSAMtype BAM SortedByCoordinate. The unmapped reads were mapped to the human GRCh38.p13 reference sequence with STAR and the parameters: --readFilesCommand gunzip -c -k --limitBAMsor-tRAM 20000000000 --outFilterType BySJout --outFilterMulti-mapNmax 20 --alignSJoverhangMin 8 --alignSJDBoverhangMin 1 --outFilterMismatchNmax 999 --outFilterMismatchNoverReadL-max 0.04 --alignIntronMin 20 --alignIntronMax 1000000 --align-MatesGapMax 1000000 --quantMode GeneCounts --outSAMtype BAM SortedByCoordinate. SAMtools was used to retain the properly paired and mapped reads (-f 3) and to create strand-specific BAM files. FPKM-normalized bigwig files were created with

deepTools2 bamCoverage tool with the parameters -bs 10 -p max –normalizeUsing RPKM.

### PRO-seq analysis

For the DLD-1 NELF-C-AID PRO-seq (GSE144786) and HeLa shINTS11 PRO-seq (GSE125535), adapters were trimmed with Cutadapt with the following options: --minimum-length 10 -q 15, 10 -j 16 -a AGATCGGAAGAGCACACGTCTGAACTCCAGTCA. The rRNA reads were removed by mapping the reads to the rRNA genes defined in human and in *Drosophila melanogaster* with STAR and the parameters --runThreadN 16 --readFilesCommand gunzip -c -k --outReadsUnmapped Fastx --limitBAMsortRAM 20000000000--outSAMtype BAM SortedByCoordinate. The unmapped reads were mapped to the human GRCh38.p13 reference genome or the *D. melanogaster* dmel r6 reference genome with STAR and the parameters: --runThreadN 16 --readFilesCommand gunzip -c –k --limitBAMsortRAM 20000000000 --outFilterMultimapNmax 1 --outFilterScoreMin 10 --outSAMtype BAM SortedByCoordinate. SAMtools was used to retain the properly paired and mapped reads (-F 4). SAMtools view with the –s option was used to subsample all the bam files to the bam file containing the lowest number of reads and then to create strand-specific BAM files. The normalization factor was calculated as: (number of *D. melanogaster* reads)/(number of human reads + number of *D. melanogaster* reads). Spiked normalized bedGraph files were created with BEDtools genomecov and the options: -bg -5 –scale (1 / normalization factor). Bigwig files were generated from the BedGraph files with the bedGraphToBigWig tool.

### ChIP-seq analysis

Adapters were trimmed with Cutadapt with the following options: --minimum-length 10 -q 15, 10 -j 16 -a AGATCGGAAGAGCACACGTCTGAACTCCAGTCA. For non-spike-in ChIP-seq (Xrn2 and its respective Input from GSE144786), trimmed reads were mapped to the human GRCh38.p13 reference genome with STAR and the parameters: --runThreadN 16 --readFilesCommand gunzip -c -k –alignIntronMax 1 --outFilterMultimapNmax 20 --limitBAMsortRAM 20000000000 --outSAMtype BAM SortedByCoordinate. SAMtools was used to retain the properly mapped reads and to remove PCR duplicates. Reads mapping to the DAC Exclusion List Regions (accession: ENCSR636HFF) were removed with BEDtools. FPKM-normalized bigwig files were created with deepTools2 bamCoverage tool with the parameters -bs 10 -p max –normalizeUsing RPKM -e 200. For spike-in ChIP-seq, trimmed reads were mapped to the *Drosophila melanogaster* BDGP6.32 and human GRCh38.p13 reference genome with STAR and the parameters: --runThreadN 16 --readFilesCommand gunzip -c -k –alignIntronMax 1 --outFilterMultimapNmax 20 --limitBAMsortRAM 20000000000 --outSAMtype BAM SortedByCoordinate. SAMtools was used to retain the properly mapped reads and to remove PCR duplicates. Reads mapping to the human DAC Exclusion List Regions (accession: ENCSR636HFF) were removed with BEDtools. For spike-in normalization, CSTF64 and INTS3 Drosophila peaks were called against their respective Inputs with MACS3 (Zhang et al, 2008) callpeak and the options -B -f BAMPE -g 1.5e8 -q 0.01 –scale-to small –call-summits. Each peak was then defined as peak summit – 150 bp to peak summit + 150 bp. Only peaks found to overlap between the DMSO and IAA 4 h conditions

were kept, using the overlapping region as the peak region. BEDtools multicov was then used to quantify Input, CSTF64, and INTS3 signals across their respective peaks, normalized for the library size and the length of the peak region, and used for calculating the spike-in normalization factor for the IAA 4 h condition, with the DMSO condition put to 1. After subsampling the human bam files to the smallest library with SAMtools view, spike-in normalized bigwig files were created with deepTools2 bamCoverage tool with the parameters -bs 10 -p max –scaleFactor Normalization_Factor -e.

### ATAC-seq analysis

Adapters were trimmed with Cutadapt with the following options: --minimum-length 10 -q 15, 10 -j 16 -A GATCGTCGGACTGTAGAACTCTGAAC -a AGATCGGAAGAGCACACGTCTGAACTCCAGTCAC. Trimmed reads were mapped to the human GRCh38.p13 reference sequence with STAR and the parameters: --runThreadN 16 --readFilesCommand gunzip -c -k –alignIntronMax 1 --limitBAMsortRAM 20000000000 --outSAMtype BAM SortedByCoordinate. SAMtools was used to retain the properly mapped reads and to remove PCR duplicates. Reads mapping to the DAC Exclusion List Regions (accession: ENCSR636HFF) and to the mitochondrial genome were removed with BEDtools. FPKM-normalized bigwig files were created with deepTools2 bamCoverage tool with the parameters -bs 10 -p max –normalizeUsing RPKM -e.

### TT-seq analysis

Low-quality bases from 3' end of paired-end raw reads were trimmed using cutadapt 4.2. The reads were aligned to the human genome assembly GRCh38 release 103 with its annotation using STAR 2.7.5. The mapped reads with primary alignment and MAPQ $\geq 20$ were retained and used for the downstream analysis. Strand-specific bigwig files were generated using deepTools bamCoverage 3.5.1. To estimate elongation rates, the transition points that correspond to the 5' end of the inhibition waves at 30-min NVP-2 treatment were called using proRate package in R 4.2.3 with BAM files for 0 h and 0.5 h NVP-2 treatment. We selected genes with >60 kb long as we assume that the distance between TSS and the inhibition wave is ~60 kb (~2 kb/min for 30 min). To improve the accuracy of transition point calling, we chose genes that showed >70% reduction in the read density at the genetic intervals (TSS + 1 kb to TSS + 20 kb) upon 30-min NVP-2 treatment. This eliminates genes with high background signal that result in poor calling of the transition points. Genes with reproducible calling of the transition points from 2 replicates were kept ($n = 130$) and used for the downstream analysis.

### Metagene analysis

Metagene profiles of genes scaled to the same length were then generated with Deeptools2 computeMatrix tool with the following parameters: -bs 10 -p max -m 4000 -b 2500 -a 2500. For metaprofiles over the last exon, the following parameters were used: -bs 10 -p max -m 500 -b 1000 -a 2500, while for Pol II EI (TT-seq/POINT-seq), we used: -bs 10 -p max -m 400 -b 0 -a 0. The plotting data were obtained with plotProfile –outFileNameData tool. Graphs representing the POINT-seq, PRO-seq, and ChIP-seq (IP/Input), and Pol II EI signals were then created with GraphPad Prism 10.1. Metagene profiles are shown as the average of two biological replicates.

### Reads quantification

Reads quantification over genomic regions (gene body: TSS to TES, termination region: TES to TES + 2.5 kb, replication regions) was performed on strand-specific bam files with BEDtools multicov and the options -s -split. The reads quantifications were then normalized for library size (either to 100 million mapped paired-end reads or the spike-in normalization factor, as indicated on the figures) and for the region's length. TSS and TES correspond to the 5' and 3'ends of the most expressed transcript, respectively.

### Termination index

The transcription termination index (TI) is defined as: $TI = log2((([TES, TES + 2,500]_{read\ counts} * normalization\ factor)/2500)/ (([TSS, TES]_{read\ counts} * normalization\ factor)/(Gene\ body\ size (TES - TSS))))$. The normalization factor is defined in Reads quantification (see above).

### Splicing efficiency

The splicing efficiency on POINT-seq was calculated as in previous study(Henfrey et al, 2023) by first parsing each bam file to obtain the list of spliced and unspliced reads with the awk command (awk '/^@/ || $6 ~ /N/' for spliced reads and awk '/^@/ || $6 ! ~ /N/' for unspliced reads). The splicing efficiency was then calculated as the number of spliced reads over total reads with BEDtools multicov and the options –s –split.

### Data visualization

Box plots and violin plots, representing min to max with first quartile, median, and third quartile values, were made with GraphPad Prism 10.1.

### Image analysis

For P-Histone H2A.X (S139), fluorescent intensity was analyzed and quantified with ImageJ (version 1.53q) using two steps: first, the DAPI channel (λEx/λEm (with DNA) = 350/461 nm) image defined the nucleus; then, the GFP channel (λEx/λEm = 495/519 nm) measured the fluorescent intensity within the nucleus. To find nuclear borders and area, threshold and watershed were applied to DAPI images. For P-Histone H2A.X (S139) peak counts, ImageJ's analyze particles tool was used with settings of 1000-Infinity and 0.00-1.00 circularity. The number of peaks and DAPI area were recorded, and peak counts were normalized to DAPI area (Peak Count/DAPI area). For PLA signal quantification, nuclei segmentation was performed using the AI-based segmentation tool Cellpose (Stringer et al, 2021). Segmentation files (.roi) created by Cellpose were then filtered to retain regions positive for MCM2 or PCNA. Quantification of PLA foci of MCM2 or PCNA-positive nuclei was performed using the Particle Analysis function in ImageJ (Fiji). The number of nuclei used for the Particle Analysis in each condition is indicated in the figures. For statistical analysis, Brunner–Munzel test were conducted to evaluate differences in signal counts between groups. Prior to conducting statistical tests, we performed an exploratory data analysis (EDA) to evaluate the distribution of the dataset. EDA revealed that more than 80% of the data values were zero for MCM2 PLA and PCNA PLA, indicating severe zero-inflation. A statistical test was not performed for such dataset since severe zero-inflation is known to violate statistical tests.

### BrdU-seq analysis

Analytical processes were performed as previously described (Daigaku et al, 2017), with optimizations for the human genome. For each sample, ~200 million PE reads were obtained. Raw reads were aligned to GRCh38 using Bowtie2 (version 2.3.5). Those that aligned to multiple genomic locations with the same mismatch scores (AS and XS scores as outputted by Bowtie2) were excluded using a custom Perl script: sam-dup-align-exclude-v2.pl (available at the GitHub site: https://github.com/yasukasu/sam-dup-align-exclude). The position of the center of each read was determined, and the number of reads in 1-kb bins across the genome were counted using a custom Perl script: pe-sam-to-bincount.pl (available at the GitHub site: https://github.com/yasukasu/sam-to-bincount). At the chromosome coordinate x, CB(x) is the count for the BrdU-IP sample; CI(x) is count for the input sample. After obtaining the genome-wide read count data, genomic bins where the count of the input samples are less than 5 (e.g., $CI(x)<5$) were excluded from further calculation for BrdU enrichment. These datasets were then normalized with the total number of reads: $NB(x) = CB(x)/\Sigma CB(x)$, $NI(x) = CI(x)/\Sigma CI(x)$. Enrichments for BrdU-incorporated fragments were calculated: $EB(x) = NB(x)/NI(x)$. When these data were used for further analysis, they were smoothed using a moving average of $2\,m + 1$; the data point for each bin is an average of $2\,m + 1$ bins: the central bin and the m bins either side. In this study, m was set 10. The normalized reads were processed by subtracting the control condition (Parental cell +IAA 4 h) from NELF-C knockdown condition (NELF-C-AID + IAA 4 h) using bigiwigCompare (version 3.5.4). Values within ±10 were considered background noise and excluded from further analysis. For POINT-seq data, SIRV normalized read coverage was similarly processed by subtracting the control condition (NELF-C-AID + DMSO) from NELF-C knockdown condition (NELF-C-AID + IAA 4 h). Then, the average read score on the termination zone (TES + 2.5 kb to TES + 5 kb) was computed using bigWigAverageOverBed(version 2). Lastly, to assess transcriptional termination defect corresponding to BrdU incorporation, subtracted POINT-seq read coverage overlapping with BrdU-IP-seq read enrichment files were quantified using bedtools intersect. The output file was subsequently separated into regions with positive or negative BrdU-IP-seq subtraction values. For statistical analysis, the Brunner–Munzel test was conducted to evaluate whether POINT-seq subtracted read coverage differed significantly between genomic regions with positive and negative BrdU-IP-seq subtraction values.

### P values and significance tests

P values were computed with Wilcoxon rank-sum test, Wilcoxon signed-rank test, paired two-sample t test (Student's t test), or Brunner–Munzel test, as indicated in each figure legend. Statistical tests were performed in GraphPad Prism 10.1 and P values for values below 0.0001 are shown as ****. Note that for Figs. 1F, 2C, 6B, EV1C, and EV8B, statistical tests were performed in R, and the exact P values are provided in the figure.

## Data availability

Raw and processed data of POINT-seq, ATAC-seq, TT-seq, ChIP-seq, and nuclear pA+RNA-seq generated in this study are

deposited in NCBI GEO and accessible through GSE253121 (https://www.ncbi.nlm.nih.gov/geo/query/acc.cgi?acc=GSE253121). Also, BrdU-IP sequencing data are deposited in NCBI GEO and accessible through GSE294597 (https://www.ncbi.nlm.nih.gov/geo/query/acc.cgi?acc=GSE294597). Raw image data associated with this study are available at BioImage Archive (https://www.ebi.ac.uk/biostudies/bioimages/studies/S-BIAD2495). All code supporting sequencing analyses are available here: https://github.com/tellierlab/NELF_Transcription_Replication_2025. Published data re-analyzed in this study can be found at GEO in the Reagents and Tools Table.

The source data of this paper are collected in the following database record: biostudies:S-SCDT-10_1038-S44319-026-00700-z.

## Peer review information

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

## Acknowledgements

We thank Dr. Mikita Suyama and Dr. Chie Kikutake for help with the computational analyses, Mrs. Eri Iwai for technical assistance, and Dr. Nick J Proudfoot and Dr. Kinga Kamieniarz-Gdula for critical comments on this paper. We also thank the Advanced Imaging Facility (RRID:SCR_020967) at the University of Leicester for support. This research used the ALICE High Performance Computing facility at the University of Leicester. This work was supported by funding to TN (Royal Society (ICA\R1\231018), MEXT/JSPS Kakenhi (JP19K24692, JP24K01957) and JST FOREST (JPMJFR2050), Astellas Foundation for Research on Metabolic Disorders, Daiichi Sankyo Foundation of Life Science, The Ichiro Kanehara Foundation for Promotion of Medical Sciences and Medical Care, The Mitsubishi Foundation, The Mochida Memorial Foundation for Medical and Pharmaceutical Research, The Naito Foundation, The NOVARTIS Foundation (Japan) for the Promotion of Science, Princess Takamatsu Cancer Research Fund, The Shinnihon Foundation of Advanced Medical Treatment Research, The Sumitomo Foundation, and The Uehara Memorial Foundation), MT (Royal Society (ICA\R1\231018), MRC (MR/W007002/1)), YO (University of Leicester F50 PhD studentship), YD (MEXT/JSPS Kakenhi (JP23H02463), JST FOREST (JPMJFR204X), Yamada Science Foundation)), AS (NIH (R35CA197569)), and Fellowship to CN (JST SPRING (JPMJSP2136), RIKAKEN HD, JSPS DC2 (JP25KJ1964)). This work was also supported in part by the MEXT Cooperative Research Project Program, Medical Research Center Initiative for High Depth Omics, and CURE (JPMXP1323015486). The infrastructure of Omics Science Center Secure Information Analysis System, Medical Institute of Bioregulation at Kyushu University provides part of the computational resource.

## Author contributions

**Chihiro Nakayama**: Data curation; Formal analysis; Funding acquisition; Investigation; Writing—original draft; Writing—review and editing. **Qi Fang**: Data curation; Formal analysis; Writing—original draft; Writing—review and editing. **Yasukazu Daigaku**: Conceptualization; Data curation; Formal analysis; Supervision; Funding acquisition; Investigation; Writing—original draft; Writing—review and editing. **Yuki Aoi**: Data curation; Formal analysis; Investigation; Writing—review and editing. **Shoko Ito**: Data curation; Investigation; Writing—

review and editing. **Mami Takahashi**: Data curation; Investigation; Writing—review and editing. **Reo Shimatani**: Data curation; Formal analysis; Investigation; Writing—review and editing. **Tamiko Minamisawa**: Data curation; Investigation; Writing—review and editing. **Yagiz Ozturk**: Data curation; Investigation; Writing—review and editing. **Hiroshi Kimura**: Investigation; Writing—review and editing. **Ali Shilatifard**: Supervision; Writing—review and editing. **Michael Tellier**: Conceptualization; Formal analysis; Supervision; Funding acquisition; Investigation; Writing—original draft; Writing—review and editing. **Takayuki Nojima**: Conceptualization; Supervision; Funding acquisition; Writing—original draft; Writing—review and editing.

Source data underlying figure panels in this paper may have individual authorship assigned. Where available, figure panel/source data authorship is listed in the following database record: biostudies:S-SCDT-10_1038-S44319-026-00700-z.

## Disclosure and competing interests statement

The authors declare no competing interests.

# Expanded View Figures

**Figure EV1.  Expression of *NELFCD* transcripts is highly upregulated in colorectal cancer cells.**

(A) Log$_2$ fold change of T vs N of the Pol II transcription-associated genes. Upregulated (>1) and down-regulated (<-1) genes are highlighted in red and blue, respectively. *NELFCD* gene is indicated with an arrow. (B) Volcano plot of the log2 fold change of Tumors (T) vs Normal tissues (N) on 835 cell cycle-related genes in COAD. *CDKN1A*, *CDKN1B*, and *CDKN1C* genes are indicated in red. The numbers of tumor (T) and normal (N) samples for COAD are provided in Table EV1. Box plots show the median (center line) and interquartile range (box, 25th–75th percentiles); whiskers indicate 1.5×IQR. (C) Log2 of the RNA expression levels of *NELFCD*, *SUPT4H1*, *CDKN1A*, and *CDKN1C* genes in N and T of the indicated tissues and tumors are compared. Box plots show the median (center line) and interquartile range (box, 25th–75th percentiles); whiskers indicate 1.5×IQR. Statistical test: Wilcoxon signed-rank test. *P* values are shown. The numbers of tumor (T) and normal (N) samples for each tissue types are provided in Table EV1. (D) Log$_2$ fold change of T vs N of *NELFCD* across indicated tissue types. (E) Comparative proteomic analysis of NELF subunits and SPT4 in human primary colon cancers and its adjacent tissues. Statistical test: negative binomial distribution-based Wald test. *P* values are shown. Box plot: minimal-to-maximal value, box center line: median, bounds of box: interquartile (25 and 75%). (F) Schematic model of transcription addicted by NELF in tumor.

▶

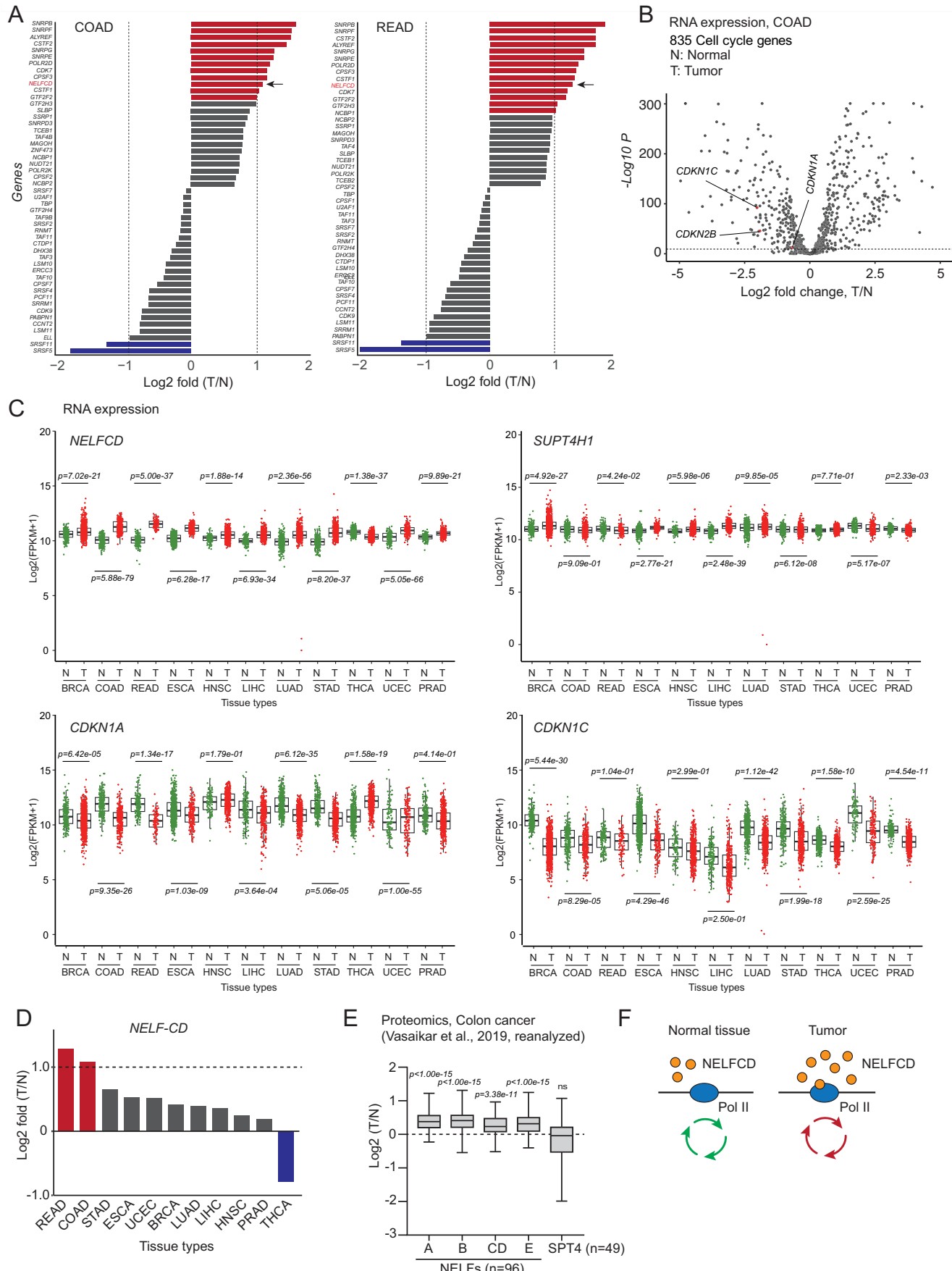

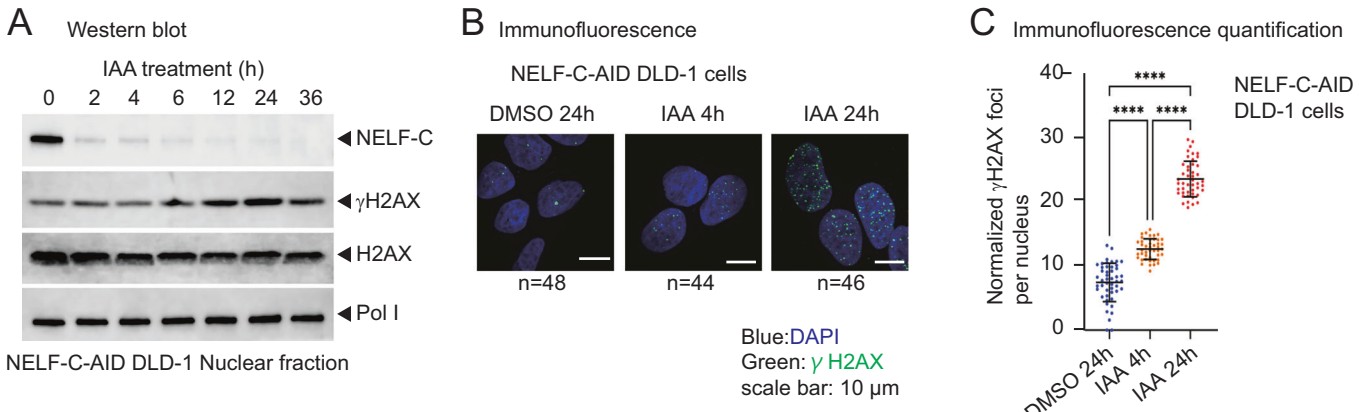

**Figure EV2. Loss of NELF induces DNA damage.**

(A) Western blot of NELF-C-AID DLD-1 cell nuclear fraction using the indicated antibodies. Treatment time (h) of IAA is also indicated. Western blot images of NELF-C and Pol I are reused in Fig. 1E. (B) Immunofluorescent image of parental and NELF-C-AID DLD-1 cells in 24 h DMSO and 4 h or 24 h IAA. Merged images with DAPI (blue) and γH2AX antibody (green). Scale bar size is 10 μm. Number of cells for quantification (for Fig. EV2C) is indicated. (C) Quantification of normalized γH2AX foci per nucleus in NELF-C-AID DLD-1 cells in 24 h DMSO and 4 h or 24 h IAA. Statistical test: Kruskal–Wallis test for NELC-AID cells. ns: not significant, ****P < 0.0001. Error bars represented the mean ± SD (biological replicates, n = 2). NELF-C-AID DLD-1 cells in 24 h DMSO (n = 48) and 4 h (n = 44) or 24 h IAA (n = 46).

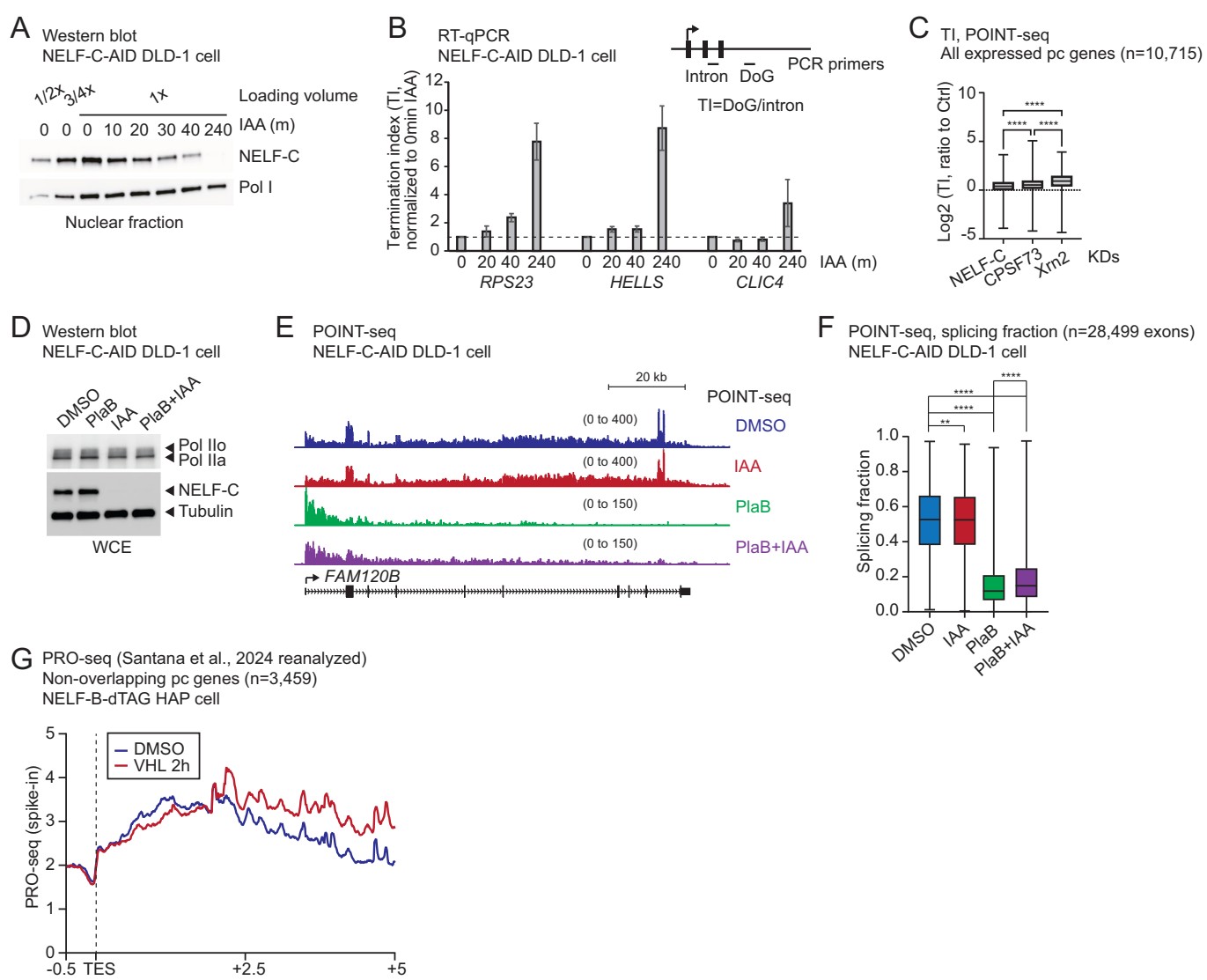

**Figure EV3.   Acute NELF depletion induces a Pol II transcription termination failure.**

(A) Western blot of NELF-C-AID DLD-1 cell nuclear fraction using the indicated antibodies. Loading volume (x1/2, x3/4 or x1) and treatment time (min) of IAA are indicated. (B) RT-qPCR analysis for TI (DoG/intron) of *RPS23*, *HELLS*, and *CLIC4* genes in NELF-C-AID DLD-1 cell. Treatment time of IAA is indicated. Primer information is available in Table EV2. Error bars represented the mean ± SEM (biological replicates, $n = 3$). (C) Box plots of TI of all expressed genes in indicated AID-tagged cells. Wilcoxon rank-sum test. ***$P < 0.001$. Box plot: minimal-to-maximal value, box center line: median, bounds of box: interquartile (25 and 75%). (D) Western blot of NELF-C-AID DLD-1 WCE (4 h DMSO, PlaB, IAA, PlaB+IAA) using the indicated antibodies. (E) Example view of POINT-seq on *FAM120B* genes in NELF-C-AID DLD-1 cells (4 h DMSO, PlaB, IAA, PlaB+IAA). (F) Quantification of splicing fraction of POINT-seq in NELF-C-AID DLD-1 cells (4 h DMSO, IAA, PlaB, PlaB+IAA). Statistical test: Friedman test. **$P = 0.001$, ****$P < 0.0001$. Box plot: minimal-to-maximal value, box center line: median, bounds of box: interquartile (25 and 75%). (G) Metagene analysis of PRO-seq around PAS of non-overlapping pc genes in NELF-B-dTAG HAP cells (DMSO and a protein degradation inducer VHL for 2 h).

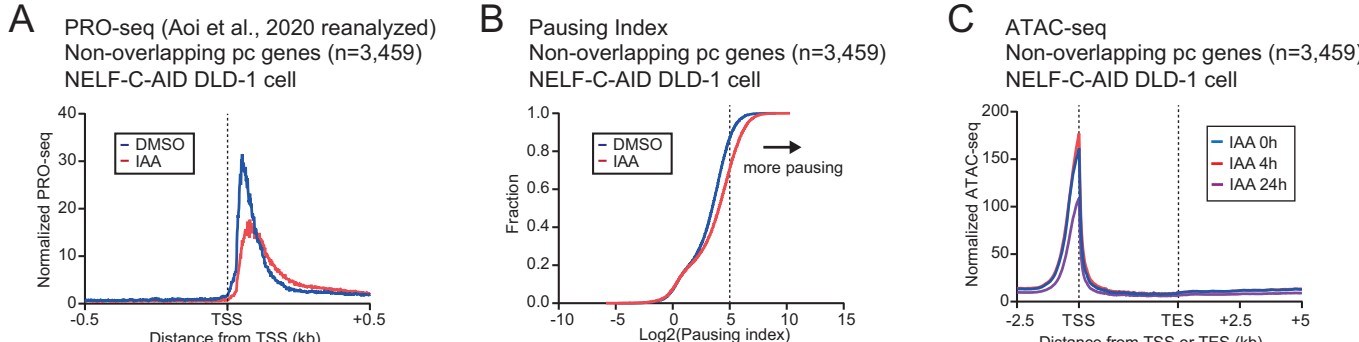

**Figure EV4.  Acute loss of NELF increases Pol II pausing index and transcriptional initiation.**

(A) Metagene analysis of published PRO-seq on TSS $-/+0.5$ kb of non-overlapping pc genes. (B) Density plot of the pausing index of non-overlapping pc genes calculated from DLD-1 NELF-C AID PRO-seq cells treated with DMSO (blue) or IAA for 4 h (red). A shift of the density plot to the right indicates a higher pausing. (C) Metagene analysis of ATAC-seq for non-overlapping pc genes in NELF-C-AID DLD-1 cells (0, 4, and 24 h IAA).

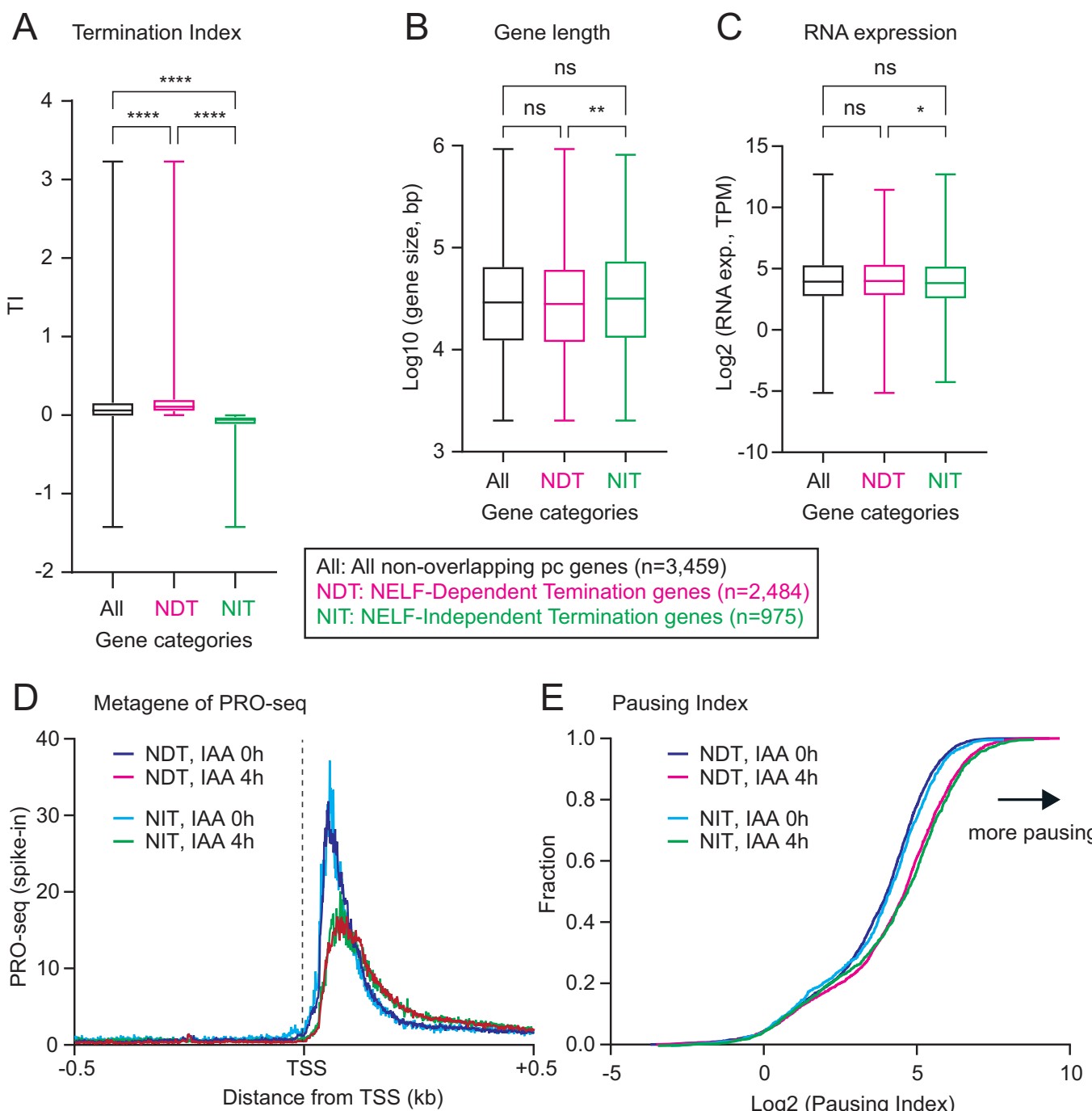

**Figure EV5. Characters of highly and lowly NELF-dependent termination genes.**

(A) Box plots of log$_2$(TI) in POINT-seq in NELF-C-AID DLD-1 cells (4 h vs 0 h IAA). Genes were classified to NELF-dependent termination (NDT, $n = 2484$) and NELF-independent termination (NIT, $n = 975$) gene categories by TI. N values are indicated on the figure panel. Statistical test: Wilcoxon signed-rank test. ****$P < 0.0001$. Box plot: minimal-to-maximal value, box center line: median, bounds of box: interquartile (25 and 75%). (B) Box plots of log$_{10}$ (gene size, bp) of the indicated categories. Statistical test: Kruskal–Wallis test. **$P = 0.0097$, ns: not significant. Box plot: minimal-to-maximal value, box center line: median, bounds of box: interquartile (25 and 75%). (C) Box plots of log$_2$ (RNA expression level, TPM) of the indicated categories. Statistical test: Kruskal–Wallis test. *$P = 0.0176$, ns: not significant. Box plot: minimal-to-maximal value, box center line: median, bounds of box: interquartile (25 and 75%). (D) Metagene analysis of published PRO-seq on TSS $-/+ 0.5$ kb of the indicated categories. (E) Density plot of the pausing index of the indicated categories calculated from NELF-C AID DLD-1 cells.

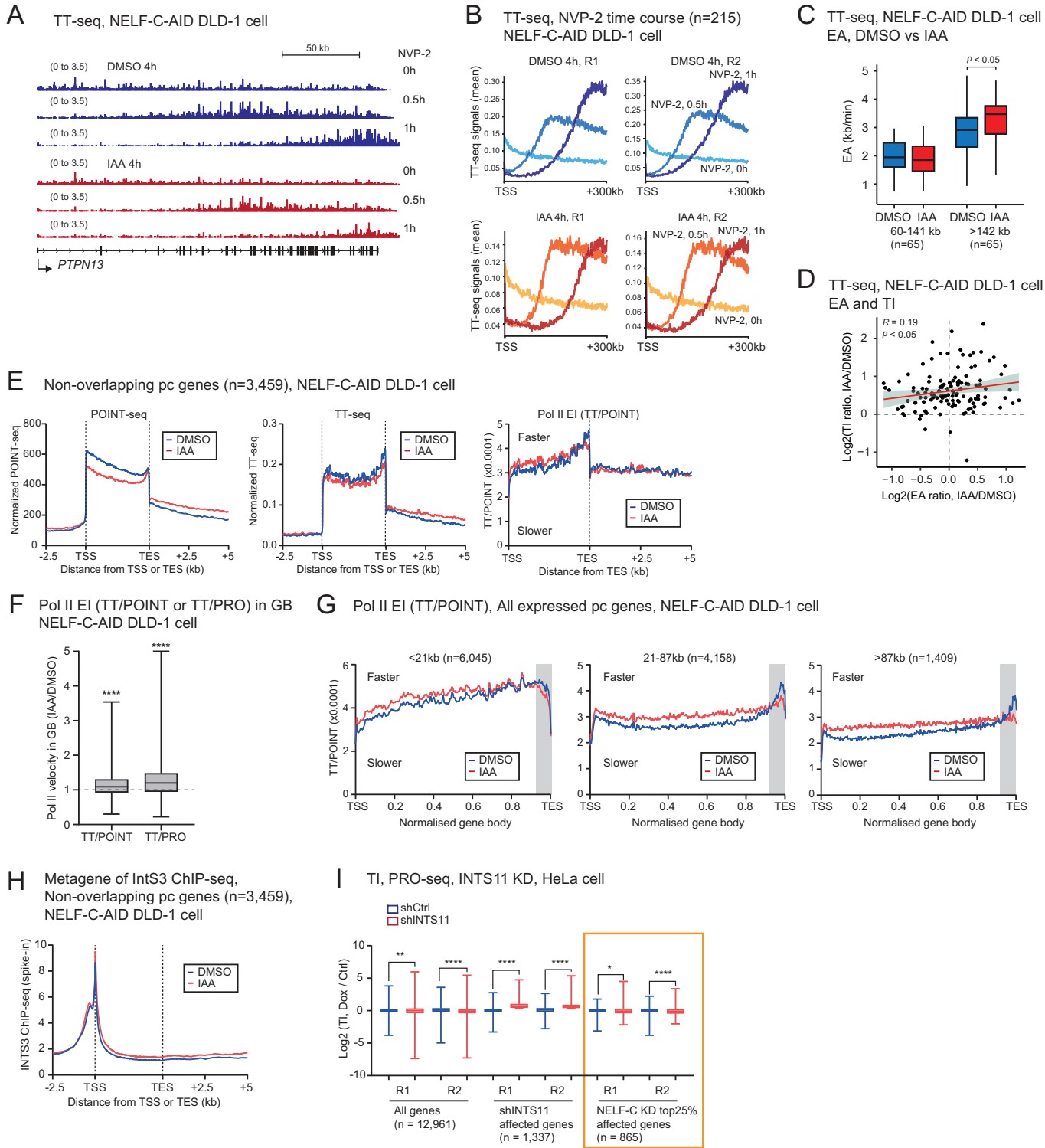

◀

**Figure EV6.   Characters of highly and lowly NELF-dependent termination genes.**

(A) Example view of TT-seq on *PTPN13* gene in the indicated cell line treated for 4 h with DMSO or IAA, followed by NVP-2 treatment for 0, 0.5, or 1 h. (B) Profiles of TT-seq mean signal in NELF-C-AID DLD-1 cells treated for 4 h with DMSO or IAA, followed by NVP-2 treatment for 0, 0.5, or 1 h. Data for 2 biological replicates with ≥300 kb genes are shown. (C) Box plots of elongation activity (EA) derived from TT-seq. Data for short and long genes are shown. $n = 65$. Statistical test: Wilcoxon rank-sum test. Box plot: minimal-to-maximal value, box center line: median, bounds of box: interquartile (25 and 75%). *P* value is 0.029. (D) Scatter plot showing the correlation between EA and TI. The regression line (red) and the 95% confidence interval (green) are also shown. $n = 129$. Statistical method: Spearman correlation. *P* value is 0.036. (E) Metagene of POINT-seq, TT-seq, Pol II EI (TT-seq/POINT-seq) for non-overlapping pc genes in NELF-C-AID DLD-1 cells treated for 4 h with DMSO or IAA. (F) Box plots of Pol II EI (TT-seq/POINT-seq and TT-seq/PRO-seq) for non-overlapping pc genes ($n = 3459$) in NELF-C-AID DLD-1 cells treated for 4 h with DMSO or IAA. Statistical test: Kruskal–Wallis test. ****$P < 0.0001$. Box plot: minimal-to-maximal value, box center line: median, bounds of box: interquartile (25 and 75%). (G) Pol II EI for indicated three gene-length classes of normalized non-overlapping pc genes in NELF-C-AID DLD-1 cells treated for 4 h with DMSO or IAA. (H) INTS3 ChIP-seq profile across normalized transcription units of non-overlapping pc genes in NELF-C-AID DLD-1 cells treated for 4 h with DMSO or IAA. (I) Box plots of the log2 of the Dox/Ctrl termination index, two biological replicates in all genes, genes with transcription readthrough which is based on previous study (Dasilva et al, 2021), and NELF-C KD affected genes (top 25%, highlighted in orange box). 0: no change in TI index upon shCtrl or shINTS11 induction by Dox. The number of genes in each category is indicated on the figure. Statistical test: Kruskal–Wallis test. *$P = 0.039$, **$P = 0.0019$, ****$P < 0.0001$. Box plot: minimal-to-maximal value, box center line: median, bounds of box: interquartile (25 and 75%).

                                                                          

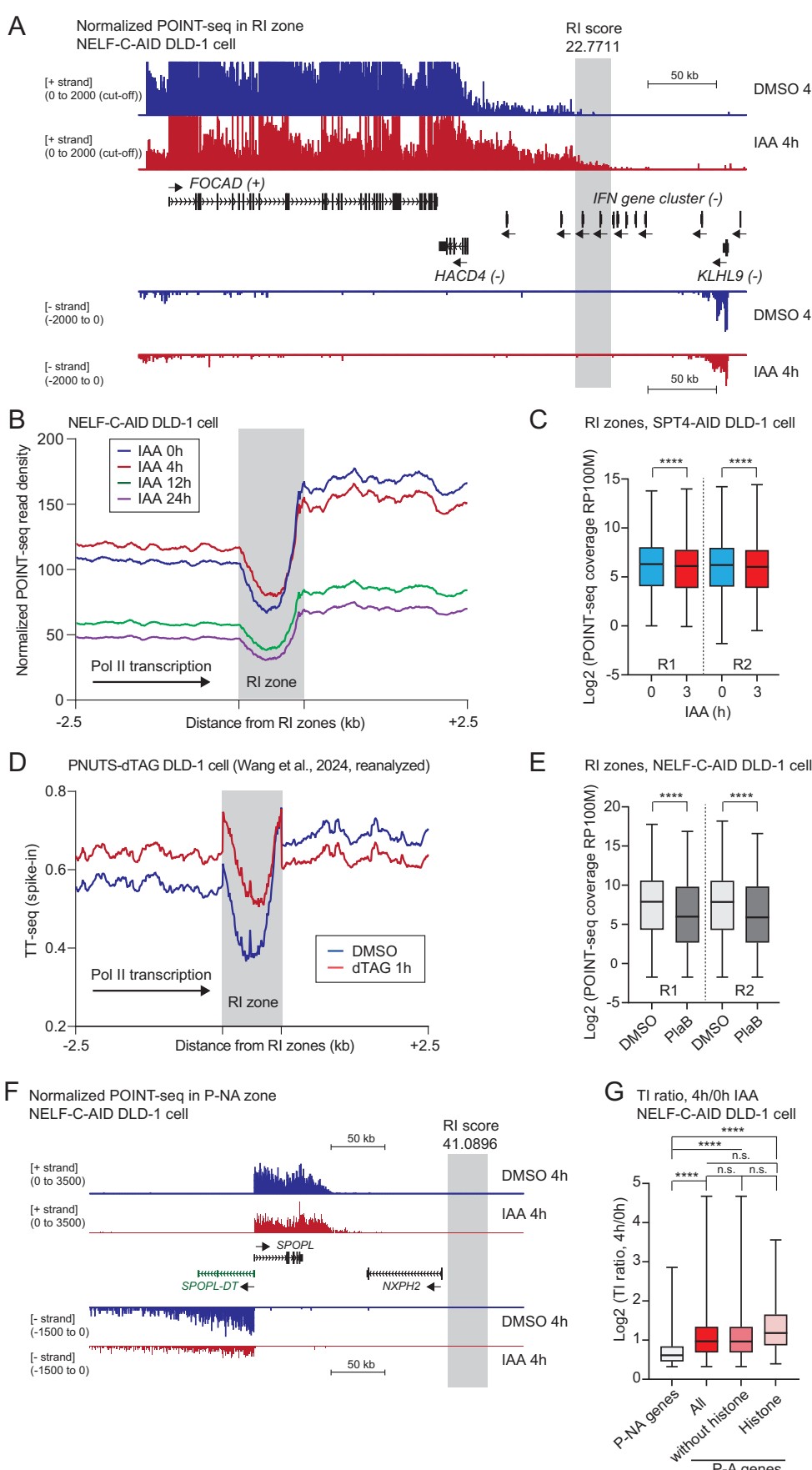

◀ **Figure EV7. Acute depletion of NELF-C causes Pol II transcription invasion into DNA replication initiation zone.**

(A) Example view of SIRV-normalized POINT-seq on *FOCAD* gene adjacent to RI zone in NELF-C-AID DLD-1 cells (4 h DMSO and IAA). Normalized POINT-seq signals are cut-off at 2000. RI zone is highlighted in gray. RI score: 22.7711. Both (+) and (−) strands are shown. (B) Metagene analysis of POINT-seq on RI zones -/+2.5 kb in NELF-C-AID DLD-1 cells (0, 4, 12, 24 h IAA). (C) Box plots of normalized POINT-seq signals in RI zones in SPT4-AID DLD-1 cells (0 h and 3 h IAA). Two replicates are shown. (R1: $n = 9997$ RI zones, R2: $n = 9597$ RI zones). Statistical test: Wilcoxon test. ****$P < 0.0001$. Box plot: minimal-to-maximal value, box center line: median, bounds of box: interquartile (25 and 75%). (D) Metagene analysis of TT-seq on RI zones $-/+2.5$ kb in PNUTS-dTAG DLD-1 cells (0 and 1 h dTAG). (E) Box plots of normalized POINT-seq signals in RI zones in NELF-C-AID DLD-1 cells (4 h DMSO and PlaB). Two replicates are shown. (R1: $n = 9997$ RI zones, R2: $n = 9597$ RI zones). Statistical test: Wilcoxon test. ****$P < 0.0001$. Box plot: minimal-to-maximal value, box center line: median, bounds of box: interquartile (25 and 75%). (F) Example view of SIRV-normalized POINT-seq on *SPOPL* gene adjacent to P-NA zone in NELF-C-AID DLD-1 cells (4 h DMSO and IAA). RI zone is highlighted in gray. RI score: 42.8401. Both (+) and (−) strands are shown. (G) Box plots of TIs in POINT-seq in NELF-C-AID DLD-1 cells (4 h vs 0 h IAA). Genes in P-NA ($n = 6651$) and P-A (all ($n = 5878$), all without RDH ($n = 5849$), and RDH ($n = 29$) genes) zones were analyzed. Statistical test: Kruskal–Wallis test. ns: not significant, ****$P < 0.0001$. Box plot: minimal-to-maximal value, box center line: median, bounds of box: interquartile (25 and 75%).

## A

PLA, NELF-C-AID DLD1 cells

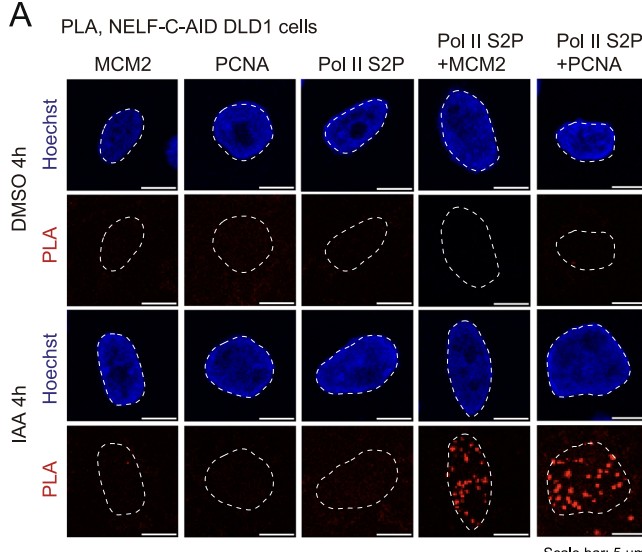

Scale bar: 5 μm

## C

Western blot

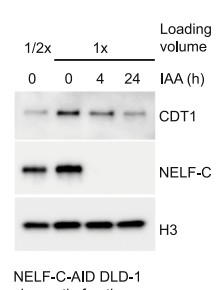

NELF-C-AID DLD-1
chromatin fraction

## D

Cell cycle, NELF-C-AID DLD-1 cell

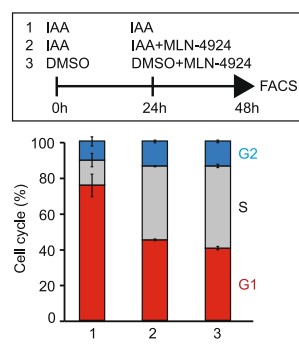

## B

PLA quantification, NELF-C-AID DLD1 cells

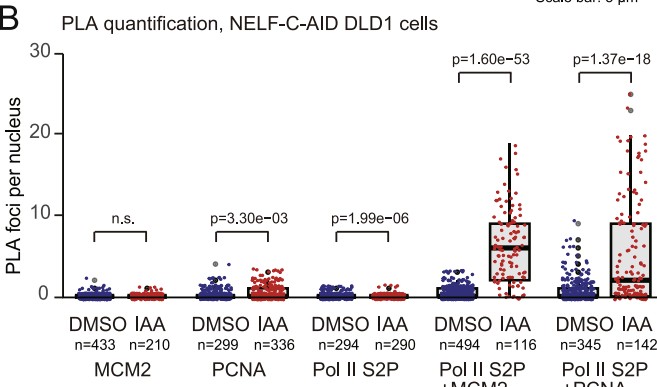

## E

BrdU dot blot
NELF-C-AID DLD-1 cell

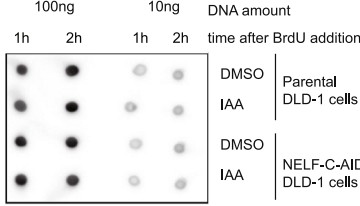

## F

BrdU IP-seq and POINT-seq
Parental and NELF-C-AID DLD-1 cells

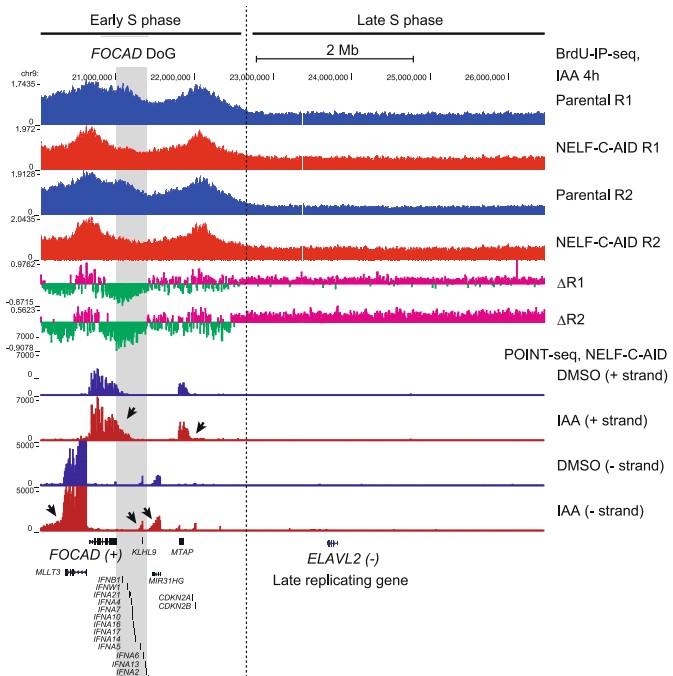

## G

BrdU IP-qPCR
Parental and NELF-C-AID DLD-1 cells

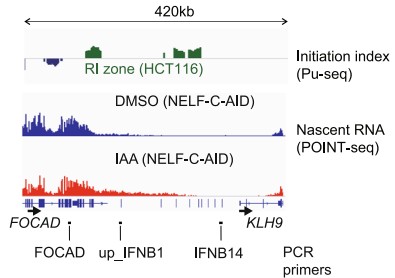

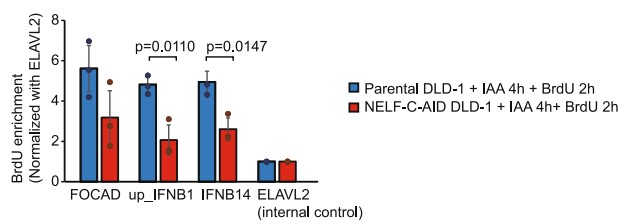

■ Parental DLD-1 + IAA 4h + BrdU 2h
■ NELF-C-AID DLD-1 + IAA 4h + BrdU 2h

**Figure EV8.  Acute depletion of NELF-C protein causes a conflict between transcription and replication and suppresses DNA replication.**

(A) Representative images of PLA with the indicated antibodies after 4 h DMSO and IAA in NELF-C-AID DLD-1 cells. Blue: Hoechst, Red: PLA. Scale bar size is 5 μm. (B) Box plots of the PLA foci per nucleus after 4 h DMSO and IAA in NELF-C-AID DLD-1 cells. The PLA was performed with indicated antibodies. Box plots show the median (center line) and interquartile range (box, 25th–75th percentiles); whiskers indicate 1.5×IQR. Dots represent individual nuclei. Statistical test: Brunner–Munzel test. Statistical test: Wilcoxon rank-sum test. *P* values are shown. not significant (n.s.). (C) Western blot of chromatin fraction of NELF-C-AID DLD-1 cells (0, 4, and 24 h IAA) using the indicated antibodies. (D) Cell cycle (%) of 24 h DMSO or MLN-4924 treated NELF-C-AID DLD-1 cells. The cells were pre-treated with IAA or DMSO for 24 h. Error bars represented the mean ± SEM (biological replicates, *n* = 3). (E) BrdU dot blot assay to evaluate global DNA synthesis. (F) BrdU-IP-seq of two biological replicates (R1 and R2) for FOCAD gene regions in parental and NELF-C-AID DLD-1 cells treated with IAA for 4 h. IP efficiencies (IP/input) are shown. BrdU plus (magenta) and minus (green) zones in Parental minus NELF-C-AID (Δ). Normalized POINT-seq profiles for (+) and (−) strands in NELF-C-AID DLD-1 cells treated with IAA for 0 h (DMSO) or 4 h (IAA) are shown. Pol II transcription termination defect are indicated by arrows. FOCAD DoG region is highlighted by gray. Early and Late S phase are separated by dashed line. (G) BrdU-IP-qPCR analysis to assess local DNA synthesis rates in region exhibiting termination defect. The top panel shows the locations of primer sets used for qPCR. The bottom panel displays the IP/input values for each locus, normalized to the *ELAVL2* locus, a late-replicating region that is not expected to replicate at the onset of the S phase (ELAVL2 locus = 1). Error bars represented the mean ± SEM (biological replicates, *n* = 3). *P* values are indicated.

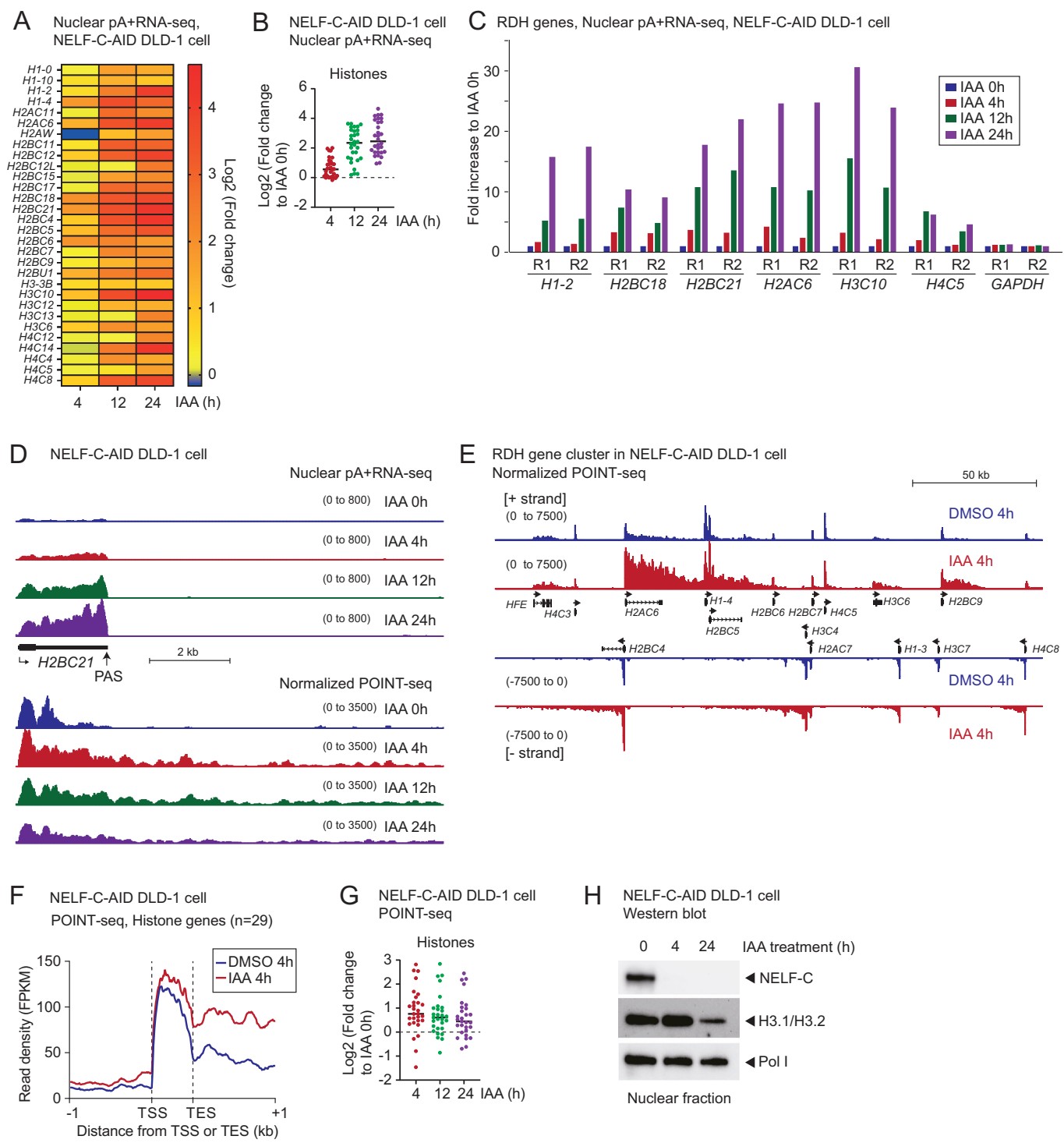

**Figure EV9.  Loss of NELF-C protein induces Pol II transcription readthrough and pA-tailed RNA (pA + RNA) production of replication-dependent histone (RDH) genes.**

(A) Heatmap analysis of nuclear pA+ RNA-seq in NELF-C-AID DLD-1 cells (4, 12, and 24 h IAA). Fold change to 0 h IAA is displayed in the indicated color. (B) Log$_2$ fold change of nuclear pA+ RNA-seq signals of RDH genes ($n = 29$) in NELF-C-AID DLD-1 cells (4, 12, and 24 h IAA to 0 h). (C) Fold change of RNA expression level (two biological replicates) of the indicated RDH genes in NELF-C-AID DLD-1 cells (0, 4, 12, and 24 h IAA). *GAPDH* gene expression was not changed. (D) Example view of nuclear pA+ RNA-seq and SIRV-normalized POINT-seq on H2BC21 gene in NELF-C-AID DLD-1 cells (0, 4, 12, and 24 h IAA). PAS is indicated with an arrow. (E) View of POINT-seq of RDH gene cluster of NELF-C-AID DLD-1 cells (4 h vs 0 h IAA). Both (+) and (−) strands are shown. (F) Metagene analysis of POINT-seq on RDH genes ($n = 29$) of NELF-C-AID DLD-1 cells (4 h vs 0 h IAA). (G) Log$_2$ fold change of SIRV-normalized POINT-seq signals on RDH genes ($n = 29$) in NELF-C-AID DLD-1 cells (4, 12, and 24 h IAA). (H) Western blot of NELF-C-AID DLD-1 nuclear fraction (0, 4, and 24 h IAA) using the indicated antibodies. Pol I is analyzed as a loading control.

