## [Peer Review File · EMBO Reports]

NELF prevents transcriptional readthrough into DNA replication zones in cancer cells

Chihiro Nakayama, Qi Fang, Yasukazu Daigaku, Yuki Aoi, Shoko Ito, Mami Takahashi, Reo Shimatani, Tamiko Minamisawa, Yagiz Ozturk, Hiroshi Kimura, Ali Shilatifard, Michael Tellier, and Takayuki Nojima

Corresponding author(s): Takayuki Nojima (taka.nojima@bioreg.kyushu-u.ac.jp), Yasukazu Daigaku (yasukazu.daigaku@jfc.or.jp), Michael Tellier (mt477@leicester.ac.uk)

Review Timeline:

Transfer Date:	3rd Nov 25
Editorial Decision:	2nd Dec 25
Revision Received:	15th Dec 25
Accepted:	13th Jan 26

Editor: Bernd Pulverer

Transaction Report: This manuscript was transferred to EMBO reports following arbitrating peer review at The EMBO Journal, which was preceded by full peer review and revision at another journal.

**Response letter to reviewer's comments
for 3rd revised ms in *[journal name redacted]***

[journal name redacted] Reviewer #5 (Remarks to the Author):

The authors provide a lengthy rebuttal to the concerns raised. Some of the aspects are clarified, including the shift in the CSTF64 and XRN2 localization in NELF deficiency. The main claims through, that NELF would "coordinate transcription termination and DNA replication", remain inadequately backed-up.

I would suggest the authors to reconsider the title to state what is shown, e.g. in style:

"NELF-depletion redistributes termination factors and increases transcription read-through in replication origins."

We have changed the title as follows, **"NELF prevents widespread transcriptional read-through into DNA replication zones of cancer cells."**

Some comments on the rebuttal:

Concern 1)

The authors state that "The expression of some genes (such as CDKN1C) are transcriptionally upregulated by NELF depletion (Please see extra Fig. 1)". This is not only some genes. There are more genes up- than down-regulated upon NELF-C depletion (Fig. S2C). The authors show more transcribing Pol II at the ends of genes (DoGs, Fig. 6d), but also more polyA+ nuclear transcripts. They continue: "We think this is caused by secondary effect on replication stress..." It remains unclear how. By less division causing more transcripts per cell? This is a 4h time point, the increase in nuclear polyA+ RNAs goes up several folds. (I could again offer the possible hypothesis on increased RNA stability or perhaps reduced export from the nucleus.)

I'm not sure what the extra Fig. 1 is aiming to show. I see an increased POINT-seq and nuclear pA+RNA-seq across the whole gene. There is a lot of text, which does not answer to the original question on why there would be more nuclear polyA+RNA if termination was severely defected.

Overall, the explanation that I read the authors to provide is that NELF-C depletion reduces

XRN2 and, particularly, CSTF64 localization to gene ends (Fig 4d), causing Pol II to transcribe longer into the termination windows (DoGs). This I can agree with. However, I would expect reduced nascent RNA cleavage (less CSTF64) free less transcripts for polyadenylation.

Unfortunately, this reviewer misunderstood our response. Pol II transcription is terminated downstream of polyadenylation site (PAS) and nascent RNA is cleaved when Pol II passes PAS (PMID: 29768209, PMID: 26575290, PMID: 25989971) (also see Figure below), although the RNA cleavage timing in transcription is still unclear. NELF-C depletion reduced CstF64 recruitment at PAS and induced transcription readthrough. We think nascent RNA can be still cleaved (RNA cleavage may be delayed) even in NELF-C depleted cells, since CstF64 was not completely lost at PAS. Also, RNA can be post-transcriptionally cleaved (PMID: 23562152, PMID: 36762421). Therefore, pA⁺ RNAs are produced from read-through transcripts. If a gene is transcriptionally and hugely upregulated by NELF-C loss, the pA⁺RNA is potentially upregulated since read-through RNAs can still be cleaved. In addition, upregulation of genes following NELF-C loss has been investigated by Ali Shilatifard's group in a recent preprint (PMID: 40599158)

Acute loss of NELF-C induces Pol II transcription termination defect. Readthrough transcript can be co-transcriptionally cleaved by CPA factor, but it may be delayed compared to control condition. The timing of RNA cleavage at PAS is still unresolved.

Normalized POINT-seq signals of some cell cycle inhibitor genes (such as *CDKN1C*) are significantly increased after NELF-C depletion (previous Extra Figure 1, see below), indicating these genes are at least transcriptionally upregulated, an observation also made independently by Ali Shilatifard's group in their recent preprint. As we explain above, read through transcripts can be polyadenylated co- or post-transcriptionally. However, we understand the pA+ RNA data will potentially confuse some readers. Additionally, we cannot exclude the possibility that NELF-C may be involved in RNA stability and export although it is hardly likely. As our pA+ RNA-seq was not performed with spike-in, in comparison to our POINT-seq, TT-seq, and CHIP-seq, we also cannot exclude that the DESeq2 normalisation, which considers that the majority of genes are not differentially expressed, is biasing the results and interpretations of the data. Therefore, we have removed all the pA+ RNA analysis from our manuscripts and added a discussion about RNA stability that may be increased after NELF-C loss.

Previous Extra Fig.1: POINT-seq and Nuclear pA+RNA-seq in NELF-C-AID DLD-1 cells

Spike-in normalized POINT-seq signals are increased after NELF-C depletion (IAA 4h) compared to control (IAA 0h). The pA+ Nuclear RNA-seq signals are also increased, but the data was normalized by DESeq2, not by spike-in.

Concern 2)

I can agree that Pol II S2P co-localises with PCNA and MCM2 upon 4h IAA. I can't see NELF "coordinate DNA replication".

We have changed the title as the reviewer suggested.

Concern 3)

I can't find the Figs S3k and S3l.

I'm not sure what the NVP-2 figure aims to show. TT-seq captures moving Pol II by definition. I still see the clear drop in signal at TES to indicate that most transcripts are cleaved at the expected site even in IAA. Yes, the drop is lower, but severe effect of termination is hard to agree on.

The NVP-2 figure (previous Extra Figure 3, see below) indicates the readthrough RNAs are actively transcribed and not stable RNA associated with chromatin, since short period treatment of a CDK9 inhibitor NVP-2 decreases TT-seq signals in downstream of gene (DoG) region (see blue (0min) and green (60 min NVP2 treatment)).

Effect of NVP-2 in DoG region on TT-seq, NELF-C-AID DLD-1 cells

Metagene profile of TT-seq with NVP-2 treatment (0, 30, 60min) in NELF-C loss. Signals of DoG are reduced by short time NVP-2 treatment. This indicates the TT RNAs in DoG is not stable.

We see the reviewer complains about our TT-seq profiles. However, TT-seq is widely used and many labs shows similar TT-seq metagene profile on protein coding genes to ours. FYI, see Cramer group's TT-seq metaprofile (PMID: 39934431). Their TT-seq profile also drops at PAS. This may be caused by Pol II pausing just before PAS although it has been currently unresolved in the transcription field.

Concern 4)

This reply does little to improve trust on POINT-seq technology. There is published record of overstatements regarding nascent nature in Pol II IP + RNA-seq assays. If empigen would 'destroy protein-RNA in chromatin and Pol II IP', there would be no Pol II associated RNA to sequence.

The authors' answer in concern 2 raises a simple way to back-up the nascent nature of the RNA

in POINT-seq:

- If the exon signal is from co-transcriptionally spliced nascent RNAs, there should be more signal at the first, and less signal at the last exons, while intronic Pol II density would remain roughly the same. In browser shots where the exon signal stays below the cut-off (e.g. Fig. 3e, 4a, 5d-e, 6a, S5 purple), I do not see such a pattern. Induced transcriptional waves could be used in these assays as well.

Unfortunately, the reviewer misunderstood why Empigen is employed in POINT-seq technology. As we mentioned in the response letter and published in our papers (PMID: 33735606, PMID: 28017589), Empigen destroys protein-RNA interactions such as spliceosome-splicing intermediates and microprocessor-pri-miRNAs in chromatin and Pol II IP. However, Empigen does not destroy Pol II-nascent RNA complex, since nascent RNA is from within Pol II active site (not associated with Pol II). See also our response to the concern 5.

The reviewer misunderstood kinetics of co-transcriptional splicing, too. Co-transcriptional splicing is not a sequential event from the first to the last introns. Especially, splicing event of the first intron is slower compared to internal introns in yeast (Neugebauer group PMID: 29903723) and human (our group PMID: 33735606). Efficiency of co-transcriptional splicing depends on many factors such as Pol II speed, intronic and exonic sequences, histone modifications, and DNA-RNA hybrid formation.

Here we emphasize again that our POINT-seq detect little contamination of steady RNAs. In our latest response letter, we explained that 4h treatment of CDK9 inhibitor DRB, which completely abolishes transcribing Pol II in chromatin, dramatically reduces POINT-seq signals (see Figure below, PMID: 33735606). This result indicates that POINT-seq detects little contamination of steady state mRNAs. If many steady state RNAs are contaminated in the POINT fraction, we should see lots of exonic signals in POINT-seq profile. Notably, we do not see them. To conclude, there is no or little RNA stability issue in our POINT-seq analyses.

CDK9 inhibitor DRB inhibits elongating Pol II. POINT-seq of 4h DRB treated HeLa cells detected little signals compared to DMSO control, indicating that POINT-seq mainly detects nascent RNAs.

Concern 5)

Answer to concern 5 is rather disturbing. The authors might confuse PRO-seq to G/PRO-cap or Start-seq techniques. PRO-seq does detect engaged Pol II complexes throughout the genome, including the DoG / termination window, analysed in multiple studies by multiple groups. In comparison, TT-seq and other in vivo labelling techniques only capture moving Pol II and - by definition - do not detect paused Pol II. My understanding is that many filter mNET-seq reads by length to avoid non-nascent RNAs, missing paused Pol II signal. I'm not sure why POINT-seq does not detect pausing.

All in all, the differences of these techniques can be used to understand the kinetics of Pol II at promoters (e.g. by Carmer lab in 2019, Gressel et al., Nat. Commun) or across genes, including termination (e.g. <https://www.rna-seqblog.com/tt-seq-maps-the-human-transient-transcriptome/>).

Very unfortunately the reviewer completely misunderstood about our original techniques, mNET-seq (PMID: 25910207) and POINT-seq (PMID: 33735606).

Our mNET-seq is now widely used in the field. For mNET-seq, chromatin DNA and Pol II-nascent RNA are digested with MNase leading to soluble chromatin and digested short RNAs (mainly 20-100nt) which are protected by Pol II elongating complex (see figure below). The short RNAs are isolated by Pol II antibody and then sequenced. After small RNA sequencing, 3'ends of RNA reads are bioinformatically analyzed as Pol II active site (plus 3'end of RNA

processing intermediates especially in the case of mNET-seq with Pol II CTD S5P).

In mNET-seq, chromatin DNA and RNA are digested by MNase to solubilize elongation Pol II complex. The native elongating Pol II complex is precipitated with Pol II CTD antibodies (against unphospho or phospho CTDs (Y1P, S2P, T4P, S5P, S7P)). After Pol II IP, the small RNAs, which are co-precipitated with Pol II, are purified and sequenced in illumina platform. 3'end of sequenced reads are bioinformatically mapped as Pol II active site (* red asterisk).

An advantage of our mNET-seq is detecting Pol II pausing and 3'end of co-transcriptionally cleaved nascent RNAs such as 5'ss of splicing intermediate at a single nucleotide resolution (PMID: 28017589, see figure below for mNET-seq/S5P, Empigen selectively washes out 5'ss signals.). Notably, Pol II associated complex such as spliceosome can be removed from Pol II by Empigen treatment in Pol II IP step, since Pol II-nascent RNAs which are derived from Pol II active site are Empigen resistant. Therefore, mNET-seq with Empigen treatment profiles only Pol II active site. Indeed, mNET-seq detects paused Pol II signals.

Nascent RNA-Pol II active site is Empigen resistant.

mNET-seq with Pol II CTD Ser5P (S5P) antibody detects Pol II active site and splicing intermediate (5'ss). Empigen is used in Pol II IP step. Pol II-Nascent RNA which is derived from Pol II active site is empigen resistant.

5'ss signals are empigen sensitive (indicated by orange arrows), but other signals are not. Also signals of mNET-seqs with other Pol II CTD antibodies (S2P, Y1P, T4P) are not affected by empigen, because all the signals are derived from Pol II active site.

For POINT-seq, we just add empigen in chromatin isolation step of mNET protocol. Empigen treated chromatin DNA can be easily digested with DNase, so nascent RNAs are not digested (nascent RNAs are intact). We named this technology, Polymerase-Intact Nascent Transcript (PMID: 33735606). The Intact nascent RNAs are fragmented for illumina sequencing platform i.e. POINT-seq. The RNA fragmentation step loses information of 3' end of nascent RNAs (Pol II active sites) in POINT-seq. That is why POINT-seq does not profile Pol II pausing. Instead, POINT-nano technology can detect Pol II pausing since nascent RNA 3' ends are polyadenylated by in vitro polyadenylation reaction and then the sequences are read from 3' end of the polyA tail where Nanopore adaptor is added.

Overall, we emphasize that our POINT-seq technology provide a pure nascent profile. It is unlikely to detect significant levels of steady RNAs, so we believe there is no or little RNA stability issue in POINT-seq analysis in this study. In addition, we reanalysed published data set of PRO-seq in NELF-B loss (PMID: 38197272) and detected Pol II transcription termination defect (new Figure EV3G). Furthermore, Shilatifard's group has just published independently of us a preprint (<https://doi.org/10.1101/2025.06.23.660200>) on NELF where they show a transcriptional readthrough with Iso-Seq and Pol II and Ser2 phosphorylation CHIP-seq. Altogether, these results support a transcriptionally active readthrough rather than a stabilization of readthrough RNAs.

Referee #1:

In this study, the authors investigate the function of the negative elongation factor (NELF) in DLD-1 colorectal cancer cells by using an auxin-inducible protein degradation (AID) system to deplete the NELF-C subunit. The best-known function of NELF is in implementing a promoter-proximal pause in elongation by RNA polymerase II (Pol II)-a key regulatory step in the transcription cycle. Surprisingly, they find that acute loss of NELF-C causes inefficient termination by Pol II, and that the resulting read-through transcription can engender transcription-replication conflicts, which may in turn block the cell division cycle. Consistent with termination being affected, loss of NELF-C reduced the occupancy of termination factors CSTF64 and XRN2 through an unknown mechanism. To explain the defect, the authors invoke an increase in elongation rate. They base this inference on results obtained by two different methods: one direct (TT-seq in the presence of the CDK9 inhibitor NVP2) and one indirect (calculating the ratio of TT-seq signals to either POINT-seq or PRO-seq signals). To attempt to explain the cell-cycle arrest, they analyzed previously published Pu-seq data, which measured the genome-wide usage of replicative DNA polymerases, and found that regions of active Pol II transcription and DNA replication initiation are mutually exclusive in unperturbed cells, whereas regions undergoing read-through transcription after acute loss of NELF-C overlap with replication initiation regions. The occurrence of transcription-replication conflicts was supported by in-situ proximity ligation assays (PLA) between elongating (pSer2-containing) Pol II and the replisome components MCM2 and PCNA, and by BrdU-IP-seq to map ongoing DNA synthesis. The results suggest a new role of NELF in transcription termination at a broad class of protein-coding genes. It should be noted, however, that previous work by the Handa lab had implicated NELF in the non-canonical termination of transcripts encoding replication-dependent histone (RDH) genes (10.1016/j.molcel.2007.04.011, which the authors should cite). The responses to the previous reviewer have mostly satisfied concerns about the use of POINT-seq, which seems justified, and for the most part, the experiments were well designed and executed. Some of the authors' interpretations are questionable, however, and many of their conclusions would need to be better supported by data (or re-evaluated) before I could recommend publication.

General concerns:

1. The authors posit that read-through by Pol II is due to increased elongation rate. However, none of the analyses presented unambiguously indicates faster elongation (see specific concerns below). This really needs to be clearly demonstrated, since it is unclear how NELF could be influencing elongation rate; unlike DSIF (SPT4/SPT5), NELF is not thought to be a component of productive transcription elongation complexes.

2. One would think the transcription-replication conflicts observed here would be a universal consequence of any perturbation that causes widespread read-through transcription by Pol II, so it's unclear whether preventing these conflicts is a specific role of NELF. Can the authors comment on this, and perhaps re-analyze data from previous studies that perturbed functions of, say, the PNUTS-PP1 complex needed for efficient termination (ref. 14, DOI: 10.1016/j.molcel.2024.10.046, DOI: 10.1016/j.molcel.2024.10.045), to ask whether transcription read-through extended into replication initiation zones in those cases?
3. Can the authors comment on the fact that G1 cell-cycle arrest due to NELF-C depletion is completely reversible upon washout of the AID inducer IAA (somewhat surprising if the cells have undergone significant replication stress)? Moreover, this reversibility provides an opportunity to establish at least a temporal correlation between the phenotypes the authors are trying to link mechanistically: Is the termination defect (excess read-through transcription) resolved with similar kinetics as the cell-cycle arrest (Fig. 2H)?
4. Besides simply citing the paper from the Handa lab mentioned above, the authors need to consider whether impaired histone production, which is tightly restricted to S phase, provides an alternative explanation for some of the phenotypes they see, such as T-R conflicts and arrest in G1 or early S.

Major, specific concerns:

1. The analysis in Figure 1 is of questionable relevance to the main thrust of the study and could be moved to the Supplement (perhaps to make room for a suitably revised Fig. EV4, which is more central). The attempts to link NELF-C to cancer add little value and lead to overstatements, such as "This result suggests that the Pol II transcription program in READ and COAD is highly dependent on NELF-CD protein" (P.6). The main point I would take from this analysis is that cancer cells have increased dependence on many components of the Pol II transcription machinery-not a new finding at this point.
2. The metagene plots in Fig. 3C suggest widespread read-through by Pol II after NELF-C degradation, but reflect an averaged result. Can the authors address how widespread this phenomenon is, i.e., percentages of genes with and without read-through transcription? They should provide statistics and analyze the features (e.g. length, expression level, pause index) that might distinguish genes prone to read-through from unaffected genes.
3. I find it difficult to reconcile the results in Fig. EV4B, suggesting that NELF-C depletion increases pause index genome-wide, with those in Fig. EV4A-a metagene plot that seems to show decreased amplitude of the downstream-shifted Pol II peak in the TSS region-and with the known functions of NELF in enforcing or stabilizing the promoter-proximal pause.
4. The use of TT-seq with NVP2 treatment to measure transcription elongation rate is not ideal. First, I take issue with the statement that "the elongation step" is "[not] affected by

CDK9 inhibition" (p. 10). While it is true that the study by the Lis lab cited (ref. 43) measured elongation rates of Pol II that had already cleared the pause release step prior to addition of a CDK9 inhibitor (flavopiridol), there was no comparison with rates when CDK9 was active. Subsequent work has clearly established that CDK9-dependent phosphorylation of SPT5 (and possibly other substrates) regulates elongation rate (ref. 14, DOI: 10.1038/s41467-018-03006-4, DOI: 10.1016/j.molcel.2025.03.021). Moreover, it is far from obvious that the methods used here detected any difference in elongation rate between NELF-depleted and control cells; the position of the wavefront of Pol II transcription is not obviously changed either in the browser track or in the metagene plots shown in Fig. EV4D-E, even though the amplitude of the wave is increased. And while the increased ratio of TT-seq to POINT-seq/PRO-seq signals is consistent with an increased elongation rate, it could also be due to increased initiation frequency (not unlikely, given the antagonistic relationship between pausing and initiation-see ref. 44). Distinguishing between these two scenarios requires a (different) direct assay, such as DRB block and release followed by TT-seq, CHIP-seq or CUT&RUN (see, for example, DOI: 10.1038/s41596-019-0262-3).

Minor concerns:

1. Figure EV1C: The proteomic analysis reveals small, albeit apparently significant, increases in NELF subunit abundance in colon cancer, but it is not clear how meaningful this is. It would help to know the sample sizes (n) and the actual p values (not simply asterisks).
2. The description of the dTAG system in different places in the manuscript is inaccurate. For example, on p. 10, the authors state, "PRO-seq of NELF-B-dTAGv1 in HAP1 cells shows a widespread termination defect...after 2h of VHL treatment (Fig. EV3G)", but dTAGv-1 is the name of the small-molecule degradation inducer, dTAG is the colloquial name given to the protein tag, and VHL is the E3 ligase adaptor protein that dTAGv-1 recruits to the target protein.
3. Figure EV2E: Why does addition of IAA in the parental cells lead to reduction in γ H2AX foci?

Response letter to Reviewer's comments

Referee #1:

In this study, the authors investigate the function of the negative elongation factor (NELF) in DLD-1 colorectal cancer cells by using an auxin-inducible protein degradation (AID) system to deplete the NELF-C subunit. The best-known function of NELF is in implementing a promoter-proximal pause in elongation by RNA polymerase II (Pol II)-a key regulatory step in the transcription cycle. Surprisingly, they find that acute loss of NELF-C causes inefficient termination by Pol II, and that the resulting read-through transcription can engender transcription-replication conflicts, which may in turn block the cell division cycle. Consistent with termination being affected, loss of NELF-C reduced the occupancy of termination factors CSTF64 and XRN2 through an unknown mechanism. To explain the defect, the authors invoke an increase in elongation rate. They base this inference on results obtained by two different methods: one direct (TT-seq in the presence of the CDK9 inhibitor NVP2) and one indirect (calculating the ratio of TT-seq signals to either POINT-seq or PRO-seq signals). To attempt to explain the cell-cycle arrest, they analyzed previously published Pu-seq data, which measured the genome-wide usage of replicative DNA polymerases, and found that regions of active Pol II transcription and DNA replication initiation are mutually exclusive in unperturbed cells, whereas regions undergoing read-through transcription after acute loss of NELF-C overlap with replication initiation regions. The occurrence of transcription-replication conflicts was supported by in-situ proximity ligation assays (PLA) between elongating (pSer2-containing) Pol II and the replisome components MCM2 and PCNA, and by BrdU-IP-seq to map ongoing DNA synthesis. The results suggest a new role of NELF in transcription termination at a broad class of protein-coding genes. It should be noted, however, that previous work by the Handa lab had implicated NELF in the non-canonical termination of transcripts encoding replication-dependent histone (RDH) genes (10.1016/j.molcel.2007.04.011, which the authors should cite). The responses to the previous reviewer have mostly satisfied concerns about the use of POINT-seq, which seems justified, and for the most part, the experiments were well designed and executed. Some of the authors' interpretations are questionable, however, and many of their conclusions would need to be better supported by data (or re-evaluated) before I could recommend publication.

We are pleased to hear that the reviewer feels the response to the previous reviewer in *Nature Communications* has satisfied his/her concerns about our nascent RNA analysis (POINT-seq). Here we have carefully provided point-by-point responses to address his/her concerns.

Accordingly, we have reorganized figures and modified the text which are highlighted in green.

General concerns:

1. The authors posit that read-through by Pol II is due to increased elongation rate. However, none of the analyses presented unambiguously indicates faster elongation (see specific concerns below). This really needs to be clearly demonstrated, since it is unclear how NELF could be influencing elongation rate; unlike DSIF (SPT4/SPT5), NELF is not thought to be a component of productive transcription elongation complexes.

As the reviewer mentioned, it remains unclear how NELF could affect Pol II elongation rate. In case that the reviewer might have missed it, we describe what we discussed for a few possible mechanisms in the previous manuscript. In short, we think Pol II elongation rate may be controlled at Pol II pausing at promoter proximal site (1st or 2nd pausing) or Pol II pausing zone that may contribute to RNA quality control recently proposed by Torben Jensen group (PMID: 39366352), while this needs to be investigated in the future study.

2. One would think the transcription-replication conflicts observed here would be a universal consequence of any perturbation that causes widespread read-through transcription by Pol II, so it's unclear whether preventing these conflicts is a specific role of NELF. Can the authors comment on this, and perhaps re-analyze data from previous studies that perturbed functions of, say, the PNUMS-PP1 complex needed for efficient termination (ref. 14, DOI: 10.1016/j.molcel.2024.10.046, DOI: 10.1016/j.molcel.2024.10.045), to ask whether transcription read-through extended into replication initiation zones in those cases?

Other studies have suggested that Pol II transcription readthrough may be involved in TRC as follows. Recently, Tian group reported that CPSF73 inhibitor, JTE-607 induces DNA damage, cell death and transcription termination defect, assuming JTE-607 induces TRC (PMID: 37528120). Nevertheless, this may not be the case of TRC, since JTE-607 increases S-phase population. Importantly, Altmeyer group demonstrated that siRNA depletion of WDR33, a CPA factor, impairs transcription termination and replication fork speed leading to replication stress (PMID: 30639241). Furthermore, we previously found that long noncoding transcription was upregulated and extended by SPT6 siRNA depletion, and the dysregulated Pol II reaching replication initiation region (Nojima, Tellier et al., *Mol Cell* 2018 PMID: 30449723). These studies suggest that TRC is regulated by Pol II transcription termination. However, a possibility that the replication stress is caused by indirect effects of long period siRNA depletion (48-72h) cannot be ruled out. As the reviewer suggested, we also re-analyzed previously published TT-seq data in PNUMS-dTAG DLD-1 cells (PMID: 39603240). Their data shows transcription termination defect (PMID: 39603240). We reanalyzed their data and detected the increased TT-

seq signals in replication initiation zones after PNUTS depletion (new Fig. EV7D). This result suggests that PNUTS loss could also affect cell cycle and DNA replication via transcriptional readthrough, although it is unclear how much they are affected. Thus, as far as we know, no clear evidence of a connection between Pol II transcription termination and TRC has been shown. In this study, we finally provide the compelling evidence of Pol II transcription termination defect-associated TRC using POINT-seq, proximity ligation assay (PLA), and DNA replication assay after a short period protein degradation (4h).

Summarising these results, TRC is not specifically caused by NELF-C depletion. However, cell cycle defect and DNA replication suppression may be dependent of transcription readthrough level.

3. Can the authors comment on the fact that G1 cell-cycle arrest due to NELF-C depletion is completely reversible upon washout of the AID inducer IAA (somewhat surprising if the cells have undergone significant replication stress)? Moreover, this reversibility provides an opportunity to establish at least a temporal correlation between the phenotypes the authors are trying to link mechanistically: Is the termination defect (excess read-through transcription) resolved with similar kinetics as the cell-cycle arrest (Fig. 2H)?

We detected that NELF-C protein level was partially recovered (36% in tubulin normalized protein level) by washing out IAA (Post-T 24h, new Fig2B). In this case, we detected fully recovery of S phase in cell cycle (new Fig 2D). In support of this, a preprint (<https://doi.org/10.1101/2025.06.23.660200>) by Ali Shilatifard's group is also showing that longer NELF-C degradation (96h) is still reversible. Notably, our time-course experiment of NELF-C depletion demonstrates that a partially depleted NELF-C protein (20min IAA, 40% in Pol I normalized protein level) induces no or a little transcription readthrough (new Figs EV3A-B). This result suggests that the termination defect may be resolved in the Post-T 24 condition.

4. Besides simply citing the paper from the Handa lab mentioned above, the authors need to consider whether impaired histone production, which is tightly restricted to S phase, provides an alternative explanation for some of the phenotypes they see, such as T-R conflicts and arrest in G1 or early S.

We added POINT-seq and pA+RNA-seq data on replication-dependent histone (RDH) genes. Consistent with the finding of the Handa lab, our POINT-seq and pA+RNA-seq detect increases of transcriptional readthrough in most of RDH genes and polyadenylated RDH RNAs after NELF depletion, respectively. The increase of RDH gene transcription readthrough after NELF

depletion may be involved in TRC at RDH gene cluster. Although RDH pA+RNA levels are significantly upregulated by NELF loss, RDH protein level is downregulated after 24h NELF depletion due to reduction of S phase population. This result suggests that pA+RDH RNAs are not efficiently translated. Importantly, RDH protein level is unaffected after 4h NELF depletion. This result supports our proposed model that transcription termination defect followed by TRC and DNA replication suppression impairs cell cycle and cellular growth of NELF depleted cells. However, we do not exclude a possibility that reduction of RDH protein level may affect cell cycle progression at later stage. We added the results in new Fig EV9 and discuss about RDH expression in Discussion part.

Major, specific concerns:

1. The analysis in Figure 1 is of questionable relevance to the main thrust of the study and could be moved to the Supplement (perhaps to make room for a suitably revised Fig. EV4, which is more central). The attempts to link NELF-C to cancer add little value and lead to overstatements, such as "This result suggests that the Pol II transcription program in READ and COAD is highly dependent on NELF-CD protein" (P.6). The main point I would take from this analysis is that cancer cells have increased dependence on many components of the Pol II transcription machinery-not a new finding at this point.

As the reviewer suggested, we moved most of the data in previous Fig. 1 to new Fig. EV1.

2. The metagene plots in Fig. 3C suggest widespread read-through by Pol II after NELF-C degradation, but reflect an averaged result. Can the authors address how widespread this phenomenon is, i.e., percentages of genes with and without read-through transcription? They should provide statistics and analyze the features (e.g. length, expression level, pause index) that might distinguish genes prone to read-through from unaffected genes.

We agree with the reviewer's comment. We thought that his type of analysis may be useful to dissect Pol II transcription termination. However, we have previously noticed the gene classification was problematic to distinguish readthrough genes. In the previous round of revision, unaffected and affected genes were classified with a particular threshold value on termination index in small gene region (for example TES+2.5kb to avoid including overlapping genes in the analysis), but this classification was hugely affected by readthrough length, distance to the next downstream gene, and gene expression level. Indeed, this was affecting downstream analysis. Therefore, we decided to employ non-biased (no classification) approach.

However, we have added back an analysis of readthrough transcription, gene length, gene expression level, and pause index on a set of non-overlapping (2.5 kb gene-free window

before/after TSS/poly(A) site) and expressed genes (new Fig EV5). From the POINT-seq analysis, 28% of this set of genes is not associated with transcriptional readthrough and these genes are on average longer than the 72% of the genes associated with a transcriptional readthrough. No difference in expression level or Pol II pausing between the two group of genes is observed before or after IAA treatment.

3. I find it difficult to reconcile the results in Fig. EV4B, suggesting that NELF-C depletion increases pause index genome-wide, with those in Fig. EV4A-a metagene plot that seems to show decreased amplitude of the downstream-shifted Pol II peak in the TSS region-and with the known functions of NELF in enforcing or stabilizing the promoter-proximal pause.

Previous Fig EV4A is showing the amount of paused Pol II while previous Fig EV4B provides the analysis of pausing index. Pausing index is calculated based on pausing level ratioed to the amount of elongating Pol II across the gene body. In a situation where the amount of paused Pol II is reduced, like following NELF-C degradation, if the amount of PRO-seq signal across the gene body is not reduced as much as the amount of paused Pol II, which is also the case for NELF-C loss (see new Fig EV4A), this will result in an increased Pol II pausing index as there is relatively more Pol II signal on promoter-proximal region than in the gene body following IAA 4h treatment than in DMSO. We also note that other studies using NELF degran approaches have also observed that the loss of NELF does not block pausing of Pol II (PMID: 32155413, 35981753, 38197272, 39416036).

4. The use of TT-seq with NVP2 treatment to measure transcription elongation rate is not ideal. First, I take issue with the statement that "the elongation step" is "[not] affected by CDK9 inhibition" (p. 10). While it is true that the study by the Lis lab cited (ref. 43) measured elongation rates of Pol II that had already cleared the pause release step prior to addition of a CDK9 inhibitor (flavopiridol), there was no comparison with rates when CDK9 was active. Subsequent work has clearly established that CDK9-dependent phosphorylation of SPT5 (and possibly other substrates) regulates elongation rate (ref. 14, DOI: 10.1038/s41467-018-03006-4, DOI: 10.1016/j.molcel.2025.03.021). Moreover, it is far from obvious that the methods used here detected any difference in elongation rate between NELF-depleted and control cells; the position of the wavefront of Pol II transcription is not obviously changed either in the browser track or in the metagene plots shown in Fig. EV4D-E, even though the amplitude of the wave is increased. And while the increased ratio of TT-seq to POINT-seq/PRO-seq signals is consistent with an increased elongation rate, it could also be due to increased initiation frequency (not unlikely, given the antagonistic relationship between pausing and initiation-see ref. 44). Distinguishing between these two scenarios requires a (different) direct assay, such as DRB

[block and release followed by TT-seq, ChIP-seq or CUT&RUN \(see, for example, DOI: 10.1038/s41596-019-0262-3\).](https://doi.org/10.1038/s41596-019-0262-3)

We appreciate the comment and toned down our conclusion of Pol II elongation rate experiments with additions of discussion about a potential limitation of our TT-seq with NVP2 (CDK9 inhibitor) experiment. However, we respectfully disagree on the reviewer's claim that the use of NVP2 is not ideal.

First, many previous studies revealed a consistent rate of elongation of ~2 kb/min, regardless of the methods. For instance, DRB/TTchem-seq, DRB/GRO-seq, and 4sUDRB-seq showed 2.07 (median), 3.1 (median), 3.5 (average) kb/min, whereas GRO-seq with Flavopiridol (FP) showed 2.1 (median), 2.4 (average) kb/min, respectively (PMID: 31915390, PMID: 24836610, PMID: 24887486, PMID: 24843027). It appears that the rates slightly vary depending on the experimental systems and computational approaches, rather than the use of DRB block/release or FP block. It should be noted that the NVP2 block method which was employed in this study has been used to study the rate of elongation (PMID: 34480849). Second, the DRB block/release method is suitable to study a role of elongation factors during early elongation, such as a role of SPT5 phosphorylation by CDK9, but does not directly address any mechanisms during late elongation. Notably, the paper cited by the reviewer stated that "The other population, already beyond this checkpoint, may no longer require Cdk9 to maintain its rate of transcription" (PMID: 29416031). This study, alongside the GRO-seq + FP study (PMID: 24843027), supports the idea that establishment of elongation-potent Pol II by CDK9 takes place during early checkpoint, but not during late elongation. In contrast, the NVP2 (or FP) block method is a direct approach to address the mechanisms underlying late elongation, given that it detects a defect in elongation of Pol II that has already passed early checkpoint. However, we could also note that SPT5 is also required for the rate of elongation beyond early checkpoint, implying that CDK9-independent function of SPT5 may contribute to stimulate the rate of elongation throughout gene bodies (e.g. via a DNA-RNA clamp (PMID: 28892040)). Also, we agree with the reviewer that the difference in the rate of elongation between NELF-depleted and control cells is small, compared to the ones observed in CDK9 inhibition (PMID: 29416031). We therefore toned down our claim regarding the impact of faster elongation rates on transcription readthrough in the revised manuscript.

We think it would be interesting to examine initiation frequency in NELF-depleted cells, although it is out of the scope in the current study. While recruitment of NELF/DSIF follows a release of initiation factors TFIIH/TFIIE (PMID: 23064645, PMID: 38604172), it is possible that loss of NELF at promoter-proximal regions may interfere new initiation, given a role of pausing in blocking initiation (PMID: 28504701). The DRB block/release method is suitable for

this type of analysis as previously shown (PMID: 24887486). We added this in the discussion part.

Minor concerns:

1. Figure EV1C: The proteomic analysis reveals small, albeit apparently significant, increases in NELF subunit abundance in colon cancer, but it is not clear how meaningful this is. It would help to know the sample sizes (n) and the actual p values (not simply asterisks).

We added the sample size and p values in the proteomic analysis (new Fig.EV1E).

2. The description of the dTAG system in different places in the manuscript is inaccurate. For example, on p. 10, the authors state, "PRO-seq of NELF-B-dTAGv1 in HAP1 cells shows a widespread termination defect...after 2h of VHL treatment (Fig. EV3G)", but dTAGv-1 is the name of the small-molecule degradation inducer, dTAG is the colloquial name given to the protein tag, and VHL is the E3 ligase adaptor protein that dTAGv-1 recruits to the target protein. We fixed it.

3. Figure EV2E: Why does addition of IAA in the parental cells lead to reduction in γ H2AX foci?

γ H2AX foci number might be affected by IAA in particular cell cycle status, but we do not have any evidence for that. Importantly, increased γ H2AX signals of western blot analysis is consistent with immunofluorescence result after addition of IAA in NELF-C-AID DLD-1 cells. Therefore, we removed the data of the parental cells (new Fig. EV2C).

Dear Dr. Nojima

Thank you for the re-submission of your revised manuscript. We have now received the enclosed report. The referee accepts the technical standard of the paper, but continues to feel that the paper is textually overinterpreted. I think it is everyone's interest to formulate the conclusions in a scholarly and robust manner which will not lead to misinterpretation by our readers. We therefore encourage you to take the well intentioned comment to heart and add the minor textual changes requested, assuming you agree with the referee - if not, I suggest that we set up a zoom-call to discuss the matter.

Please also note the minor cosmetic callout issue noted by the referee.

best wishes,

Bernd Pulverer

~~~~~  
Bernd Pulverer, Ph.D.  
Chief Editor, EMBO Reports  
EMBO  
Meyerhofstrasse 1, D-69117 Heidelberg  
Tel: +4962218891501  
bernd.pulverer@embo.org  
~~~~~

Referee #1:

This is a revised version of a manuscript I reviewed previously for the EMBO Journal. In it, the authors have addressed most of my concerns and strengthened the paper. I still have concerns with the purported analyses of Pol II elongation rates. I am willing to set aside my methodological qualms about the TTseq analysis with NVP-2, which detected modestly increased elongation speed at a small number of genes, apparently detected by computational analysis in Fig. EV6D but not clearly visible in the browser tracks or metagene plots in Fig. EV6A,B. However, one cannot safely infer an increased "elongation velocity" solely from the so-called elongation index, a ratio of transcriptional output (TT-seq) to Pol II occupancy (POINT-seq or PRO-seq), as implied by the cartoon in Fig. 4A. Transcriptional output is also influenced by initiation frequency, which, the authors now conclude, is indeed increased by NELF-C loss at 4 hr of IAA treatment (p. 10, Fig. EV4C). Fortunately, showing an effect on elongation rate is not central to the story, so de-emphasizing it, e.g. by replacing "elongation velocity" (or "rate") with "elongation index," will not significantly detract from the major conclusions, in my opinion.

Remaining, specific concern:

1. On pages 12-13, there are numerous call-outs to panels B-E of Fig. 6 that should be to Fig. 4.

Response letter to Reviewer's comments

Referee #1:

This is a revised version of a manuscript I reviewed previously for the EMBO Journal. In it, the authors have addressed most of my concerns and strengthened the paper. I still have concerns with the purported analyses of Pol II elongation rates. I am willing to set aside my methodological qualms about the TTseq analysis with NVP-2, which detected modestly increased elongation speed at a small number of genes, apparently detected by computational analysis in Fig. EV6D but not clearly visible in the browser tracks or metagene plots in Fig. EV6A,B. However, one cannot safely infer an increased "elongation velocity" solely from the so-called elongation index, a ratio of transcriptional output (TT-seq) to Pol II occupancy (POINT-seq or PRO-seq), as implied by the cartoon in Fig. 4A. Transcriptional output is also influenced by initiation frequency, which, the authors now conclude, is indeed increased by NELF-C loss at 4 hr of IAA treatment (p. 10, Fig. EV4C). Fortunately, showing an effect on elongation rate is not central to the story, so de-emphasizing it, e.g. by replacing "elongation velocity" (or "rate") with "elongation index," will not significantly detract from the major conclusions, in my opinion.

We truly appreciate the reviewer's comments which have improved our manuscript. We have replaced "elongation rate" with "elongation activity (EA)" and "elongation velocity" with "elongation index (EI)" in text and figures (Figs. 4 and EV6) to further tone them down as the reviewer suggested.

Remaining, specific concern:

1. On pages 12-13, there are numerous call-outs to panels B-E of Fig. 6 that should be to Fig. 4.

We have fixed it.

Dr. Takayuki Nojima
Kyushu University
Medical Institute of Bioregulation
3-1-1 Maidashi
Higashi-ku
Fukuoka 812-8582
Japan

Dear Dr. Nojima,

I am delighted to accept your manuscript for publication online and in the next available issue of EMBO reports. Thank you for your contribution to our journal.

You may qualify for financial assistance for your publication charges - either via a Springer Nature fully open access agreement or an EMBO initiative. Check your eligibility: <https://link.springer.com/journal/44319/how-to-publish-with-us>

Yours sincerely,

Bernd Pulverer

~~~~~  
Bernd Pulverer, Ph.D.  
Chief Editor, EMBO Reports  
EMBO  
Meyerhofstrasse 1, D-69117 Heidelberg  
Tel: +4962218891501  
[bernd.pulverer@embo.org](mailto:bernd.pulverer@embo.org)  
~~~~~

>>> Please note that it is EMBO Reports policy for the transcript of the editorial process (containing referee reports and your response letter) to be published as an online supplement to each paper. If you do NOT want this, you will need to inform the Editorial Office via email immediately. More information is available here: <https://link.springer.com/partners/embo-press/editorial-policies#Peer%20review>